# Auto-inhibition of PRC2 by the broadly expressed long isoform of AEBP2

Marlena Mucha [1,10], Zhihao Lai [2,10], Nicholas J McKenzie[2,10], Francesca Matrà[1,10], Marion Boudes [2], Sarena F Flanigan[2], Maria Teresa Alejo-Vinogradova[3], Craig Monger[1], Qi Zhang [2], Darragh Nimmo[1], Evan Healy[1,2], Ademar J Silva[1], Daniel Angelov [1], David M Reck [1], Gráinne Holland [1], Zeynep Eda Atmaca [4], Helen E King [5,6], Maeve Hamilton[1], Eleanor Glancy[1], James Nolan[1], Robert J Weatheritt [5,7], Oliver Bell [4], Michiel Vermeulen [3,8], Chen Davidovich [2,9,11✉] & Adrian P Bracken [1,11✉]

## Abstract

**Polycomb Repressive Complex 2 (PRC2) is an essential chromatin regulator responsible for mono-, di- and tri-methylating H3K27. Control of PRC2 activity is a critical process in development and disease, yet no inhibitory cofactor has been identified in somatic cells. Here, we show that the alternative isoforms of its accessory subunit AEBP2, namely AEBP2$^S$ (short) and AEBP2$^L$ (long), perform opposite functions in modulating PRC2 activity. Contrary to prior assumptions that AEBP2 enhances PRC2 function, we find that the widely expressed AEBP2$^L$ isoform inhibits it. AEBP2$^L$ is expressed throughout embryogenesis and adulthood and inhibits PRC2 DNA binding, histone methyltransferase activity, and binding to target genes. In contrast, AEBP2$^S$, expressed during early embryogenesis, promotes PRC2 DNA-binding activity and is essential for de novo repression of target genes during the transition from naïve to primed pluripotency. Mechanistically, through high-resolution cryo-EM and mutagenesis, we show that the recently evolved, negatively charged N-terminal region of AEBP2$^L$ inhibits PRC2. We propose a scenario in which the N-terminus of AEBP2$^L$ arose in vertebrates to restrain PRC2 activity in somatic cells.**

**Keywords** Polycomb; Trithorax; AEBP2; PRC2; Cryo-EM
**Subject Categories** Chromatin, Transcription & Genomics; Development

## Introduction

Polycomb repressive complex 2 (PRC2) is a multiprotein chromatin repressor complex responsible for mono-, di- and tri-methylation of H3K27 (H3K27me1/2/3) (Margueron and Reinberg, 2011; Laugesen et al, 2019). Core PRC2 contains one of two catalytic subunits: EZH1 or EZH2, together with SUZ12 and EED (Schuettengruber et al, 2017). The recruitment of PRC2 to chromatin is mediated by accessory subunits (Laugesen et al, 2019; Glancy et al, 2021; Bracken et al, 2019). These subunits bind to the PRC2 core complex, and their specific combinations result in different types of holo-PRC2 complexes (Hauri et al, 2016; Laugesen et al, 2019). Holo-PRC2 complexes are targeted to chromatin via different mechanisms, and they synergise to promote H3K27me3 deposition (Healy et al, 2019; Højfeldt et al, 2019). While the stoichiometry of the accessory subunits of PRC2 dynamically changes during cell differentiation (Kloet et al, 2016; Oliviero et al, 2016), it is not known whether PRC2 accessory subunits can restrain chromatin binding.

EZHIP (EZH Inhibitory Protein) is a recently discovered negative regulator of PRC2 histone methyltransferase activity (Hübner et al, 2019; Jain et al, 2019; Piunti et al, 2019; Ragazzini et al, 2019). The expression of EZHIP is normally restricted to germinal cells, but it can become aberrantly expressed in a rare form of paediatric glioma, called posterior fossa group A ependymoma (PFA) (Hübner et al, 2019). PFA ependymomas are characterised by a global reduction of H3K27me3, commonly driven by EZHIP (Jenseit et al, 2022). EZHIP is thought to inhibit the spreading of H3K27me3 following the initial recruitment of PRC2 to chromatin, which then alters gene expression to promote tumorigenesis (Hübner et al, 2019; Jain et al, 2019; Piunti et al, 2019; Ragazzini et al, 2019). The EZHIP gene arose during placental

[1]Smurfit Institute of Genetics, Trinity College Dublin, Dublin, Ireland. [2]Department of Biochemistry and Molecular Biology, Biomedicine Discovery Institute, Faculty of Medicine, Nursing and Health Sciences, Monash University, Clayton, VIC, Australia. [3]Department of Molecular Biology, Faculty of Science, Radboud Institute for Molecular Life Sciences, Oncode Institute, Radboud University Nijmegen, Nijmegen 6525 GA, The Netherlands. [4]Departments of Biochemistry and Molecular Medicine, and Stem Cell and Regenerative Medicine, Norris Comprehensive Cancer Center, Keck School of Medicine, University of Southern California, Los Angeles, CA 90033, USA. [5]EMBL Australia, Garvan Institute of Medical Research, Sydney, NSW, Australia. [6]St. Vincent Clinical School, University of New South Wales, Darlinghurst, NSW, Australia. [7]School of Biotechnology and Biomolecular Sciences, University of New South Wales, Sydney, NSW, Australia. [8]Division of Molecular Genetics, The Netherlands Cancer Institute, Amsterdam, The Netherlands. [9]EMBL-Australia, Clayton, VIC, Australia. [10]These authors contributed equally: Marlena Mucha, Zhihao Lai, Nicholas J McKenzie, Francesca Matrà. [11]These authors contributed equally: Chen Davidovich, Adrian P Bracken. ✉E-mail: chen.davidovich@monash.edu; adrian.bracken@tcd.ie

mammal evolution, and its discovery raises an intriguing question: do additional negative regulators of PRC2 exist in non-germinal cells? Such a discovery could provide insights into how the chromatin occupancy of PRC2 may be adjusted to preserve stem cell potency or could be dysregulated in disease.

AEBP2 is an accessory subunit of PRC2, and its exact function remains elusive. Previous studies have produced apparently contradictory results regarding its ability to bind chromatin and enhance PRC2 activity in vitro. Some suggest AEBP2 is a potential transcriptional repressor (He et al, 1999; Cao et al, 2002; Kim et al, 2009; Ciferri et al, 2012; Kalb et al, 2014; Lee et al, 2018; Kasinath et al, 2021), while others report that it is capable of promoting gene activation (Grijzenhout et al, 2016; Conway et al, 2018; Leicher et al, 2020; Lin et al, 2021). Strikingly, *Aebp2* null mice exhibit anterior homeotic transformations, a phenotype typically associated with mutations in genes encoding the Trithorax group proteins (Ringrose and Paro, 2004; Grijzenhout et al, 2016). This finding supports the hypothesis that AEBP2 antagonises Polycomb function as the Trithorax group proteins oppose Polycomb repressive activity (Schuettengruber et al, 2017). Another report showed that the two main isoforms of AEBP2, the short (AEBP2$^S$) and the long (AEBP2$^L$), bind the promoter of the *Snai2* gene in mouse thymus tissue, yet AEBP2$^S$ promotes its repression, whereas AEBP2$^L$ promotes its transcription (Kim et al, 2015). These studies indirectly suggest contrasting functions for the two AEBP2 isoforms, although the extent of this phenomenon, its biological relevance, and the mechanisms behind it are unknown.

In this study, we tested the hypothesis that the AEBP2$^S$ and AEBP2$^L$ isoforms have opposite functions in modulating PRC2-mediated gene repression. We show that AEBP2$^S$, but not AEBP2$^L$, is required for the correct repression of de novo Polycomb target genes during embryonic stem cell differentiation. While the expression of AEBP2$^S$ is restricted to early development, AEBP2$^L$ is widely expressed in both embryonic and adult tissues. We identified mammalian-specific acidic tracts within the distinctive N-terminus of AEBP2$^L$ that confer its ability to inhibit PRC2. AEBP2$^L$ uses its N-terminus to impair PRC2 binding to DNA in vitro and at Polycomb target genes in cells. Taken together, we propose a model wherein AEBP2$^L$ functions to restrain PRC2 activity in somatic cells.

## Results

### AEBP2$^L$ inhibits the DNA-binding and histone methyltransferase activities of PRC2 in vitro

Pursuant to the idea that AEBP2 functions are isoform-specific, we wished to directly compare isoform-specific effects on the DNA binding and histone methyltransferase activities of PRC2. Transcription from the *Aebp2* locus in mouse and human results in two main categories of isoforms, which we named AEBP2-Long (AEBP2$^L$) and AEBP2-Short (AEBP2$^S$). In humans, there are three confirmed isoforms, which in this study are referred to as AEBP2$^{L(iso1)}$ (Q6ZN18-1), AEBP2$^{L(iso2)}$ (Q6ZN18-2) and AEBP2$^S$ (Q6ZN18-3) (Fig. 1A). The two AEBP2$^L$ isoforms are nearly identical, with AEBP2$^{L(Iso2)}$ lacking 14 amino acids in the C-terminus of the protein (Fig. EV1A).

We expressed and purified core PRC2 (EZH2, SUZ12, EED and RBBP4) with and without AEBP2$^S$ or AEBP2$^L$ (Fig. 1B) and performed electrophoretic mobility shift assays (EMSAs) to quantify their DNA-binding activity (Figs. 1C,D and EV1B). These experiments were focused on PRC2-AEBP2, without JARID2, as a reductionist approach (more below on experiments with PRC2-AEBP2-JARID2; Fig. 6). It is important to note that the PRC2.2 complex containing AEBP2, but lacking JARID2, remains biologically relevant. This is partly because in cell types such as lymphocytes, keratinocytes, and endothelial cells, JARID2 is cleaved into a shorter isoform that cannot bind to PRC2 (Al-Raawi et al, 2019). In addition, in line with the variable expression of JARID2 across cell types, endogenous proteomics of PRC2 in human diploid fibroblasts detected AEBP2, but not JARID2 (McCole et al, 2025). We used a fluorescently labelled DNA probe designed to form a 24-base-long dsDNA with a sequence from a CpG island of the *CDKN2B* locus (Figs. 1C,D and EV1B). PRC2–AEBP2$^S$ had the highest affinity to DNA and was the only complex for which a complete binding curve could be recorded ($K_d = 45.5 \pm 3.6$ nM, Hill $= 2.3 \pm 0.4$; Fig. 1D). Conversely, core PRC2 alone, PRC2–AEBP2$^{L(iso1)}$ and PRC2–AEBP2$^{L(iso2)}$ complexes did not substantially bind the DNA probe, which indicates a much lower affinity ($K_d > 4000$ nM; Fig. 1D). To more comprehensively determine the apparent differences in the DNA binding activities of the complexes, we also performed similar assays using fluorescence anisotropy using the same DNA probe. This corroborated the EMSA findings, with PRC2–AEBP2$^S$ having significantly higher DNA-binding affinity than core PRC2 alone and PRC2–AEBP2$^L$ (Fig. EV1C). The dissociation constants of PRC2–AEBP2$^S$ were largely sensitive to the salt concentration (Fig. EV1C), indicating a strong electrostatic effect that contributed to DNA-binding.

Next, we aimed to investigate the effect of AEBP2$^L$ on the histone methyltransferase (HMTase) activity of PRC2. For each of the complexes, we measured PRC2 HMTase activity against 3.6 kb of chromatinized DNA derived from the *ATOH1* locus (Fig. 1E). While core PRC2 alone and PRC2–AEBP2$^S$ exhibited comparable HMTase activities, PRC2–AEBP2$^L$ was almost completely inactive (Fig. 1E). Collectively, our results thus far indicate that PRC2-AEBP2$^S$ is active in DNA-binding and in HMTase for a chromatin substrate, while PRC2-AEBP2$^L$ is impaired in these activities. As all the previously identified DNA- and nucleosome-interacting regions in AEBP2 are shared by all its isoforms (Lee et al, 2018; Kasinath et al, 2021) (Fig. 1A), these data allude to an antagonistic role for the N-terminal region that is unique to AEBP2$^L$.

### AEBP2$^L$ impairs PRC2 binding to target genes in mouse embryonic stem cells

Based on the finding that PRC2-AEBP2$^S$ exhibited stronger DNA-binding than PRC2-AEBP2$^L$, we hypothesised that ectopically expressed AEBP2$^S$ would bind to chromatin in cells more strongly than AEBP2$^L$. To test this, we ectopically expressed human AEBP2 isoforms to allow for a more direct comparison with the in vitro experiments in Fig. 1. This comparison is reassured by the fact that the mouse and human proteins are highly conserved (Fig. EV2A). We generated complete *Aebp2* KO mouse embryonic stem cells (ESCs) by targeting the first common exon with CRISPR-Cas9, thereby inducing a frameshift mutation leading to early termination

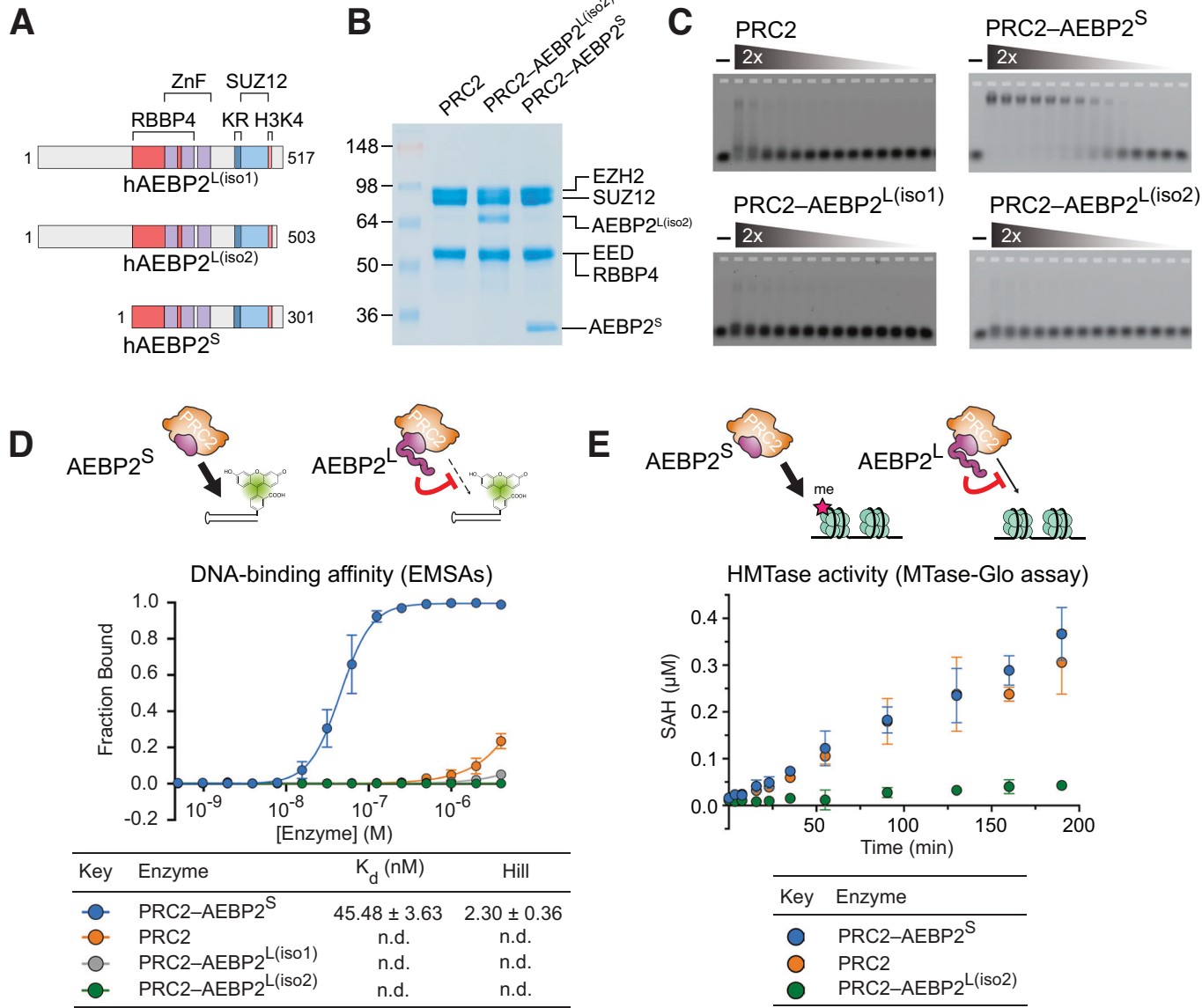

**Figure 1. AEBP2^L inhibits the DNA-binding and HMTase activities of PRC2 in vitro.**

(A) Schematic representation of the domain architecture of human AEBP2^S and AEBP2^L (Isoforms 1 and 2). Zinc fingers (ZnF) and the KR-motif that were implicated in DNA and nucleosome binding are indicated (Lee et al, 2018; Kasinath et al, 2021). RBBP4- and SUZ12-interacting regions are indicated. (B) Coomassie blue-stained 10% acrylamide SDS–PAGE of PRC2 complexes that were used for binding assays. PRC2–AEBP2^L(iso1) is included separately in Fig. EV7. (C) EMSA was performed using a twofold dilution of proteins as indicated, starting from 4000 nM protein concentration and using 5 nM of 24 bp CpG DNA probe. Representative replicate is presented, while data from all replicates are provided in Fig. EV1B. See Fig. EV1C for fluorescence anisotropy binding assays carried out using the same proteins and DNA probe. (D) EMSA used to quantify the affinity of PRC2 complexes to fluorescein-labelled DNA (bottom), accompanied by a model figure describing the proposed mechanism (top). The data represent the mean of three independent experiments carried out on different days. Error bars represent standard deviation. Dissociation constants (K_d) and Hill coefficients are indicated in the table, including their standard error. (E) MTase assays were performed to measure the level of HMTase activity towards a chromatinised 3.6kbp DNA of sequence from the *ATOH1* locus. The enzyme concentration was 50 nM and the chromatin concentration was 1000 nM nucleosome core particle equivalent. Data represent the mean of three independent experiments carried out on different days and error bars represent standard deviation. Source data are available online for this figure.

of all transcripts (Fig. EV2B). This ESC line is hereafter referred to as *Aebp2* KO. We then ectopically expressed either human AEBP2^L(iso1) or AEBP2^S in *Aebp2* KO ESCs (Fig. 2A) and confirmed their comparable protein levels by Western blot, albeit above the endogenous levels (Fig. 2B). These ESC lines are hereafter referred to as *Aebp2* KO + L and *Aebp2* KO + S, respectively. Next, to dissect the potential differences in AEBP2 isoform function on

chromatin, we performed quantitative ChIP-seq with exogenous reference genome spike-in (ChIP-Rx) analysis of AEBP2 using an antibody capable of recognising both isoforms. Average plots of AEBP2 signal on Polycomb target genes, as well as genome browser views of representative examples (*Prmt8*, *Pax7* and the extended *HoxA* gene cluster) showed that both isoforms were capable of binding at Polycomb target genes. However, AEBP2^S exhibited

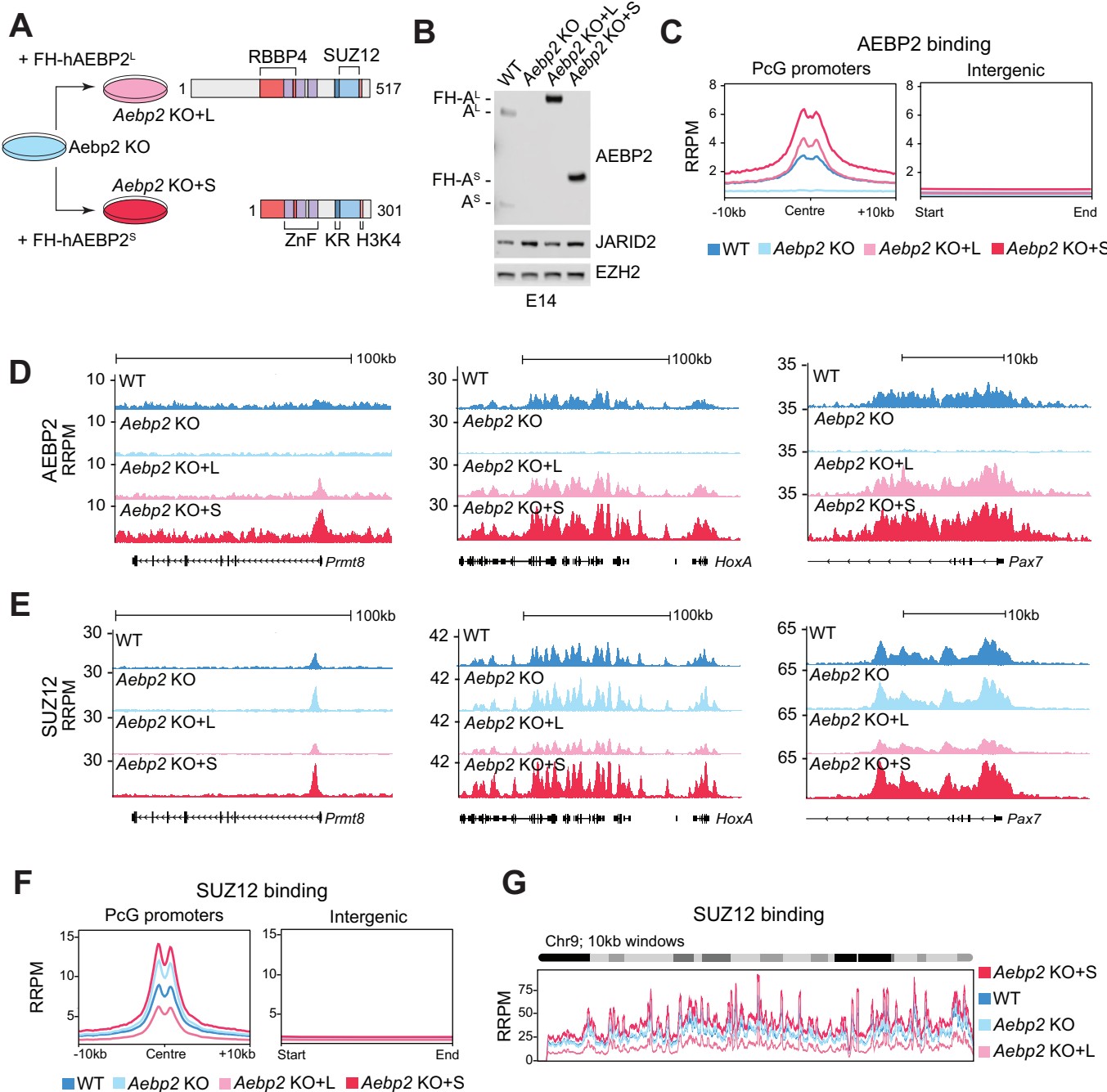

**Figure 2. Ectopic expression of AEBP2^L leads to reduced PRC2 binding on target genes in mouse embryonic stem cells.**

(A) Schematic representation of the strategy of transduction of *Aebp2* KO ESCs with FLAG- and HA-tagged human AEBP2^L/S rescue expression constructs. Zinc fingers (ZnF) and the KR-motif (KR) that were implicated in DNA and nucleosome binding are indicated (Lee et al, 2018; Kasinath et al, 2021). RBBP4- and SUZ12-interacting regions are indicated. (B) Western blot analyses of *Aebp2* KO mouse embryonic stem cells and ESCs stably overexpressing the AEBP2^L/S constructs using the indicated antibodies. (C) Average ChIP-Rx signal profiles of AEBP2 at PcG promoters (n = 2457) and intergenic regions (n = 133,903) in WT, *Aebp2* KO, *Aebp2* KO + L and *Aebp2* KO + S cell lines. RRPM denotes Rx-normalised reads per million. (D) UCSC genome browser representations of Rx normalised reads per million (RRPM) for AEBP2 antibody at three representative PcG target genes (*Prmt8*, *HoxA* and *Pax7*) in WT, *Aebp2* KO, *Aebp2* KO + L and *Aebp2* KO + S cell lines. (E) UCSC genome browser representations of Rx normalised reads per million (RRPM) for SUZ12 antibody at three representative PcG target genes (*Prmt8*, *HoxA* and *Pax7*) in WT, *Aebp2* KO, *Aebp2* KO + L and *Aebp2* KO + S cell lines. (F) Average ChIP-Rx signal profiles of SUZ12 at PcG promoters (n = 2457) and intergenic regions (n = 133,903) in WT, *Aebp2* KO, *Aebp2* KO + L and *Aebp2* KO + S cell lines. (G) Upper: Ideogram of mouse chromosome 9. Lower: SUZ12 RRPM signal within 100 kb bins along chromosome 9 in WT, *Aebp2* KO, *Aebp2* KO + L and *Aebp2* KO + S cell lines. Source data are available online for this figure.

                                                                                                                                                          

stronger binding compared to AEBP2$^L$ (Fig. 2C,D). We further validated these results by ChIP-qPCR (Fig. EV2C). Taken together, these data show that ectopic AEBP2$^S$ is capable of binding to Polycomb target genes more strongly than AEBP2$^L$ in ESCs.

We next sought to determine the effect of AEBP2 isoforms on core PRC2 chromatin occupancy. To evaluate the genome-wide binding profiles of PRC2 in the presence of each isoform, we performed ChIP-Rx of SUZ12 in the *Aebp2* KO and rescue ESC lines. This showed that PRC2 binding was differentially affected on Polycomb target genes depending on which AEBP2 isoform was expressed (Fig. 2E,F). We validated these results by ChIP-qPCR at several Polycomb target genes (Fig. EV2D). To further delineate the genome-wide binding profiles of SUZ12, we partitioned the mouse genome into 10 kb and 100 kb bins and quantified the relative abundance of SUZ12 ChIP-Rx reads per bin in each cell line (Figs. 2G and EV2E). A visualisation of SUZ12 read counts across chromosome 9 showed greater enrichment for SUZ12 in the *Aebp2* KO + S context over *Aebp2* KO, while SUZ12 was less enriched in *Aebp2* KO + L compared to *Aebp2* KO (Fig. 2G). This data was also corroborated genome-wide, beyond chromosome 9 (Fig. EV2E). Taken together, these data suggest that AEBP2$^S$ and AEBP2$^L$ perform opposing functions in regulating PRC2 localisation on chromatin: ectopic expression of AEBP2$^S$ increases the chromatin occupancy of PRC2, whereas AEBP2$^L$ reduces it. However, while this assay is valuable for investigating the relative chromatin binding abilities of AEBP2 isoforms in cells, the strikingly opposite effects we see are likely exaggerated due to their overexpression. Therefore, to explore the effects on H3K27me3 deposition or transcription, we instead focused on developing a system to specifically knock out each isoform and study their individual functions (Figs. 3 and 4).

## AEBP2$^S$ but not AEBP2$^L$ is essential for the repression of de novo Polycomb target genes

The expression of AEBP2 isoforms has been reported to be developmentally regulated (Kim et al, 2009, 2015). To investigate this in more detail, we examined the expression of transcripts coding for the human and mouse AEBP2$^L$ and AEBP2$^S$ in different cell types and tissues from CAGE-seq data (Fig. 3A,B) (Kawaji et al, 2017). In mouse, AEBP2$^S$ is expressed in ESCs, placenta and trophoblasts, but is otherwise largely absent in most tissues (Fig. 3A). In human cells, AEBP2$^S$ is expressed mainly in the testis and is otherwise expressed at very low levels (Fig. 3B). In contrast, AEBP2$^L$ is expressed at substantial levels in most tissues in both mouse and human (Fig. 3A,B). To gain insights on AEBP2 isoforms at the protein level, we performed immunoblotting on protein extracts from mouse embryos and soft tissues from adult mice (Fig. EV3A,B). Strikingly, AEBP2$^L$ is the predominant isoform detected at the protein level as early as embryonic day 9.5 (E9.5), while AEBP2$^S$ is already undetectable at that stage (Fig. EV3B).

To expand on the findings of the rescue experiments in Fig. 2 in a more physiologically relevant context, we employed CRISPR-Cas9 genome editing to generate mouse ESC lines lacking either the endogenous short or long *Aebp2* isoform (Fig. 3C). According to the GENCODE VM23 mouse assembly, there are two long isoforms and three short isoforms of *Aebp2* (Fig. 3D). The long isoforms all share the same promoter, whereas at least three promoters are responsible for the transcription of *Aebp2$^S$* mRNA (denoted as P1,

P2 and P3), all located near the *Aebp2$^L$* promoter. A deletion encompassing all *Aebp2$^S$* promoters was thus impossible, as this would also ablate *Aebp2$^L$*. Therefore, we needed to determine which of the P1, P2 and P3 promoters are active in ESCs. We analysed the enrichment of the H3K4me3 histone modification, known to be associated with promoters, as well as RNA-Seq signal, at the *Aebp2* locus in wild-type ESCs. This revealed that they overlapped the most at promoter P1, suggesting that it is responsible for driving most of the expression of AEBP2$^S$ in mouse ESCs (Fig. 3E). Using this knowledge, we generated an isogenic panel of ESCs in which isoform-specific promoters were knocked out using CRISPR (Fig. 3F). These cell lines are hereafter referred to as WT, LKO, SKO and *Aebp2* KO. We confirmed the complete knockout of AEBP2$^L$, while AEBP2$^S$ was reduced by ~89%, with the remaining AEBP2$^S$ likely originating from the P2 or P3 promoters (Figs. 3G and EV4A). In addition, knockout of either isoform did not considerably affect the protein levels of the other isoform (Figs. 3G and Fig. EV4A). Furthermore, the loss of AEBP2$^S$ or AEBP2$^L$ did not affect the global levels of other PRC2 proteins, except for a small increase in MTF2 abundance in the SKO and *Aebp2* KO cell line (Fig. 3G). A knockout of MTF2 in ESCs has been reported to increase the levels of AEBP2$^S$, with no change to the AEBP2$^L$ isoform (Højfeldt et al, 2019), suggesting an antagonistic relationship between AEBP2$^S$ and MTF2, likely for binding SUZ12 (Chen et al, 2018a). We did not observe global changes in H3K27me2/3 upon the deletion of either or both *Aebp2* isoforms (Fig. EV4B). This is supported by previous findings, wherein knock-out of six PRC2 accessory proteins in ESCs, including both isoforms of AEBP2, did not lead to changes in global H3K27me2/3 levels (Højfeldt et al, 2019). To test the ability of endogenous AEBP2 isoforms to bind chromatin, we performed cellular fractionations of our isogenic KO ESC lines (Fig. EV4C). Consistent with the reduced chromatin binding ability of AEBP2$^L$ demonstrated in Figs. 1 and 2, the relative levels of AEBP2$^L$ were higher in the nuclear soluble fraction compared to the chromatin-bound fraction. In contrast, the majority of AEBP2$^S$ was bound to chromatin, consistent with its greater chromatin binding ability.

To investigate the composition of PRC2 in our *Aebp2* knock-out cell lines, we performed SUZ12 and JARID2 immunoprecipitations (IP) coupled with mass spectrometry (IP-MS; Fig. 3H). The IP-MS revealed that the PRC2 core and PRC2.1 accessory proteins did not differentially bind to SUZ12 upon loss of AEBP2 isoforms (Fig. 3I). Furthermore, it revealed that both isoforms of AEBP2 interact with SUZ12 and can therefore assemble into PRC2. Interestingly, JARID2 was still immunoprecipitated with SUZ12 even in the complete absence of AEBP2 (Fig. 3I). However, the interaction between JARID2 and SUZ12 in the *Aebp2* KO cell line was weaker than in WT ESCs, indicating that the binding of JARID2 to PRC2 is optimal in the presence of either of the AEBP2 isoforms (Fig. EV4D). These findings were recapitulated in JARID2 IP-MS, where the binding of JARID2 to core PRC2 proteins was reduced upon loss of both AEBP2 isoforms. Importantly, these data also confirm that both AEBP2$^S$ and AEBP2$^L$ can form a complex with JARID2 (Figs. 3J and EV4E). Interestingly, we also observed an increased binding of MTF2 to JARID2, either in a complete absence of AEBP2, as reported previously by Grijzenhout et al (Grijzenhout et al, 2016), or in the absence of only AEBP2$^S$ (Fig. EV4E).

Given that both AEBP2 isoforms are co-expressed in mouse ESCs and during early development, we decided to explore their relative

                                                                      

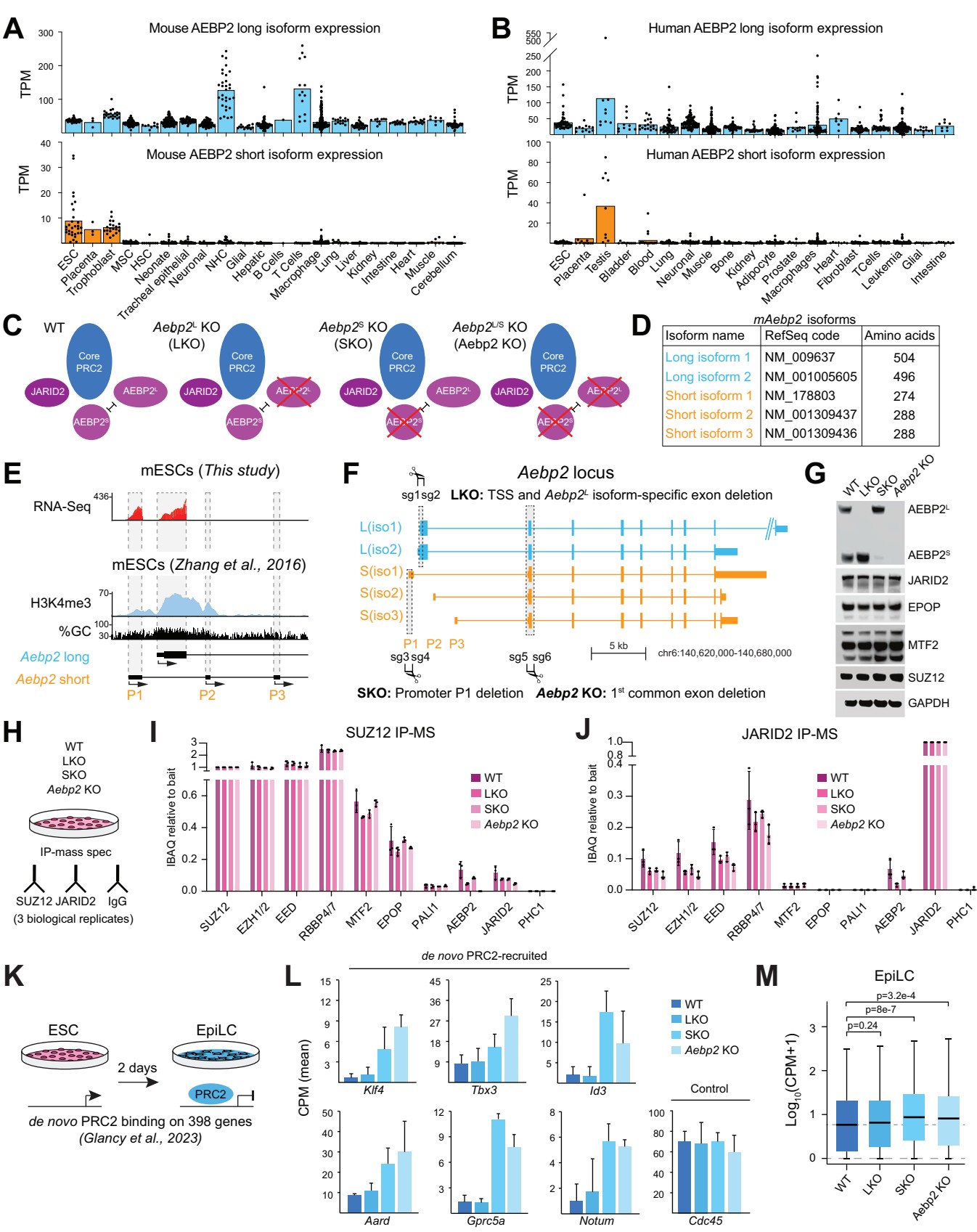

◄  **Figure 3.  AEBP2$^S$ but not AEBP2$^L$ is required for repression of de novo Polycomb target genes during EpiLC differentiation.**

(A) Bar charts representing the mean expression of transcripts coding for AEBP2$^L$ (top) and AEBP2$^S$ (bottom) isoforms, as measured by CAGE-seq, in representative mouse tissues and cell lines. ESC: embryonic stem cells; MSC: mesenchymal stem cells; HSC: hematopoietic stem cells; NHC: natural helper cells; TPM: transcripts per million. (B) Bar charts representing the mean expression of transcripts coding for AEBP2$^L$ (top) and AEBP2$^S$ (bottom) isoforms, as measured by CAGE-seq, in representative human tissues and cell lines. ESC: embryonic stem cells; MSC: mesenchymal stem cells; HSC: hematopoietic stem cells; NHC: natural helper cells; TPM: transcripts per million. (C) Schematic representations of PRC2.2 compositions in the four isogenic mouse ESC lines generated in this study: Wild-type E14 (WT), *Aebp2$^L$* KO (LKO), *Aebp2$^S$* KO (SKO), and *Aebp2$^{L/S}$* KO (*Aebp2* KO). (D) Table of annotated mouse *Aebp2* isoforms, their respective RefSeq codes and number of amino acids. (E) UCSC genome browser representation of H3K4me3 ChIP-Seq and RNA-Seq reads as well as %GC content at the *Aebp2* locus in wild-type ESCs. Positions of the *Aebp2$^L$* TSS and the three *Aebp2$^S$* promoters are indicated and overlaid on top of the tracks in orange. ChIP-Seq data downloaded from Zhang et al, 2016 (Zhang et al, 2016). (F) Schematic representation of the *Aebp2* locus and the CRISPR-Cas9 strategy taken to knock out *Aebp2$^S$* and *Aebp2$^L$*. Brackets represent locations of the sgRNAs used to excise each region. sgRNA sequences are available in Dataset EV1. (G) Western blot analyses using the indicated antibodies on whole-cell lysates from the chosen clones of WT, LKO, SKO and *Aebp2* KO cell lines. (H) Diagram illustrating the experiment set-up for immunoprecipitation coupled with mass spectrometry (IP-mass spec) on the four indicated ESC lines, using SUZ12, JARID2 and IgG antibodies. Three IPs were performed for each antibody on independent biological replicates. (I) Bar graphs showing the abundance of peptides mapping to PRC2 proteins (iBAQ) relative to bait (SUZ12), in the WT, LKO, SKO and *Aebp2* KO cell lines. PRC1 component PHC1 is included as a negative control. The error bars represent the standard deviation based on three independent experiments. (J) Bar graphs showing the abundance of peptides mapping to PRC2 proteins (iBAQ) relative to bait (JARID2), in the WT, LKO, SKO and *Aebp2* KO cell lines. PRC1 component PHC1 is included as a negative control. The error bars represent the standard deviation based on three independent experiments. (K) Diagram illustrating the experiment set up for differentiation of naïve ESCs into primed epiblast-like cells (EpiLC), and the expression and PRC2 occupancy dynamics on genes included in the downstream analysis. Data from Glancy et al, 2023 (Glancy et al, 2023). (L) Bar graphs showing mRNA levels of representative de novo PRC2-recruited genes and a control (*Cdc45*) in EpiLCs derived from *Aebp2* KO, LKO, SKO and matched WT ESCs, as measured by Quant-Seq. Error bars represent standard deviation based on three independent replicates. (M) Boxplots representing mRNA levels of the 398 PRC2-recruited genes in EpiLCs measured by Quant-Seq. Shown is the mean of three independent biological replicates and the indicated *P* values were determined by the Wilcoxon rank-sum test. The boxplots represent the interquartile range (Q1 to Q3), with a median indicated by the thick line, and the minimum and maximum values indicated by the whiskers. Source data are available online for this figure.

contributions to Polycomb-mediated gene repression in a model of directed ESC differentiation. We induced naïve ESCs to undergo differentiation towards primed, post-implantation pre-gastrulation epiblast-like cells (EpiLCs) and performed Quant-Seq (Fig. 3K). We validated the differentiation by analysing the mRNA levels (mean CPM) of genes linked to naïve (*Nanog* and *Prdm14*) and primed (*Fgf5* and *Dnmt3b*) pluripotency, demonstrating their respective down- and up-regulation (Fig. EV5A). In addition, we confirmed that both AEBP2 and JARID2 proteins are maintained in WT EpiLC cells (Fig. EV5B). Next, we explored the gene expression changes resulting from the loss of specific AEBP2 isoforms and found that the SKO and *Aebp2* KO EpiLCs exhibited a greater number of differentially expressed genes compared to WT and LKO EpiLCs (Fig. EV5C; Dataset EV4). We compared the expression of 149 genes previously identified as most differentially expressed during ESC to EpiLC differentiation (Hayashi et al, 2011). This revealed a trend—though not statistically significant—of similar expression patterns between WT and LKO, and between SKO and Aebp2 KO (Fig. EV5D,E). Importantly, to assess the direct consequences of AEBP2 isoform loss, we focused on a set of 398 genes that we recently identified to be repressed during ESC to EpiLC differentiation, with de novo PRC2 recruitment and partial dependence on the Polycomb-like and JARID2 accessory proteins (Glancy et al, 2023). While expression of these genes remained high and unchanged in undifferentiated ESCs (Fig. EV5F), they were less repressed during EpiLC differentiation in SKO and *Aebp2* KO, but not LKO, compared to WT (Fig. 3L,M). Taken together, these data suggest that AEBP2$^S$ is the main AEBP2 isoform contributing, like the Polycomb-like and JARID2 accessory proteins, to de novo repression of Polycomb target genes during ESC to EpiLC differentiation.

## AEBP2$^L$ antagonises PRC2 in mouse embryonic stem cells

Next, we evaluated the genome-wide localisation of AEBP2, SUZ12 and JARID2 by ChIP-seq/Rx and the deposition profiles of H3K27me3 by CUT&RUN in our *Aebp2* knock-out ESC lines. The SKO ESCs displayed lower AEBP2 occupancy at Polycomb

target genes, compared to LKO, while the full *Aebp2* KO had signal reduced to background levels (Figs. 4A–C and EV6A). Previous studies reported increased enrichments of SUZ12 and H3K27me3 at Polycomb promoters in ESCs lacking both isoforms of AEBP2 (Grijzenhout et al, 2016; Conway et al, 2018). We corroborated and expanded these results, finding that the specific loss of AEBP2$^L$ led to more significant increases in SUZ12 and H3K27me3 at Polycomb target genes, compared to loss of AEBP2$^S$ alone (Figs. 4A–C and EV6A–C). This again supports our proposal that AEBP2$^L$ is a negative regulator of PRC2. Interestingly, the loss of AEBP2$^L$ (in LKO) or the loss of both AEBP2 isoforms (*Aebp2* KO) also caused increases of JARID2 at Polycomb target genes (Fig. 4A–C). Thus, JARID2 can not only integrate into the PRC2 complex without AEBP2 isoforms but can also target chromatin independently of AEBP2.

Taken together, these analyses indicate that the long and short isoforms of AEBP2 have different chromatin binding capabilities. Collectively, this data supports a model in which AEBP2$^L$, but not AEBP2$^S$, functions to antagonise the binding of PRC2 to Polycomb target genes in embryonic stem cells, thereby restricting the deposition of H3K27me3.

## The N-terminal region of AEBP2$^L$ is disordered and utilises mammalian-specific acidic tracts to inhibit PRC2

Next, we wanted to determine which part of AEBP2$^L$ is responsible for its antagonistic role in regulating PRC2 function. The entire C-terminus of AEBP2$^L$ (amino acids 202–517) is encoded by exons shared with AEBP2$^S$. Thus, we hypothesised that the unique N-terminus of AEBP2$^L$ (amino acids 1–201) would likely confer its inhibitory activity. The N-terminus displays a distinctively strong negative charge (Fig. 5A, in red), in contrast to the positively charged DNA- and nucleosome-binding regions closer to the C-terminus (Fig. 5A, in blue). We identified two sequences of 10–15 amino acids each within the N-terminal region of AEBP2$^L$, composed entirely of glutamic (Glu, E) and aspartic acids (Asp, D).

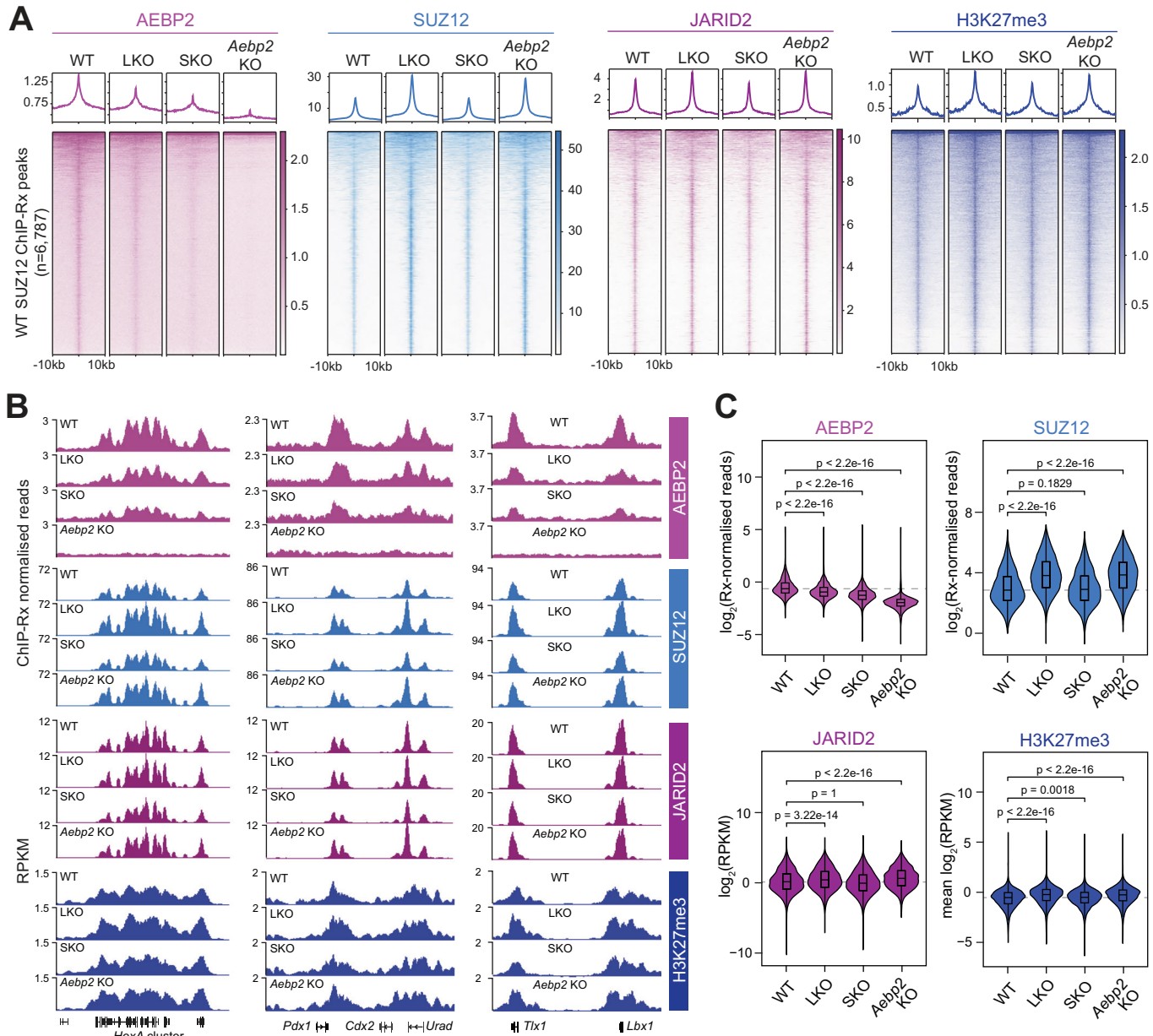

**Figure 4. Loss of AEBP2$^L$ but not AEBP2$^S$ leads to increased PRC2 and H3K27me3 on Polycomb target genes in mouse ESCs.**

(A) Average plot and heatmap representations of ChIP-Rx-normalised reads for AEBP2, SUZ12 and RPKM values for JARID2 at SUZ12-bound peaks in wild-type ESCs ($n = 6787$) in WT, LKO, SKO and *Aebp2* KO. CUT&RUN of H3K27me3 is also represented at these sites. Plots are centred on the region midpoint $+/-$10kb. Relative intensities are indicated for each antibody. (B) UCSC genome browser representations of ChIP-Rx normalised reads per million (RRPM) for AEBP2 and SUZ12 antibodies, and RPKM for JARID2 antibody at three representative Polycomb target loci (*HoxA*, *Pdx1/Cdx2*, and *Tlx1/Lbx1*) in WT, LKO, SKO and *Aebp2* KO ESC lines. UCSC genome browser representations of the CUT&RUN normalised reads (RPKM) for the H3K27me3 antibody at the same loci. (C) Violin plots showing the log$_2$ abundance of normalised reads for AEBP2, SUZ12, JARID2 and H3K27me3 at WT SUZ12-bound promoters ($n = 3650$). H3K27me3 violin plots represent log$_2$ mean RPKM values of three independent biological CUT&RUN replicates. The indicated *P* values were calculated using the Wilcoxon rank-sum test with continuity correction (right-sided for JARID2, SUZ12 and H3K27me3, and left-sided for AEBP2). The inside boxplots represent the interquartile range (Q1 to Q3), with a median indicated by the thick line, and the minimum and maximum values indicated by the whiskers.

We refer here to these regions as acidic tracts (Fig. 5A,B). Sequence conservation analysis revealed that the length and sequence of these N-terminal acidic tracts are highly varied across vertebrate species but conserved across placental mammals (Fig. 5B). These observations led us to hypothesise that the negatively charged acidic tracts of AEBP2$^L$ evolved recently to interfere with DNA binding and,

therefore, methyltransferase activity of PRC2, possibly via electrostatic repulsions.

An in silico analysis of the AEBP2$^L$ protein sequence predicted that the entire N-terminus region is highly disordered, in contrast with the structured zinc finger domains and SUZ12 binding helix in the C-terminus (Fig. 5A). However, no structures of PRC2

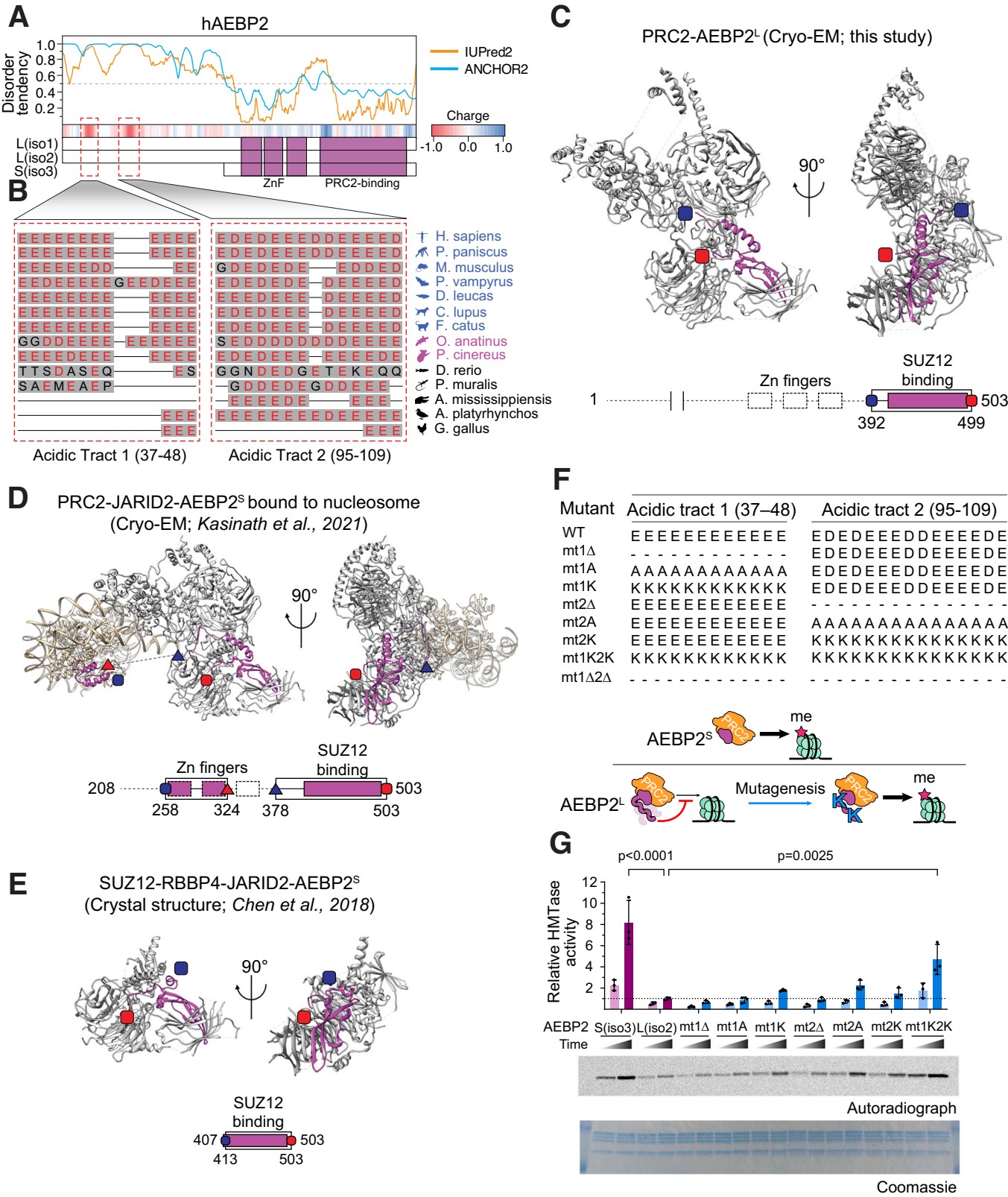

**A** hAEBP2

**B** Acidic Tract 1 (37–48)  Acidic Tract 2 (95–109)

**C** PRC2-AEBP2$^L$ (Cryo-EM; this study)

**D** PRC2-JARID2-AEBP2$^S$ bound to nucleosome (Cryo-EM; *Kasinath et al., 2021*)

**E** SUZ12-RBBP4-JARID2-AEBP2$^S$ (Crystal structure; *Chen et al., 2018*)

**F**

| Mutant | Acidic tract 1 (37–48) | Acidic tract 2 (95–109) |
|---|---|---|
| WT | E E E E E E E E E E E | E D E D E E D D E E E D |
| mt1Δ | - - - - - - - - - - - | E D E D E E D D E E E D |
| mt1A | A A A A A A A A A A A | E D E D E E D D E E E D |
| mt1K | K K K K K K K K K K K | E D E D E E D D E E E D |
| mt2Δ | E E E E E E E E E E E | - - - - - - - - - - - - |
| mt2A | E E E E E E E E E E E | A A A A A A A A A A A A |
| mt2K | E E E E E E E E E E E | K K K K K K K K K K K K |
| mt1K2K | K K K K K K K K K K K | K K K K K K K K K K K K |
| mt1Δ2Δ | - - - - - - - - - - - | - - - - - - - - - - - - |

**G**

Figure 5. The N-terminal region of AEBP2$^L$ is disordered and utilises mammalian-specific acidic tracts to inhibit PRC2.

(A) Predicted disordered regions of AEBP2$^L$ represented using two different disorder prediction algorithms: IUPred2A and Anchor2 (Mészáros et al, 2018; Erdős and Dosztányi, 2020). Negative and positively charged residues of AEBP2$^L$ are coloured underneath the disorder chart in red and blue, respectively. Aligned with the sequence are AEBP2$^{L(iso1)}$, AEBP2$^{L(iso2)}$ and AEBP2$^S$ with the zinc fingers and SUZ12 binding domains in purple. (B) Multiple sequence alignments showing the evolutionary conservation of two domains within the isoform-specific N-terminus of AEBP2$^L$. Placental mammals are shown in blue, non-placental mammals in purple and non-mammals in black. Red text indicates negatively charged amino acid residues. (C) Cryo-EM structure of PRC2 in a complex with AEBP2$^{L(iso1)}$ at 3.6 Å resolution. Structured regions of AEBP2 are marked in magenta with their N-terminal and C-terminal amino acids marked in blue and red, respectively. PRC2 is marked in grey. Below, the regions of AEBP2$^{L(iso1)}$ that were resolved using Cryo-EM are boxed, while the dashed line represents regions that were present in the construct but were not resolved. (D) Previously determined Cryo-EM structure of PRC2–JARID2-AEBP2$^{S(iso3)}$ (PDB ID 6WKR (Kasinath et al, 2021)) presented using identical colour code, with a nucleosomal construct in gold. (E) Crystal structure of SUZ12[76–545]-RBBP4-JARID2[147–165] in complex with the C-terminal fragment of AEBP2$^{L(iso2)}$[407-503] (PDB ID 5WAI) (Chen et al, 2018b) presented using identical colour code. (F) Sequences of the generated mutants of AEBP2$^L$ N-terminal acidic tracts. Model figure represents the tested hypothesis, that the presence of the wild-type AEBP2$^L$ N-terminal acidic domains inhibits HMTase activity. When the acidic tracts are mutated to lysine residues, HMTase activity if partially restored. (G) HMTase assays of different PRC2–AEBP2 complexes, including wild-type and mutant AEBP2, with a polynucleosomal substrate. Methylation levels of H3 were determined by 14C-autoradiography (upper gel), and histone proteins were visualised using Coomassie stain (lower gel). Bar plots represent the mean of quantification using densitometry, and error bars represent standard deviation based on three independent experiments. See Fig. EV7 for SDS–PAGE and gel filtration chromatography of holo-PRC2 complexes. The p-values were calculated using the ANOVA test and Tukey's multiple comparisons test using Prism. Source data are available online for this figure.

containing AEBP2$^L$ are currently available, as all previous studies used either AEBP2$^S$ or its partial sequence, without its N-terminus (Ciferri et al, 2012; Chen et al, 2018b; Kasinath et al, 2018, 2021). Therefore, we solved the high-resolution cryo-electron microscopy (Cryo-EM) structure of PRC2 in a complex with the full-length AEBP2$^L$ (Figs. 5C and EV7A; Table 1). We were able to solve the structure at a resolution of 3.6 Å, which to our knowledge is the highest resolution of a PRC2–AEBP2 obtained to date. Yet, we were unable to identify electron density for the N-terminal region of AEBP2$^L$, consistent with its predicted disorder propensity (Fig. 5A,C). Importantly, we observed no structural changes to the catalytic lobe of PRC2 or to the DNA- and SUZ12-binding domains of AEBP2$^L$ compared to the conformation of AEBP2$^S$ within the context of the PRC2–AEBP2$^S$–JARID2 complex (Kasinath et al, 2018) and the partial PRC2-AEBP2 complex that was determined using X-ray crystallography (Chen et al, 2018b) (Fig. 5D,E). These results imply that the N-terminus of AEBP2$^L$ is flexible and highly dynamic and strongly indicate that its presence does not affect the structure of the rest of PRC2. Therefore, given its negative charge, we hypothesised that the N-terminus of AEBP2$^L$ antagonises PRC2 binding to chromatin via repulsive electrostatic interactions with DNA rather than by affecting the conformation of the PRC2 catalytic core.

To test this hypothesis, we purified mutant PRC2–AEBP2$^L$ complexes (Figs. 5F and EV7C). In each of these mutants, all the amino acids in one or both acidic tracts were either deleted, mutated to alanine residues, or subjected to charge swap via mutagenesis to lysine residues (Figs. 5F and EV7C). We next performed histone methyltransferase activity assays to determine how the different AEBP2$^L$ mutants affect the activity of PRC2 (Fig. 5G). To allow for electrostatic interactions or repulsions to take place normally, the assays were performed under a near-physiological salt concentration of 100 mM KCl. Mutating all the amino acids in the second acidic tract into alanine residues (mt2A) increased the HMTase activity of PRC2 by over twofold, consistent with an inhibitory function of the second acidic tract (Fig. 5G). Charge swap mutations in either the first (mt1K) or second (mt2K) acidic tract of AEBP2$^L$ increased the histone methyltransferase activity of PRC2 by ~1.5-fold. The activity of PRC2–AEBP2$^L$ was increased substantially when all the negatively charged amino acids in both the acidic tracts of AEBP2$^L$ were converted into lysines

(mt1K2K). This mutant rescued most of the activity of PRC2, bringing it to over half the activity of PRC2–AEBP2$^S$ and over 4-fold the activity of the wild-type PRC2–AEBP2$^L$ (Fig. 5G).

The deletion of either acidic tract (mt1Δ and mt2Δ) was insufficient to alleviate the inhibitory activity of AEBP2$^L$ (Fig. 5G). This could point to functional redundancy between the two motifs or for the involvement of additional determinants in inhibiting PRC2. Yet, the second acidic tract in AEBP2$^L$ is involved in inhibiting PRC2, given that mt2A increased the activity of PRC2 by >twofold. This result, together with the positive effect seen for the charge-swap mutation mt1K2K, point to electrostatic interactions as a determinant. An electrostatic effect is further supported by the lack of AEBP2$^L$-mediated inhibition of HMTase activity against DNA-free H3 histones (Fig. EV7B). These results fit well with the observations that the negatively charged N-terminal region of AEBP2 impairs the DNA-binding of PRC2 in vitro (Fig. 1) and the chromatin binding activity of PRC2 in cells (Figs. 2 and 4). Collectively, these results point to the N-terminal region of AEBP2$^L$ as a mammalian-specific negative regulator of PRC2.

## AEBP2$^L$ antagonises PRC2.2

Our proteomic analyses indicate that either AEBP2 isoform can occupy the holo-PRC2.2 complex with JARID2 (Figs. 3J and EV4E). We therefore sought to study AEBP2$^L$ and AEBP2$^S$ in the context of a PRC2.2 complex that includes JARID2. We first generated an AEBP2$^L$ construct that included an internal tobacco etch virus (TEV) protease site within the coding sequence, immediately downstream of the N-terminal region of AEBP2$^L$ (Fig. 6A). This protein, termed here AEBP2$^{L(TEV)}$, was co-expressed and co-purified using the baculovirus system together with EZH2, SUZ12, EED, RBBP4 and the full-length JARID2, to form the PRC2-AEBP2$^{L(TEV)}$-JARID2 complex. Upon cleavage at the TEV sequence using a TEV protease, PRC2-AEBP2$^{L(TEV)}$-JARID2 was converted into PRC2-AEBP2$^S$-JARID2 (Fig. 6A,B). This approach was taken out of necessity, as we were unable to produce a stable PRC2-AEBP2$^S$-JARID2 complex by simple means of co-expression (Fig. 6E, right lane). Nevertheless, our TEV approach provided the advantage of a direct comparison between the two AEBP2 isoforms without batch effect complications, such as altered subunit stoichiometry or variations in post-translational modifications.

**Table 1. Cryo-EM data collection, refinement and validation statistics.**

| Data collection parameters | PRC2-AEBP2[L] |
|---|---|
| Magnification | 105,000 |
| Voltage (kV) | 300 |
| Camera | Gatan K3 |
| Electron exposure (e$^-$ Å$^{-2}$) | 60 |
| Defocus range (µm) | −0.5 to −2.5 |
| Pixel size (Å) | 0.86 |
| Symmetry imposed | C1 |
| Map resolution (Å) | 3.64 |
| FSC threshold | 0.143 |
| **Refinement statistics** | |
| Model composition | |
| Non-hydrogen atoms | 13,974 |
| Protein residues | 1765 |
| r.m.s. deviations | |
| Bond lengths (Å) | 0.003 |
| Bond angles (°) | 0.782 |
| **Validation statistics** | |
| Molprobity score | 2.10 |
| Rotamers outliers (%) | 0.13 |
| Clashscore, all atoms | 12.22 |
| Ramachandran plot | |
| Favoured (%) | 91.51 |
| Allowed (%) | 8.26 |
| Outliers (%) | 0.23 |

We then treated the PRC2-AEBP2$^{L(TEV)}$-JARID2 complex with or without TEV and subjected it to HMTase assays in vitro, using a reconstituted chromatin substrate (Fig. 6C). In agreement with our analysis of PRC2-AEBP2 (Fig. 1), cleaving the N-terminal region of AEBP2$^L$ substantially increased the HMTase activity of PRC2.2 (Fig. 6C). This data indicates that the N-terminal region of AEBP2$^L$ inhibits the activity of PRC2-AEBP2$^{L(TEV)}$-JARID2 on a chromatinized substrate. To directly investigate if the N-terminal region of AEBP2$^L$ interferes with chromatin binding, we performed EMSA for PRC2-AEBP2$^{L(TEV)}$-JARID2 using a fluorescently labelled mononucleosome probe of 182 bp DNA (NCP$_{182}$; Fig. EV8A,B). EMSA of PRC2-AEBP2$^{L(TEV)}$-JARID2 were carried out with or without TEV cleavage, in a TEV concentration that does not allow the protease to bind to the nucleosome (Fig. EV8C). TEV cleavage of PRC2-AEBP2$^{L(TEV)}$-JARID2 only slightly increased the apparent affinity of PRC2.2 for nucleosomes (Fig. EV8B). It is possible that the increment in affinity is underestimated, as the affinity of PRC2-AEBP2$^{L(TEV)}$-JARID2 to NCP$_{182}$ (apparent $K_d = 24$ nM) is already close to the probe concentration (5 nM). Nevertheless, TEV cleavage led to the formation of two bands (red asterisks in Fig. EV8A), implying a different binding mode. Importantly, if AEBP2$^L$ inhibits PRC2 through interference with chromatin binding, it would be expected that the catalytic centre of PRC2 is not directly inhibited. Hence, to formally exclude the possibility of direct inhibition of catalysis, we treated PRC2-AEBP2$^{L(TEV)}$-JARID2

in the presence and absence of TEV and performed a HMTase assay using an H3 histone tail peptide substrate (Fig. 6D). As expected, PRC2-AEBP2$^{L(TEV)}$-JARID2 exhibited the same HMTase activity irrespective of whether the N-terminal region was cleaved or not. This data indicates that the catalytic site of PRC2 is not inhibited and fits with the inhibition of the chromatin binding activity of PRC2, mediated by the N-terminal region of AEBP2$^L$.

To determine the role of the acidic tracts in AEBP2, we produced recombinant PRC2-AEBP2$^L$-JARID2 wild-type and the acidic tracts mutant complexes—mt1K2K, mt1A2A and mt1Δ2Δ (see Fig. 5F for the sequences and Fig. 6E for the proteins). The wild-type and mutants were then subjected to a HMTase assay with a Michaelis–Menten kinetic analysis using reconstituted chromatin as a substrate (Figs. 6F,I and EV8D). All the mutants increased the catalytic efficiency of PRC2 by 2.2- to 2.6-fold compared to the wild-type, in agreement with an inhibitory role of the acidic tracts of AEBP2$^L$ (Fig. 6I). Accordingly, both the PRC2-AEBP2$^L$-JARID2 mt1A2A and mt1Δ2Δ mutants showed an increased production of H3K27me3 modification on chromatin in vitro (Figs. 6G and EV8E). Fluorescence anisotropy binding assays indicated that either deletion of the two acidic tracks in AEBP2$^L$ (PRC2-AEBP2$^{Lmt1Δ2Δ}$-JARID2) or mutating them to alanine residues (PRC2-AEBP2$^{Lmt1A2A}$-JARID2) or lysines (PRC2-AEBP2L$^{mt1K2K}$-JARID2), increase the affinity of PRC2.2 for a 46 bp CpG DNA probe by >15-fold $K_d$ (Fig. 6J and Fig. EV8H). These observations indicate that the acidic tracts of AEBP2 inhibit the DNA-binding activity of PRC2.2. EMSA using the fluorescently labelled mononucleosome probe NCP$_{182}$ quantified a small but consistent reduction in $K_d$ (up to twofold; Figs. 6H,I and EV8F,G), in agreement with increased affinity to nucleosomes or altered binding mode. Qualitatively, these observations agree with previous studies, where PRC2 mutants defective in the interactions with the substrate nucleosome exhibited a substantial defect in HMTase but only a moderate change in the apparent affinity for nucleosome probes during EMSA (Gail et al, 2024; Finogenova et al, 2020). Contributing factors could be additional unaffected nucleosome-interacting surfaces on PRC2, as discussed previously (Gail et al, 2024; Finogenova et al, 2020). Nevertheless, the data thus far indicate that the acidic tracts of AEBP2$^L$ reduce the HMTase activity of PRC2.2 on chromatin.

If AEBP2$^L$ inhibits PRC2.2 solely by antagonising its ability to bind chromatin in cells, then one would expect that this effect (Figs. 2 and 4) can be nullified by forcing chromatin interaction through ectopic tethering. We therefore expressed TetR fusion proteins of AEBP2$^L$ and AEBP2$^S$ in ESCs that include a chromosomal-integrated GFP reporter gene downstream of a TetO DNA binding array (Fig. EV9A,B), as previously done with other Polycomb group proteins (Moussa et al, 2019). Forced recruitment of both TetR-AEBP2$^L$ and TetR-AEBP2$^S$ resulted in comparable PRC2-dependent GFP repression in over 90% of the cells (Fig. EV9C–F). While we were unable to obtain a sufficient expression level of the TetR-AEBP2$^L$ mt1K2K mutant (Fig. EV9B), the TetR-AEBP2$^L$ mt1Δ2Δ mutant was stably expressed (Fig. EV9B) and exhibited similar reporter gene repression as the wild-type AEBP2 proteins (Fig. EV9C). Importantly, ChIP-qPCR showed that repression correlated with similar SUZ12 and JARID2 binding and deposition of H3K27me3 on the TetO array in cells expressing either TetR-AEBP2$^L$, TetR-AEBP2$^S$ or TetR-AEBP2$^L$ mt1Δ2Δ (Fig. EV9G). As expected, outside the tethered site, TetR-AEBP2$^S$

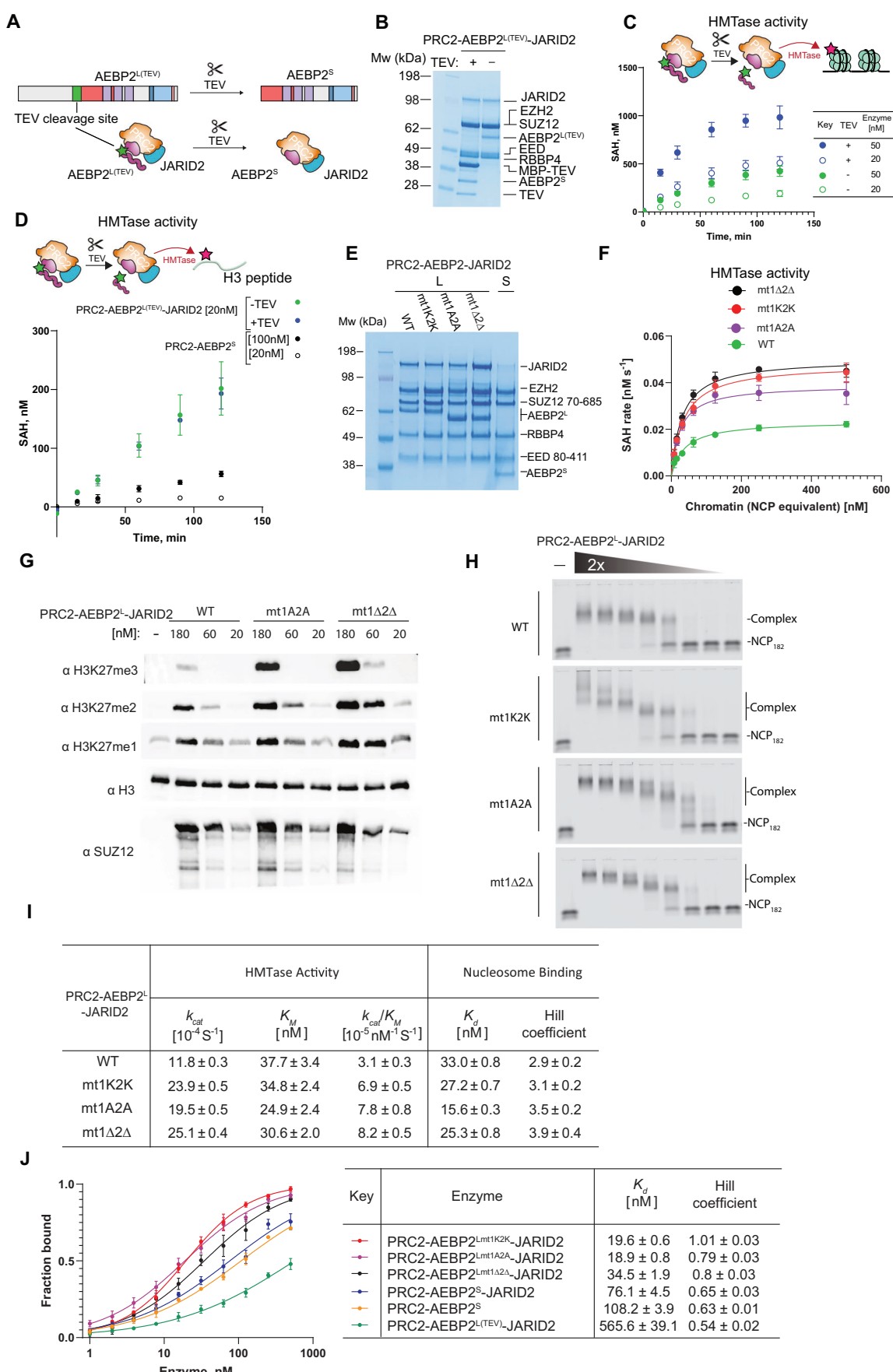

◄

**Figure 6. AEBP2$^L$ antagonises PRC2.2.**

(A) A schematic representation of AEBP2$^{L(TEV)}$ construct and its product after cleavage using TEV protease (marked with scissors). The TEV cleavage site is shown as a green box and a green star. (B) Coomassie-stained SDS–PAGE of PRC2-AEBP2$^{L(TEV)}$-JARID2 in the presence or absence of TEV protease. Protein subunits are annotated. AEBP2$^S$ and AEBP2$^{L(N-terminal)}$ are the two cleavage products of AEBP2$^{L(TEV)}$: the short isoform of AEBP2 and the N-terminal region of the long isoform, respectively. (C) HMTase activity assay of the PRC2-AEBP2$^{L(TEV)}$-JARID2 enzyme with or without TEV protease and with 500 nM chromatin (NCP equivalent), 25 µM SAM and enzyme concentrations, as indicated. Reaction was stopped at 15, 30, 60, 90 and 120 min. Means represent the concentration of SAH that was produced at different time points and the error bars represent a standard error over three independent replicates that were carried out on 3 different days. (D) HMTase activity assays of the PRC2-AEBP2$^{L(TEV)}$-JARID2 enzyme with or without TEV protease were done as in (C), with the exception that 100 µM H3 peptide was used as the substrate. PRC2-AEBP2$^S$ was included as a control (indicated in black circles) in concentrations as indicated. Means and error bars represent the average SAH concentration and the standard error, respectively, based on three independent replicates performed on 3 different days. (E) Coomassie-stained SDS–PAGE of purified PRC2-AEBP2$^S$-JARID2 and PRC2-AEBP2$^L$-JARID2 wild-type and mutants, as indicated. (F) Michaelis–Menten kinetic analysis quantifies the HMTase activity of 20 nM wild-type PRC2-AEBP2$^L$-JARID2 and its mutants in 25 µM SAM for 60 min in the presence of twofold dilutions of chromatin, starting from 500 nM (NCP equivalent). Means represent averages of three independent replicates that were carried out on 3 different days, and the error bar shows the standard deviation. Data was fitted with a nonlinear regression of the Michaelis–Menten model (constant lines) to derive $K_m$ and $k_{cat}$ and their standard error (indicated in (I)). Prior to this experiment, a progress curve was done to confirm that the assay was carried out during the linear range of the reaction (Fig. EV8D). (G) A western blot was carried out after in vitro HMTase assays with antibodies as indicated. The indicated enzymes were assayed in the indicated concentration with 500 nM chromatin (NCP equivalent) and 25 µM SAM. (H) The affinities of wild-type and mutants PRC2-AEBP2$^L$-JARID2 for a Cy5-labelled NCP$_{182}$ nucleosome probe were assayed using EMSA. The probe concentration was 5 nM and the protein concentration was subjected to twofold serial dilution starting from 500 nM. Additional two independent replicates that were carried out on different days are presented in Fig. EV8F, binding curves are presented in Fig. EV8G and the derived dissociation constant ($K_d$) and Hill coefficients are in (I). (I) Kinetic constants derived from the HMTase assays in (E) and equilibrium binding constants derived from the EMSA experiment in (G) and Fig. EV8F. (J) Fluorescence anisotropy binding assays quantified the affinity of the indicated PRC2.2 complexes for a 5 nM CpG46 DNA probe in a binding buffer of 20 mM Tris 8.0, 100 mM NaCl, 0.1 mM ZnCl$_2$, 0.2 mg/ml BSA, 0.002% NP40 and 2 mM 2-mercaptoethanol. Enzyme: assayed complexes, as indicated. $K_d$: dissociation constant. Data represent the mean of three independent experiments that were carried out on different days and the error bars represent standard deviation. Source data are available online for this figure.

showed stronger binding to two Polycomb target genes compared with TetR-AEBP2$^L$, while TetR-AEBP2$^L$ mt1Δ2Δ exhibited intermediate binding. Taken all together, these results further support our model that the N-terminal region of AEBP2$^L$ antagonises PRC2.2 through interference with chromatin binding.

# Discussion

In recent years, we and others have shown that the Polycomb system has greatly expanded and sub-functionalised in vertebrates through gene duplication (Li et al, 2011; Alekseyenko et al, 2014; Liefke and Shi, 2015; Conway et al, 2018; Hübner et al, 2019; Piunti et al, 2019; Ragazzini et al, 2019; Beringer et al, 2016). For example, the Polycomb-like (Pcl) gene underwent successive gene duplication events from one gene in the fly to three in vertebrates (*PHF1*, *MTF2* and *PHF19*) (Owen and Davidovich, 2022; Brien et al, 2015). All Polycomb-like proteins (PCLs) are mutually exclusive with AEBP2 (Hauri et al, 2016; Chen et al, 2018a), and AEBP2 is expressed from a single locus in organisms from fly to human. Our study points to a further expansion in the number of accessory subunits available to regulate PRC2 during cell differentiation through the utilisation of alternative transcription start sites within the *Aebp2* gene locus. We identified a mechanism by which two isoforms of AEBP2 can perform contrasting functions in regulating PRC2 activity. This adds to the growing evidence for divergent functions of Polycomb proteins, such as the recently discovered EZHIP, which is a negative regulator of PRC2 in the specific context of the germline. We propose that the negative regulation of PRC2 by its own AEBP2$^L$ subunit is a prevalent biological phenomenon. AEBP2$^L$ is the predominant isoform in somatic cells from at least mid-gestation (mouse E9.5 embryos; Fig. EV3B) and is by far the most highly expressed AEBP2 isoform in most human and mouse adult tissues (Fig. 3A,B). This is fundamental because many of the biochemical and structural biology studies performed to date were done using the AEBP2$^S$ isoform that has little biological relevance after early embryogenesis. On a mechanistic level, we demonstrated that the N-terminal region of AEBP2$^L$ inhibits the HMTase activity of PRC2 by interfering its chromatin binding activity without directly blocking its catalytic site (Fig. 6). We further identified the AEBP2$^L$-specific, acidic N-terminal tracts as contributing to this inhibitory function.

Seemingly contradictory evidence in several previous studies proposed opposite functions for AEBP2, either as a transcriptional repressor (He et al, 1999; Cao et al, 2002; Kim et al, 2009; Ciferri et al, 2012; Kalb et al, 2014; Lee et al, 2018; Kasinath et al, 2021) or activator (Grijzenhout et al, 2016; Conway et al, 2018; Lin et al, 2021; Leicher et al, 2020). Given the data herein, it is plausible that some of the discrepancies between previous studies were ascribable to the use of only AEBP2$^S$ or AEBP2$^L$, or to variations of their expression levels in different model systems. We show that AEBP2$^S$ acts as a Polycomb protein, being better than AEBP2$^L$ at binding to Polycomb target genes and being capable of promoting the de novo repression of Polycomb target genes during differentiation of ESCs to EpiLCs. In contrast, AEBP2$^L$ has a lower affinity for DNA in vitro (Fig. 1) and is the predominant isoform in the soluble nuclear fraction (Fig. EV4C). Yet, it is still capable of localising to Polycomb target genes in ESCs (Figs. 2C,D and EV2B). However, ectopically expressed AEBP2$^L$ reduced the amount of SUZ12 on chromatin compared to AEBP2$^S$ and even compared to the no-rescue control cells (Fig. 2E–G). Supporting this, the loss of AEBP2$^L$, but not the loss of AEBP2$^S$, led to increased PRC2 binding on Polycomb target genes in ESCs. The increased JARID2 binding on Polycomb target genes in the LKO and full *Aebp2* KO ESCs indicates that AEBP2$^L$ is also capable of restricting the chromatin occupancy of PRC2.2 in cells that express JARID2. This agrees with the antagonistic activity of the N-terminal region of AEBP2$^L$, which inhibits the HMTase activity of JARID2-containing PRC2.2 selectively against a chromatinized substrate in vitro (Fig. 6C,D). While the increased H3K27me3 in the absence of

AEBP2$^L$ would not be predicted to directly affect upstream non-canonical PRC1, it would be expected to lead to increased canonical PRC1 recruitment and gene repression (Bracken et al, 2019).

The C-terminal region that is shared by AEBP2$^L$ and AEBP2$^S$ is responsible for binding to the PRC2 core, H2AK119ub1 and the nucleosome (Ciferri et al, 2012; Chen et al, 2018b; Kasinath et al, 2018; Zhang et al, 2019; Kasinath et al, 2021). While this region of AEBP2 is largely conserved, conservation analysis of the AEBP2$^L$ sequence revealed that its N-terminal region has arisen recently (Fig. 5B) and is largely disordered (Fig. 5A). This made structural studies of AEBP2$^L$ challenging, with all currently published structures of AEBP2 opting for AEBP2$^S$, which is sufficient to form a complex with the PRC2 core (Ciferri et al, 2012; Kasinath et al, 2018) and to bind a nucleosome (Kasinath et al, 2021). We solved a high-resolution structure of PRC2–AEBP2$^L$ using Cryo-EM, which indeed shows that AEBP2$^L$ maintains the catalytic and structural form of the PRC2 core in the same conformation as in PRC2–AEBP2$^S$, and that the AEBP2$^L$ N-terminal is flexible and dynamic. AEBP2$^S$ has been shown to enhance PRC2 HMTase activity in vitro (Cao et al, 2002; Kalb et al, 2014; Lee et al, 2018), and the KR motif present in both short and long AEBP2 isoforms is responsible for enhanced nucleosome binding and HMTase stimulation (Lee et al, 2018). This, combined with our binding assays (Fig. 1), the Cryo-EM structure (Fig. 5C), and the histone methyltransferase assays that were carried out on nucleosomal substrates (Figs. 5G, 6C), DNA-free histones and histone tail peptides (Figs. 6D and EV7B), all collectively point to the same inhibitory mechanism: the AEBP2$^L$ N-terminal region interferes with the DNA-binding activity of the PRC2-AEBP2 complex.

The idea that different protein isoforms are expressed from the same locus, yet have different functions, is emerging as a paradigm in chromatin biology. For example, different isoforms of BRD4 explain why earlier studies reported the BRD4 reader protein as a tumour suppressor, while others described it as an oncogene (Crawford et al, 2008; Shi et al, 2014; Shu et al, 2016; Han et al, 2020; Wu et al, 2020). Of note, our labs have previously characterised a vertebrate-specific splice-form of the *LCOR* gene (PALI1), which contains an additional extended exon, called *C10orf12* (Conway et al, 2018; Zhang et al, 2021a). This splice-form, but not the canonical form of LCOR, binds to PRC2 and modulates its activity. In addition, a recent study reported a novel isoform of SUZ12, present only in placental mammals, which promotes PRC2 dimerisation and deposition of H3K27me3 at Polycomb target genes (Arecco et al, 2024). Such in-depth analyses of alternative isoforms are often hindered by the lack of isoform-specific antibodies, which necessitates the use of tags or isoform-specific genetic knockouts to accurately assign a function. Considering our results, future studies will need to consider which isoform of AEBP2 is being utilised in experiments, given the opposing functions of AEBP2$^S$ and AEBP2$^L$. This will likely affect studies considering not only PRC2–AEBP2 complexes, but also PRC2–AEBP2–JARID2 complexes. More broadly, while further research is needed to understand the potential functions of hybrid complexes containing JARID2 and MTF2, our findings highlight the importance of distinguishing PRC2 subtypes based on subunit isoforms, rather than solely relying on the previously established classification of PRC2.1 and PRC2.2 (Hauri et al, 2016).

Our work also helps explain the unexpected phenotype of *Aebp2* KO mice, which exhibit a Trithorax skeletal phenotype and die perinatally (Grijzenhout et al, 2016). We propose a model in which the antagonistic action of AEBP2$^L$ functions to limit the activity of PRC2 in somatic cells. Importantly, our findings underscore that even moderate increases in PRC2 binding and H3K27me3 deposition at Polycomb target genes, as observed in ESCs lacking AEBP2$^L$, can have significant biological consequences, consistent with Trithorax loss-of-function phenotypes observed in vivo. While this would require further investigation, we speculate that loss of only the AEBP2$^L$ isoform would lead to increased H3K27me3 and consequent cPRC1-mediated repression of Polycomb target genes, thereby phenocopying the Trithorax phenotype observed in *Aebp2* KO mice. The short and long AEBP2 isoforms were previously implicated as being embryo-specific and adult-specific, respectively (Kim et al, 2009). However, data herein demonstrates that AEBP2$^L$ is broadly expressed in both embryos and in adult tissues and becomes the predominant isoform prior to mid-gestation (Figs. 3A,B and EV3A,B). Taken together, our work indicates that both isoforms of AEBP2 are crucial during mammalian development to differentially counterbalance the chromatin-binding activities of PRC2.

# Methods

**Reagents and tools table**

| Reagent/resource | Reference or source | Identifier or catalogue number |
|---|---|---|
| **Experimental models** | | |
| AEBP2 KO cell lines (E14) | This study | N/A |
| Rescue mESCs (E14) | This study | N/A |
| mESCs TetR-fussed AEBP2$^L$ and AEBP2$^S$ | This study | N/A |
| C57BL/6J mice | | |
| **Recombinant DNA** | | |
| pLENTI 3xFlag-2xHA (empty) | Bracken lab | N/A |
| pLENTI 3xFlag-2xHA-AEBP2$^S$ | This study | N/A |
| pLENTI 3xFlag-2xHA-AEBP2$^L$ | This study | N/A |
| pPAX | Brien et al, 2015 | N/A |
| pVSVG | Brien et al, 2015 | N/A |
| pSpCas9 (BB)-2A-eGFP | Addgene | Px458 |
| MoClo Baculo Toolkit | Lai et al (2025) | Addgene Kit #1000000256 |
| biGBac toolkit | Weissmann et al (2016) | Addgene Kit #1000000088 |
| pFB1.HMBP.PrS.EZH2 | Zhang et al (2019) | Addgene #125161 |
| pFB1.HMBP.PrS.JARID2 | Zhang et al (2019) | Addgene #125165 |
| pFB1.HMBP.PrS.SUZ12 | Zhang et al (2019) | Addgene #125162 |
| pFB1.HMBP.PrS.EED | Zhang et al (2019) | Addgene #125163 |

| Reagent/resource | Reference or source | Identifier or catalogue number |
|---|---|---|
| pFB1.HMBP.PrS.RBBP4 | Zhang et al (2019) | Addgene #125164 |
| pFB1.HMBP.PrS.AEBP2(iso3) | Zhang et al (2019) | Addgene #125165 |
| pFB1.HMBP-Prs.AEBP2iso2 | This study | |
| pFB1.HMBP-Prs.AEBP2iso1 | This study | |
| pFB1.HMBP-Prs.AEBP2iso2.mt1A | This study | |
| pFB1.HMBP-Prs.AEBP2iso2.mt1K | This study | |
| pFB1.HMBP-Prs.AEBP2iso2.mt1Δ | This study | |
| pFB1.HMBP-Prs.AEBP2iso2.mt2A | This study | |
| pFB1.HMBP-Prs.AEBP2iso2.mt2K | This study | |
| pFB1.HMBP-Prs.AEBP2iso2.mt2Δ | This study | |
| pFB1.HMBP-Prs.AEBP2iso2.mt1A2A | This study | |
| pFB1.HMBP-Prs.AEBP2iso2.mt1K2K | This study | |
| pFB1.HMBP-Prs.AEBP2iso2.mt1Δ2Δ | This study | |
| pBIG1A.HMBP.PrS.PRC2-trunc | This study | |
| pMB.PRC2-AEBP2LTEV-JARID2 | This study | |
| pMB. AEBP2LTEV | This study | |
| pMB.MBP.JARID2 | This study | |
| ATOH1_pUC18 | Zhang et al (2021) | Addgene #191250 |
| pET3.H3.1 | Luger et al (1997) | |
| pET3.H2A | Luger et al (1997) | |
| pET3.H2B | Luger et al (1997) | |
| pET3.H4 | Luger et al (1997) | |
| pET3.H2A-T120C | Zhang et al (2021) | |
| pWidom601.CDKN2Bp.601 | Zhang et al (2019) | |
| **Antibodies** | | |
| AEBP2 | Cell Signalling Technology | 14129S |
| SUZ12 | Cell Signalling Technology | 3737S |
| EPOP | Active Motif | 61753 |
| MTF2 | Proteintech | 16208-1-AP |
| GAPDH | Proteintech | 60004-1 |
| JARID2 | Cell Signalling Technology | 13594S |
| EZH2 (WB of tissues) | Abcam | Ab191250 |
| EZH2 (WB, IP) | Bracken et al, 2019 | AC22 |
| CBX7 | Abcam | 21873 |
| FLAG | Sigma-Aldrich | F1804 |
| TBP | Cell Signalling Technology | 8515 |
| H3K27me1 | Active Motif | 61015 |
| H3K27me2 (MTase) | Abcam | AB24684 |
| H3K27me2 (WB) | Cell Signalling Technology | 9728 |
| H3K27me3 | Cell Signalling Technology | 9733 |
| H3 | Abcam | ab1791 |
| IgG (ChIP) | Millipore | 12-370 |
| IgG (IP) | Cell Signalling Technology | 2729 |
| IgG Donkey anti-mouse IgG HRP (MTase) | Jackson Immuno Research | 715-035-150 |
| IgG Goat anti-rabbit IgG-HRP (MTase) (discontinued) | Santa Cruz Biotechnology | SANTSC-2030 |
| IgG Goat anti-rabbit IgG-HRP (MTase) | Abcam | AB6271 |
| IgG Goat Anti-Rabbit (H + L) Secondary Antibody, DyLight™ Antibody 800 4X PEG (TetO) | Invitrogen | SA5-35571 |
| IgG Goat Anti-Mouse (H + L) Secondary Antibody, DyLight™ Antibody 680 (TetO) | Invitrogen | 35518 |
| **Oligonucleotides and other sequence-based reagents** | | |
| AEBP2 CRISPR sgRNAs | This study | Dataset EV1 |
| ChIP-qPCR primers | This study | Dataset EV1 |
| TetO system sequences | This study | Dataset EV1 |
| CpG46 DNA probe | Zhang et al (2021) | |
| CpG24 DNA probe | Zhang et al (2019) | |
| **Chemicals, enzymes and other reagents** | | |
| 2i-PD0325901 | Sigma-Aldrich | PZ0162 |
| 2i-CHIR99021 | Sigma-Aldrich | SML1046 |
| Trypsin-EDTA (0.25%) | Gibco | 25200056 |
| GMEM medium | Sigma-Aldrich | G5154 |
| NDiff227 medium | Takara | Y40002 |
| Activin A | Peprotech | 120-14 |
| bFGF Recombinant Human Protein | Gibco | 13256029 |

| Reagent/resource | Reference or source | Identifier or catalogue number |
|---|---|---|
| Fibronectin | Millipore | FC010 |
| FBS | Gibco | 10270106 |
| Penicillin/streptomycin | Gibco | 15140122 |
| GlutaMax | Gibco | 35050061 |
| NEAA | Gibco | 11140035 |
| Lipofectamine 2000 | Thermo Fisher Scientific | 11668019 |
| QuantSeq 3' mRNA-Seq Library Prep Kit FWD | Lexoge | 015.24 |
| CUTANA™ pAG-MNase | Stratech Scientific | 15-1116 |
| CUTANA™ Concanavalin A-Conjugated Paramagnetic Beads | Stratech Scientific | 21-1411 |
| Formaldehyde | Sigma-Aldrich | 252549 |
| Triton X-100 | Sigma-Aldrich | T8787 |
| Proteinase K | Sigma-Aldrich | P2308 |
| Benzonase | Sigma-Aldrich | E1014 |
| RNase A | Thermo Fisher Scientific | CEN0531 |
| AMPure beads | Beckman Counter | A63881 |
| Protein G Dynabeads | Invitrogen | 10004D |
| Protein A Dynabeads | Invitrogen | 10002D |
| High Sensitivity D1000 Reagents | Agilent | 5067-5585 |
| High Sensitivity D1000 ScreenTape | Agilent | 5067-5584 |
| Qubit dsDNA High Sensitivity Assay Kit | Thermo Fisher Scientific | Q32854 |
| NEBNext Ultra II DNA Library Kit | Illumina | E7645 |
| NEBNext Multiplex Oligos for Illumina | New England Biolabs | 7335 |
| Monarch PCR & DNA Cleanup Kit | New England Biolabs | T1030L |
| Luna Universal qPCR Master Mix | New England Biolabs | M3003E |
| Qiagen MinElute PCR Purification kit | Qiagen | 280024 |
| Pierce™ RIPA buffer | ThermoScientific | 89901 |
| EDTA-free protease inhibitor cocktail | Roche | 4693132001 |
| IGEPAL CA-630 | Merck | I8896 |
| Phusion DNA Polymerase | NEB | M0530 |
| Gibson Assembly | NEB | E2611 |
| Superose 6 Increase 10/300 | Cytiva | 29091596 |
| NEBridge Ligase Master Mix | NEB | M1100 |
| BsaI-HFv2 | NEB | R3733 |
| BsmBI-v2 | NEB | R0739 |
| NP-40 Alternative | Millipore | 492016 |

| Reagent/resource | Reference or source | Identifier or catalogue number |
|---|---|---|
| PMSF | Sigma | P7626-25G |
| Benzamidine HCl | Sigma | B6506 |
| Bestatin | Sigma | 200484 |
| E64 | Abcam | AB141418 |
| Leupeptin hemisulphate | Sigma | L2884 |
| Aprotinin | Abcam | AB146286 |
| Pepstatin A | Millipore | 516481 |
| Amylose resin | NEB | E8021 |
| SAM | NEB | B9003S |
| HiTrap Heparin HP affinity columns | Cytiva | 17040703 |
| 3-8% Tris-Acetate gel | ThermoFisher | EA0375BOX |
| MES SDS Running buffer | ThermoFisher | NP0002 |
| 2-Mercaptoethanol | Sigma-Aldrich | M3148 |
| InstantBlue Coomassie protein stain | Expedeon | ISB1L |
| HiTrap Q HP column | Cytiva | 17115401 |
| Slide-A-Lyzer dialysis device | ThermoFisher | 69572 |
| Dialysis tubing | Spectrum | 888-11527 |
| BCA assay | ThermoFisher | 23252 |
| S-[methyl-14C]-adenosyl-L-methionine | PerkinElmer | NEC363050UC |
| LDS loading dye | Thermo Fisher Scientific | NP0007 |
| MTase-Glo assays | Promega | V7602 |
| Nitrocellulose membrane | Amersham Protran | 10600002 |
| StartingBlock | Sigma | 37539 |
| SuperSignal™ West Pico PLUS Chemiluminescent Substrate | Sigma | 34580 |
| Nonidet P40 | Roche | 11754599001 |
| Bovine serum albumin | NEB | B9000S |
| Salmon sperm DNA | Sigma | 15632011 |
| Recombinant BSA | NEB | B9200S |
| Methyl-PEG-NHS-Ester | Thermo Fisher Scientific | 22685 |
| Quantifoil (R1.2/1.3, Cu 200) | Quantifoil | |
| MAX Efficiency DH10Bac Competent Cells | ThermoFisher | 10361012 |
| XL1-Blue Competent Cells | Agilent | 200249 |
| BL21 (DE3) Rosetta | Merck | 70954-3 |
| SF9 | ThermoFisher | 11496015 |
| Hi5 | ThermoFisher | B85502 |
| **Software** | | |
| Botwie2, v2.3.4.3 | Langmead and Salzberg (2012) | http://bowtie-bio.sourceforge.net/bowtie2/index.shtml |

| Reagent/resource | Reference or source | Identifier or catalogue number |
|---|---|---|
| Fastqc, v0.11.9 | Andrews (2010) | http://www.bioinformatics.babraham.ac.uk/projects/fastqc/ |
| macs2, v2.2.7.1 | Feng et al (2012) | https://github.com/macs3-project/MACS/releases/tag/v2.2.7.1 |
| samtools, v1.9 | Bonfield et al (2021) | http://www.htslib.org/ |
| bedtools, v2.27.1 | Quinlan and Hall (2010) | https://bedtools.readthedocs.io/en/latest/ |
| deeptools, v3.3.0 | Ramirez et al (2016) | https://deeptools.readthedocs.io/en/develop/ |
| DESeq2, v1.22.1 | Love et al (2014) | https://bioconductor.org/packages/release/bioc/html/DESeq2.html |
| featureCounts, v1.6.4 | Liao et al (2019) | https://rdocumentation.org/packages/Rsubread/versions/1.16.1 |
| R:pheatmap,v1.0.12 | Kolde (2025) | https://cran.r-project.org/web/packages/pheatmap/index.html |
| R:ggplot2, v3.3.3 | Wickham (2016) | https://cran.r-project.org/web/packages/ggplot2/index.html |
| R v3.5.1 | R Core Team | https://cran.r-project.org/ |
| Prism v9.4.0 | GraphPad | https://www.graphpad.com/features |
| Image quant 8.1 | Cytiva | |
| Relion v3.1.2 | Scheres (2012) | https://relion.readthedocs.io/en/latest/Reference/Conventions.html |
| MotionCor2 | Zheng et al (2017) | |
| CTFFIND4 | Rohou and Grigorieff (2015) | |
| crYOLO | Wagner et al (2019) | https://cryolo.readthedocs.io/en/stable/ |
| CryoSPARC | Punjani et al (2017) | https://cryosparc.com/ |
| Chimera | Pettersen et al (2004) | https://www.cgl.ucsf.edu/chimera/ |
| COOT | Emsley et al (2010) | https://www2.mrc-lmb.cam.ac.uk/personal/pemsley/coot/ |
| PHENIX | Afonine et al (2018) | https://phenix-online.org/documentation/overviews/cryo-em_index.html |
| MOLPROBITY | Chen et al (2010) | |
| **Other** | | |
| Cryomiller-precellys evolution touch | Bertin | |
| Qubit 3.0 Fluorometer | Invitrogen | |
| 2200 TapeStation System | Agilent | |

| Reagent/resource | Reference or source | Identifier or catalogue number |
|---|---|---|
| Licor Odyssey Fc Imager | Licor | |
| ChemiDoc imaging system | Bio-Rad | |
| PHERAstar plate reader | BMG Labtech | |
| FLUOstar OPTIMA plate reader | BMG Labtech | |
| Titan Krios electron microscope | FEI | |
| K3 Summit direct electron detector | Gatan | |
| Vitrobot Mark IV | FEI | |
| Typhoon Trio phosphorimager | Cytiva | |
| Typhoon 5 | Cytiva | |
| NextSeq 500 | Illumina | |
| NextSeq 500 High Output v2 kit (75 cycles) | Illumina | |

## Cell culture

E14 mouse embryonic stem cells (ESCs) were grown on gelatinised culture dishes in GMEM (Sigma) supplemented with 20% FBS (Gibco), 100 U/mL penicillin, 100 U/mL streptomycin (Gibco), 50 µM β-mercaptoethanol (Sigma), 1:100 GlutaMax, 1:100 non-essential amino acids (Gibco), 1 mM sodium pyruvate (Gibco), 1:500 homemade leukaemia inhibitory factor (LIF), and the 2i GSK inhibitor CHIR99021 (Millipore) and MEK inhibitor (PD0325901) at final concentrations of 3 µM and 1 µM, respectively.

For Epiblast-like cellular differentiation, ESCs were counted and washed twice with PBS (Sigma) to get rid of residual FBS and pluripotency factors. In total, $0.6 \times 10^6$ cells were seeded onto six-well dishes coated with 16 µg/ml Fibronectin in NDiff227 media (Takara) supplemented with Activin A, bFGF and KSR. The media was changed after 24 hr, and cells were harvested for analysis 48 h post seeding.

## Generation of stable cell lines

Lentiviral particles were generated using PEI transfection of HEK293T cells with 4 µg plasmid of interest (pLENTI 3xFlag-2xHA AEBP2$^{L/S}$ combined with 2 µg PAX8 packaging and 3 µg VSVG envelope vectors. The supernatant containing viral particles was harvested at 48 and 72 h, filtered through a 0.45-µm filter and stored at 4 °C. ESCs were infected with lentivirus using the spinoculation method in a six-well dish. In brief, cells were seeded at 60–80% confluency and allowed to adhere to the dish prior to infection. Then, the media was replaced with viral supernatant supplemented with 1:1000 Polybrene (stock concentration 50 mg/mL), the plate was sealed with parafilm and spun at RT, 1700 rpm for 1 h, after which media was gently replaced. 24 h post-infection, an antibiotic (1 µg/mL Puromycin) was added to the media to allow for selection of clones expressing the desired constructs. Cells were grown in an antibiotic until the complete death of the control, non-transduced cells, after which expression of the constructs was confirmed by western blot.

## Generation of Aebp2 knock-out ESCs cell lines using CRISPR-Cas9

$Aebp2^L$ and $Aebp2^S$ knock-out ESCs were generated by transfecting WT E14 ESCs with 2 pSpCas9 (BB)-2A-eGFP vectors (Addgene, px458) containing gRNAs targeting each side of $Aebp2^L$ Exon 1 or $Aebp2^S$ Promoter 1, using Lipofectamine 2000 as per the manufacturer's recommendations. Complete knock out of all Aebp2 isoforms was generated by targeting each side of the first common exon. sgRNA sequences are available in Dataset EV1. The same vector without sgRNA was used as a negative control (EV). Forty-eight hours after transfection, the GFP high population of cells was collected by FACS using BD FACSAria Fusion High Performance Cell Sorter and individual cells with the highest GFP levels were seeded to each well on 96-well plates. Individual clones were expanded and genotyped by amplifying the region around the PAM site and screening for deletions by gel electrophoresis and on the protein level by western blot.

## Preparation of cellular lysates and western blotting

Cells were scraped down, washed in PBS and lysed in ice-cold High Salt buffer (50 mM Tris-HCl, pH 7.2, 300 mM NaCl, 0.5% (v/v) NP-40, 1 mM EDTA pH 7.4, 2 μg/mL Aprotinin, 1 μg/mL Leupeptin, 1 mM PMSF). Cells were then sonicated and incubated for 20 min at 4 °C while rotating to ensure sufficient lysis. The lysates were then clarified at 14,000 rpm at 4 °C for 25 min. Lysates were then separated on SDS–PAGE gels and transferred to nitrocellulose membranes. Membranes were subsequently probed using the relevant primary and secondary antibodies. Relative protein levels were then determined by chemiluminescence in an Odyssey LiCOR Fc imaging system.

## Cellular fractionations

Cytoplasmic protein extract was prepared by resuspending the cell pellet in three volumes of Buffer A (25 mM HEPES pH 7.6, 5 mM $MgCl_2$, 25 mM KCL, 0.05 mM EDTA, 10% glycerol, 0.1% NP40, 2 μg/mL Aprotinin, 1 μg/ml Leupeptin, 1 mM PMSF) and incubated on ice for 15 min. The lysate was then dounced 8 times with a loose pestle to break down the cell membrane. Lysates were centrifuged for 5 min at 500 rcf at 4 °C to pellet the nuclei and the supernatant was collected and labelled 'Cytoplasmic fraction'. For preparation of the nuclear soluble fraction nuclear pellets were resuspended in Buffer S1 (120 mM NaCl, 20 mM HEPES, 1.5 mM $MgCl_2$, 0.2 mM EDTA, 10% glycerol, 2 μg/mL Aprotinin, 1 μg/mL Leupeptin, 1 mM PMSF) and dounced 15 times with a tight pestle to break down the nuclear membrane, and then centrifuged for 15 min at maximum speed to pellet the chromatin bound and insoluble proteins. The supernatant was collected and labelled 'Nuclear soluble fraction'. Finally, the chromatin-bound fraction was extracted by resuspending the remaining pellet in Buffer S2 (420 mM NaCl, 20 mM HEPES, 1.5 mM $MgCl_2$, 0.2 mM EDTA, 10% glycerol, 2 μg/mL Aprotinin, 1 μg/mL Leupeptin, 1 mM PMSF) and rotated at 4 °C for 30 min. Samples were diluted with an equal volume of Buffer D (25 mM HEPES, 5 mM $MgCl_2$, 0.2 mM EDTA, 2 μg/mL Aprotinin, 1 μg/mL Leupeptin, 1 mM PMSF). Benzonase was added and sampled rotated at 4 °C for 1.5 h. Samples were centrifuged at maximum speed for 30 min and the supernatant collected and labelled 'Chromatin bound fraction'. The protein concentrations of all fractions were normalised by Bradford assay prior to running samples on SDS–PAGE gels.

## Immunoprecipitation coupled with mass spectrometry (IP-mass spec)

Cells were harvested, washed with PBS and resuspended in buffer A (25 mM HEPES pH 7.6, 5 mM $MgCl_2$, 25 mM KCL, 0.05 mM EDTA, 10% glycerol, 0.1% NP40, 2 μg/ml Aprotinin, 1 μg/ml Leupeptin, 1 mM PMSF). Samples were incubated at 4 °C for 10 min rotating. Nuclei were spun at 500 rcf for 10 min at 4 °C and the supernatant was discarded. Nuclear pellets were then resuspended in Buffer C (20 mM HEPES pH 7.9, 0.2 mM EDTA, 1.5 mM $MgCl_2$, 20% glycerol, 420 mM NaCl, 2 μg/ml Aprotinin, 1 μg/ml Leupeptin, 1 mM PMSF). In all, 3 M $(NH_4)_2SO_4$ was added to buffer C for a final concentration of 300 mM. Samples were incubated on ice for 20 min and then spun at 350,000 rcf for 15 min at 4 °C. The supernatant was collected and was added to 300 mg/ml of $(NH_4)_2SO_4$. Samples were mixed thoroughly and incubated for 20 min rotating at 4 °C. Subsequently, they were spun again at 350,000 rcf for 15 min at 4 °C. The supernatant was now discarded, and the pellet was resuspended in IP buffer (300 nM NaCl, 50 nM Tris-HCl pH 7.5, 1 nM EDTA, 1% Tritox-X100, 2 μg/ml Aprotinin, 1 μg/ml Leupeptin, 1 mM PMSF). Bradford assays were performed to quantify the total protein concentration. Up to 1 mg of nuclear protein was used for each IP experiment, with 250 U/ml of Benzonase Nuclease (Sigma). Antibodies were added to protein samples and incubated overnight, rotating at 4 °C. Protein G Dynabeads (Invitrogen) were washed five times with IP buffer before addition to the samples. Subsequently, they were left rotating for 2 h at 4 °C. Beads were collected on a magnet, the supernatant was aspirated, and IP buffer was added to wash the beads. This wash step was repeated five times, ending in protein elution in Bolt LDS. After the addition of 1 M DTT and incubation at 95 °C for 10 min, samples were used for western blot analysis.

## IP-mass spec analysis

All immunoprecipitations for mass spectrometry were performed in triplicate. After the final wash, beads were resuspended in 50 μL elution buffer (2 M Urea, 100 mM Tris pH 8, 10 mM DTT) and incubated 20 min on a shaker (1300 rpm) at RT. After incubation, iodoacetamide was added to a final concentration of 50 mM, followed by 10 min shaking in the dark at RT. Partial digestion and elution from the beads were initiated by adding 0.25 mg Trypsin (Promega; V5113) for 2 h. The supernatant containing the IP samples was collected and the beads were resuspended in 50 μL elution buffer followed by a 5 min incubation shaking at RT. Both supernatants were combined, and 0.1 mg Trypsin was added, followed by overnight incubation at RT. Samples were acidified by adding trifluoroacetic acid (TFA) (final concentration 0.5%). The resulting digested samples were desalted and purified using a homemade StageTip for which two C18 discs were inserted into a 200 μL tip and then placed in a homemade adaptor. The peptides were eluted from StageTips with buffer B (80% acetonitrile, 0.1% formic acid), concentrated to 5 μL by SpeedVac centrifugation at room temperature, and filled up to 12 μL using buffer A (0.1% formic acid). Pulldown samples were measured on an Orbitrap Exploris 480 (Thermo Fisher Scientific) using a gradient from 7 to

30% Buffer B for 44 min followed by washes at 60% then 95% Buffer B resulting in total of 60 min data collection time. Scans were collected in data-dependent top speed mode with dynamic exclusion set at 45 s. Acquired mass spectra were analysed with MaxQuant 2.1.4.0 with default settings, with the addition of deamination as variable modification and match between runs set with fractions, also algorithms for label-free quantification and iBAQ (intensity-based absolute quantification) were enabled. Peptide search was done against a mouse UniProt protein database downloaded in June 2017 and modified to include the amino acid sequences for PALI1. All MaxQuant output was analysed using Excel (version 16.85) and Graphpad (version 10.1.0).

## Chromatin immunoprecipitations (ChIP)

Cells were washed once with PBS before crosslinking for 10 min with PBS containing 1% formaldehyde (Sigma). Crosslinking was quenched with 0.125 M Glycine for 5 min before two PBS washes. The crosslinked cells were lysed in 6 mL of SDS-Lysis buffer (100 mM NaCl, 50 mM Tris pH 8.1, 5 mM EDTA pH 8.0, 0.02% $NaN_3$, 0.5% SDS, 2 µg/mL Aprotinin, 1 µg/mL Leupeptin, 1 mM PMSF). Chromatin was pelleted by centrifugation at 1200 rpm for 5 min at room temperature. The supernatant was then discarded, and the chromatin was resuspended in 3 mL of ChIP buffer (2:1 dilution of SDS-Lysis buffer: Triton dilution buffer [100 mM Tris pH 8.6, 100 mM NaCl, 5 mM EDTA pH 8.0, 0.02% $NaN_3$, 5% Triton X-100, 2 µg/mL Aprotinin, 1 µg/mL Leupeptin, 1 mM PMSF]). Chromatin was sheared to ~200–600 bp fragments by sonication on Soniprep 150 probe sonicator for a total of 4.5 min at 50% amplitude, pulsing for 1 s on 4 s off. Sonicated chromatin was incubated overnight with antibodies while rotating at 4 °C. Following clarification, the chromatin was incubated for 3 h with 40 µL of protein A or G Dynabeads (ThermoFisher). After incubation, the beads were washed three times in Mixed Micelle Buffer (150 mM NaCl, 20 mM Tris pH 8.1, 5 mM EDTA pH 8.0, 5.2% Sucrose, 0.02% $NaN_3$, 1% Triton X-100, 0.2% SDS), twice with Buffer 500 (0.1% Sodium Deoxycholate, 1 mM EDTA pH 8.0, 50 mM HEPES pH 7.5, 1% Triton X-100, 0.02% $NaN_3$), twice with LiCl detergent wash (0.5% sodium deoxycholate, 1 mM EDTA pH 8.0, 250 mM LiCl, 0.5% NP-40, 10 mM Tris pH 8.0, 0.02% $NaN_3$) and finally one wash with TE. Immunoprecipitated material was eluted from the beads in Elution buffer (0.1 M $NaHCO_3$, 1% SDS) while shaking for 1 h at 65 °C. The supernatant was retained and incubated overnight at 65 °C while shaking to reverse the crosslinks. The eluted complexes were then subject to RNase (Thermo Fisher) and Proteinase K (Sigma) treatment prior to DNA clean up with Qiagen MinElute PCR Purification Kit (Qiagen, 28006). ChIP enrichments were analysed by qPCR using the SYBR Green I detection chemistry (M3003E NEB) on an Applied Biosystems Quant Studio 3 platform.

## ChIP-Rx library preparation

Quantitative chromatin immunoprecipitation relative to a reference exogenous genome (ChIP-Rx) was performed, as described previously. A total of 10% human chromatin (from NTERA2 embryonic carcinoma cells) was added to each ESC chromatin lysate at the beginning of the workflow before sonication. Once exogenous and ESC chromatin were combined, the sample was treated as a single ChIP-Seq experiment until completion of DNA sequencing. Following the ChIP experiment, the precipitated DNA

was quantified using the Qubit dsDNA High Sensitivity Assay Kit (ThermoFisher, Q32854). A Total of 0.5–10 ng of DNA from each ChIP-Rx experiment was used for library preparation using the NEBNext Ultra II DNA Library Kit for Illumina (E7645) and NEBNext Multiplex Oligos for Illumina (Set#1, NEB #7335). Following adaptor ligation, DNA was PCR amplified for 5–11 cycles, depending on the amount of input DNA. DNA purification was then performed using Beckman Coulter Genomics Ampure XP (A63881). The quality of DNA libraries was analysed on a High Sensitivity D1000 Screen Tape (Agilent). The resulting libraries were then used for cluster generation and sequencing using an Illumina NextSeq 500, with 75 bp read length.

## Cut&Run

Cells were collected using Accutase (STEMCELL Technologies) and counted to obtain 1 million cells/antibody tested. Cells were crosslinked using 0.1% formaldehyde at room temperature for 1 min. Glycine at 0.125 M was added to quench the formaldehyde, followed by a 5-min incubation at room temperature. Fixed cells were washed with PBS and incubated on ice for 10 min in nuclear extraction buffer (20 mM HEPES pH 7.5, 10 mM KCL, 0.1% Triton X-100, 20% Glycerol, 1× protease inhibitor cocktail and 0.5 mM spermidine). Nuclei were collected by cold centrifugation (4 °C) at 600 × g. Pellets were subsequently resuspended in cold nuclei extraction buffer (100 µL per sample). The CUT&RUN experiments were performed following the Epicypher CUT&RUN Protocol v1.5.1. The Monarch PCR & DNA Cleanup Kit (New England Biolabs) protocol was used to purify the DNA.

## Preparation of extracts from mouse tissues and their immunoblotting

Mice used in this study were C57BL/6 J (RRID:IMSR_JAX:000664), wild-type (no genetic modification). They were agisted in specific-pathogen-free (SPF) conditions at the Monash Animal Research Platform (MARP) and Animal Research Facility (ARL) at Monash University; they were housed in OptiMICE EVC caging system, with dust-free sawdust bedding, and fed irradiated mice pellets (Barastoc - Mice Breeder Cubes Irradiated). Tissues were harvested from freshly culled adult mice (234-day-old female for nuclear extract, 241-day-old female for total protein extracts) or newborn mice on Postnatal day 0 (P0), and immediately flash-frozen. To obtain embryos in the desired embryonic days, timed mating of mice was set up. Dams were sacrificed at the desired embryonic day (E9.5, E12.5 and E16.5), and embryos were extracted, humanely killed and flash-frozen. P0 mice and embryos were unsexed. All animal experimentation was conducted following the Australian National Health and Medical Research Council Code of Practice for the Care and Use of Animals for Scientific Purposes guidelines for housing and care of laboratory animals and performed in accordance with Institutional regulations after pertinent review and approval by Monash University Animal Ethics Committee (Animal Breeding Ethics #27334; Animal Ethics Research #36532).

For total protein extracts, tissues or whole embryos were resuspended in 4 mL PierceTM RIPA buffer (ThermoScientific) and lysed using a drill-fitted Dounce homogeniser (20 strokes). The lysates were then clarified at 21,000 × g at 4 °C for 20 min.

For nuclear extracts, nuclear purification from tissues was adapted from (Loft et al, 2021). Frozen tissues were cut in pieces of ~5 × 5 mm, transferred to a 2-mL Eppendorf Safe-Lock tube containing 50 2.0-mm ball bearings, and pulverised using a cryogenic mixer mill (cycles of 45 s at 30 Hz each, with cooling in liquid nitrogen for 1 min in between cycles, until the tissue is pulverised). The pulverised tissues were washed by resuspension in ice-cold PBS, followed by centrifugation at 2000 × g for 3 min. The pellet was resuspended in 4 mL Low Sucrose Buffer (LSB; 250 mM Sucrose, 15 mM Tris-HCl, pH 7.5, 5 mM $MgCl_2$, 25 mM KCl, 0.5 mM Spermidine, 0.15 mM Spermine, 1 mM DTT and EDTA-free protease inhibitor cocktail (Roche, 4693132001; 1 tablet/ 50 mL)) and were subjected to ten rounds of homogenising in a Dounce homogeniser. In all, 300 μL of 5% IGEPAL CA-630 (Merck, I8896) was added and mixed. The tube was incubated for 5 min on ice and was subjected to five more rounds in the Dounce homogeniser. The homogenate was filtered through a 100-μm filter unit into a 15-mL conical tube and spun for 10 min at 600 × g. The pellet was thoroughly resuspended in nine volumes of HSB (LSB with 2 M Sucrose) and split in aliquots of up to 1.8 mL in 2 mL tubes. The samples were spun for 15 min at 15,000 × g at 4 °C, and the pellet was washed in WB solution (LSB with 0.35% IGEPAL CA-630 (Merck, I8896)). Nuclear extract preparation was adapted from (Healy et al, 2019). The nuclei pellet was resuspended in ice-cold High Salt buffer (50 mM Tris-HCl, pH 7.2, 300 mM NaCl, 0.5% (v/v) NP-40, 1 mM EDTA pH 7.4 and EDTA-free protease inhibitor cocktail (Roche, 4693132001) (1 tablet/50 mL)). Cells were then sonicated and incubated for 20 min at 4 °C while rotating to ensure sufficient lysis. The lysates were then clarified at 21,000 × g at 4 °C for 20 min.

Protein content was estimated using Bradford assay. Lysates were then separated on SDS–PAGE gels and transferred to nitrocellulose membranes. Membranes were subsequently probed using the relevant primary and secondary antibodies. Relative protein levels were then determined by chemiluminescence in a ChemiDoc imaging system (Bio-Rad). Antibodies that were used for immunoblotting are listed in the Reagents and Tools table.

## Cloning, expression and purification of recombinant proteins

The open reading frames of human AEBP2 isoforms were cloned into an expression vector with the pFastBac1 backbone, under a 3C protease-cleavable 6xHis-MBP tag. DNA coding for the N-terminal region of AEBP2, containing either mutations or deletions of the two acidic tracts, were synthesised by Genscript. For construct and primer sequences, see Dataset EV3. For each AEBP2 mutant, the N-terminal and C-terminal regions of AEBP2 were PCR amplified using Phusion DNA Polymerase (NEB #M0530) and then cloned into the expression vector of a pFastBac1 backbone, containing a PreScission-cleavable N-terminal hexahistidine-MBP tag, using Gibson Assembly (NEB #E2611). Baculovirus production, titration, infection, and cell harvesting and the purification of PRC2–AEBP2 wild-type and the mutants were carried out as previously described (Zhang et al, 2019), with the exception that a Superose 6 Increase 10/300 size-exclusion column was used instead (GE Healthcare). All the proteins were snap-frozen in liquid nitrogen and stored at −80 °C as single-use aliquots.

PRC2-AEBP2$^L$-JARID2 and its mutant complexes were expressed using three expression vectors, including pFB1.HMBP-JARID2, pFB1.HMBP-AEBP2iso2 (isoform 2, either wild-type or mutant) and a multigene biGBac construct (Weissmann et al, 2016) consist of HMBP-PrS-EZH2, HMBP-PrS-SUZ12(70–685), HMBP-PrS-EED(80–411) and HMBP-PrS-RBBP4, where PrS is a 3C-protease (PreScission) cleavable tag and HMBP is a polyhistidine-maltose binding protein tag (termed pBIG1A.HMPRC2-trunc; see Dataset EV5 for the plasmid sequence). These constructs were used to generate baculovirus stocks in SF9 cells and the baculoviruses were then used for the co-infection of Hi5 cells for protein production.

The PRC2-AEBP2$^{L(TEV)}$-JARID2 was expressed from a multigene baculovirus expression plasmid (pMB.PRC2-AEBP2LTEV-JARID2) where transcription units for all six full-length MBP-tagged subunits (EZH2, SUZ12, EED, RBBP4 and AEBP2 $^{L(TEV)}$) were assembled into one construct by Golden Gate assembly (NEB #M1100, NEB #R3733 and NEB #R0739) using the MoClo Baculo toolkit (Lai et al, 2025) and the sequence was validated using whole plasmid sequencing (Primordium; see Dataset EV5 for the multigene plasmid sequence). Another baculovirus expression vector was generated using the same strategy for the expression of MBP-tagged JARID2 (pMB.MBP.JARID2; see Dataset EV5 for the plasmid sequence). All the multigene baculovirus expression plasmids were validated by whole plasmid sequencing (Primordium) and annotated by pLannotate (McGuffie and Barrick, 2021) (Dataset EV5). Baculovirus stocks were generated using Sf9 cells. For protein expression, Hi5 cells were co-infected with a baculovirus stock that was generated using the multigene plasmid pMB.PRC2-AEBP2LTEV-JARID2 together with an excess amount of baculovirus stock that was made from the pMB.JARID2 plasmid, aiming to increase the stoichiometric incorporation of JARID2 into the PRC2-AEBP2$^{L(TEV)}$-JARID2 complex.

Purification of PRC2-AEBP2$^L$-JARID2 and PRC2-AEBP2$^{L(TEV)}$-JARID2 was similar to a previously described method(Zhang et al, 2019), with some modifications that were made, aiming to reduce the proteolysis of JARID2. Specifically, the frozen pelleted cells were lysed using TBSL Buffer (50 mM Tris pH 8 at 25 °C, 300 mM NaCl, 15% glycerol, NP-40 Alternative (Millipore, 492016), 1 mM TCEP, 1 mM PMSF (Sigma), 5 mM benzamidine HCl (Sigma, B6506), 80 μM Aprotinin (Abcam, AB146286), 4 μM Bestatin (Sigma, 200484), 1.4 μM E64 (Abcam, AB141418), 2.1 μM Leupeptin hemisulphate (Sigma, L2884) and 1.5 μM Pepstatin A (Millipore, 516481)) at 4 °C. The lysate was clarified by centrifugation at 30,000 × g for 30 min using F14-6x250y rotor, and the supernatant was incubated with amylose resin (NEB #E8021) for 40 min with rotation at 4 °C. The bound resin was batch washed with 20 bead volumes of TBSL without protease inhibitors, 20 bead volumes of TBS500 (50 mM Tris pH 8 at 25 °C, 500 mM NaCl), 10 bead volumes of TBS150 (50 mM Tris pH 8 at 25 °C, 150 mM NaCl) and then eluted in 4 bead volume of TBS150 supplemented with 1 mM TCEP and 10 mM Maltose. Heparin affinity purification was carried out as described for the PRC2-AEBP2 complexes using TBS150 as buffer A and TBS2000 (50 mM Tris pH 8 at 25 °C, 2 M NaCl) as buffer B. The desired fractions from Heparin affinity purification were pooled and diluted 1:1 v/v in 2× automethylation buffer (20 mM Tris pH 8 at 25 °C, 5 mM $MgCl_2$, 50 μM SAM (NEB, B9003S) and 2 mM TCEP) and were incubated at 4 °C for 14–18 h.

Gel filtration was done in TBS200 (50 mM Tris pH 8 at 25 °C, 200 mM NaCl). Protein was supplemented with TCEP to 1 mM before being frozen in liquid nitrogen.

To resolve all PRC2 subunits using SDS–PAGE, 3–8% Tris-Acetate gel (ThermoFisher #EA0375BOX) and 1× MES SDS Running buffer (ThermoFisher #NP0002) were used. In all, 3 µg PRC2 complexes were supplemented to a final concentration of 1X LDS sample buffer (ThermoFisher #NP0007) with 1% 2-Mercaptoethanol (Sigma-Aldrich #M3148) and heated at 95 °C for 5 min before loading onto a Tris-Acetate gel. The gel was run for 32 min at 200 V before staining with InstantBlue Coomassie protein stain (Expedon, #ISB1L). To generate TEV-free PRC2-AEBP2$^S$-JARID2 for fluorescence anisotropy assays, PRC2-AEBP2$^{L(TEV)}$-JARID2 was produced as above until the elution from the heparin column, where then TEV protease was added to 0.2 mg/ml and was incubated with the sample in the same automethylation buffer described above for the PRC2-AEBP2$^{L(TEV)}$-JARID2 complex. The TEV enzyme was removed from the produced PRC2-AEBP2$^S$-JARID2 complex during the subsequent size exclusion chromatography step, which was carried out as described above for the PRC2-AEBP2$^{L(TEV)}$-JARID2 complex.

## Nucleosome reconstitution

Histones, octamers, and polynucleosome reconstitution were carried out as previously described (Zhang et al, 2021a). In brief, recombinant histones were purified from inclusion bodies and reconstituted into histone octamers. *ATOH1* DNA was amplified using Pfu DNA polymerase and purified by ion exchange chromatography using HiTrap Q HP column (GE #17-1154-01) with a 10 c.v. gradient starting with buffer A (20 mM Tris-HCl, pH 7.5 at 25 °C, 150 nM NaCl) into 50% buffer B (20 mM Tris-HCl, pH 7.5 at 25 °C, 2 M NaCl), followed by isopropanol precipitation. Chromatin was assembled using gradient salt dialysis at 4 °C. Specifically, chromatin was reconstituted by initially titrating across a range of octamer ratios (from 1:12 to 1:24 DNA:octamer molar ratio) in a 20 µL mixture of 0.15 µM DNA, 20 mM Tris pH 7.5 at 25 °C, 2 M KCl, 1 mM EDTA and 10 mM DTT. For each of these samples, gradient salt dialysis was used at 4 °C in a dialysis device (ThermoFisher #69572), starting from refolding buffer (20 mM Tris pH 7.5 at 25 °C, 2 M KCl, 1 mM EDTA and 1 mM DTT) to a medium salt buffer containing 20 mM Tris pH 7.5 at 25 °C, 250 mM KCl, 1 mM EDTA and 1 mM DTT over 18 h, and the final step dialysis was carried out using a low salt buffer containing 20 mM Tris pH 7.5 at 25 °C, 2.5 mM KCl, 1 mM EDTA, 1 mM DTT. Quality of chromatinized DNA was assessed by 0.8% agarose TBE gel electrophoresis, and the most appropriate molar ratio of DNA:octamer was selected for a large-scale batch. Large-scale reconstitution was conducted as above, except in a volume of 0.2–2 mL using dialysis tubing (Spectrum #888-11527). To concentrate the assembled chromatin, MgCl$_2$ was added to a final concentration of 20 mM, the mixture was incubated for 15 min at room temperature followed by 15 min on ice, and then centrifuged at 4 °C for 20 min at 20,000 rcf. Precipitate was resuspended in the low salt buffer. The concentration of the nucleosome core particles in the arrays was measured using a BCA assay (ThermoFisher #23252). The fluorescently labelled mononucleosome probe used in binding assays was made using Cy5-labelled H2A histones as previously described (Zhang et al, 2021a).

## In vitro HMTase activity assays using radiolabelled S-adenosyl-L-methionine

For the HMTase reactions, each 10 µL reaction contained 0.6 µM PRC2, either 1.2 µM H3.1 or 0.6 µM chromatinized DNA (the concentration of chromatinized DNA was defined as nucleosome core particle-equivalent, such that the concentration of each histone protein was 1.2 µM), and 5 µM S-[methyl-14C]-adenosyl-L-methionine (PerkinElmer, #NEC363050UC). Reactions were incubated in the reaction buffer A (50 mM Tris-HCl pH 8.0 at 30 °C, 100 mM KCl, 2.5 mM MgCl$_2$, 0.1 mM ZnCl$_2$, 2 mM 2-mercaptoethanol and 0.1 mg/mL BSA, 5% v/v glycerol) for 20 or 60 min at 30 °C. Reactions were stopped by adding 4X LDS loading dye (Thermo Fisher Scientific, #NP0007) supplemented with 4% 2-Mercaptoethanol (Sigma-Aldrich #M3148) to a final concentration of 1X LDS and heating at 95 °C for 5 min. The reactions were then loaded onto 16.5% SDS–PAGE gels and ran on ice for 120 min at 160 V in 1× Tris-glycine buffer. Gels were stained with InstantBlue Coomassie protein stain (Expedeon, #ISB1L) before vacuum-drying for 1 h at 80 °C. Dried gels were then exposed to a storage phosphor screen for several days before acquiring radiograms using a Typhoon 5 Imager (GE Healthcare). All experiments were performed in three independent replicates that were carried out on three separate days. Densitometry was carried out using ImageLab software (Bio-Rad). Relative activity was calculated by dividing all densitometry values by the mean value of the AEBP2$^{L(iso2)}$ 60 min data point. The resulting values were then plotted with bars showing standard error using GraphPad Prism software (Version 9.4.0).

## MTase-Glo assays

MTase-Glo assays (Promega V7602) were performed as specified by the manufacturer with some modifications similar to as done previously (Zhang et al, 2021a). Ahead of HMTase assays in Fig. 1, to account for potential batch-to-batch variations that were previously proposed between the automethylation level of EZH2 (Wang et al, 2019), we allowed for the automethylation of PRC2 to occur prior to the main MTase reaction, in the absence of chromatin substrate. Specifically, a SAM pre-incubation mixture was created to include a Reaction Buffer (50 mM Tris-HCl pH 8.15 at 25 °C, 100 mM KCl, 0.5 mM MgCl$_2$, 0.1% Tween-20 and 5 mM DTT), 5 µM PRC2, 20 µM SAM and 10 µM PALIK1241me3 peptide and was incubated for 3 h at 30 °C. The reaction mixtures were then diluted and combined with a chromatin substrate to initiate the histone MTase reaction for the time course. Histone MTase reaction mixture consisted of 50 nM PRC2, 1 µM chromatinized ATOH1 DNA substrate (the concentration defined based on nucleosome-core particle-equivalent, such that the concentration of each histone is 2 µM) and 25 µM SAM in Reaction Buffer, with a reaction volume of 8 µL per time point. Reactions were incubated at 30 °C and quenched using 1 µL of 2.4% v/v TFA, with the reaction stopped at 11 time points between $t = 0$ to $t = 190$ min (time points in H:MM:SS format: 0:00, 03:30, 07:30, 15:40, 23:00, 35:00, 55:00, 1:30:30, 2:10:00, 2:40:00 and 3:10:00). For each enzyme, an identical reaction was set up without the chromatinized ATOH1 DNA substrate, to account for residual automethylation activity of PRC2 and the quantified SAH production values for these were later subtracted from their respective main MTase reaction mixture. The

luminescence signal was developed using reagents supplied with the MTase-Glo assay kit. The plate was read on BMG FLUOstar OPTIMA plate reader (BMG Labtech). SAH levels over the time course for each enzyme were graphed using GraphPad Prism software (Version 9.4.0). Three independent replicates for each condition were performed on 3 different days.

For progress curves, the HMTase reaction was initiated by mixing equal volume of 2× enzyme mix and 2× substrate mix, each in 1× Reaction Buffer as described above, to form the final enzyme and substrate concentrations that are indicated in the figure legend. For HMTase assays with the PRC2-AEBP2$^{L(TEV)}$-JARID2 construct, protein was diluted to 1 µM in 1× Reaction Buffer and incubated with 0.4 mg/ml TEV protease at room temperature. Then, the mixture was further diluted in 1× Reaction Buffer to form the 2× enzyme mix. The 2× enzyme mix and the 2× substrate mix were prepared at room temperature for ~20 min and then cooled on ice for 5 min right before mixing them to initiate the reaction. Then, the 0 min time point was taken, and the reaction was initiated by putting the tubes on a thermocycler preheated to 30 °C. Reactions were stopped at the indicated time points by mixing 4 µL of the reaction mixture with 2 µL of 0.48% v/v TFA. SAH detection was then carried out using the MTase-Glo™ Methyltransferase Assay (Promega V7602), according to the instructions of the manufacturer.

For Michaelis–Menten kinetic analysis, HMTase assays were carried out as described above, using enzymes and substrates in concentrations that are indicated in the figure legend. These reactions were stopped after 60 min by cooling the mixture on ice and adding the same amount of TFA as indicated above. SAH detection was then carried out using the MTase-Glo™ Methyltransferase Assay (Promega V7602), according to the instructions of the manufacturer. The data is fitted using GraphPad Prism software to a nonlinear regression of the Michaelis–Menten kinetic model.

## Immunoblotting of HMTase reaction products

The HMTase reaction was carried out as described above for the Michaelis–Menten kinetic analysis, using enzymes and substrates with concentrations that are indicated in the figure legends. The only exception is that the reactions were stopped after 60 min by adding 10 µl 4× NuPAGE™ LDS Sample Buffer with 1% (v/v) beta-mercaptoethanol into each 30 µL reaction. Volumes of 2.4 µL, 8 µL, 8 µL and 16 µL samples were loaded on gels dedicated for blotting using H3, H3K27me1, H3K27me2 and H3K27me3 antibodies, respectively. The mixture was then separated using SDS–PAGE, using a homemade 10-well 18% acrylamide gel (BioRed) that ran at 160 V for 2 h in tris-glycine buffer on ice. Proteins were transferred to nitrocellulose membrane (Amersham Protran, 10600002) in Tris-glycine transfer buffer supplemented with 20% EtOH and 0.1% SDS for 30 min over ice using 0.5 Amp (Bio-Rad). For the blotting of SUZ12 and JARID2, membranes were cut such that H3K27me1 and H3K27me2 could be immunoblotted separately at the bottom part of the membrane. The blots were blocked by StartingBlock (Sigma, 37539) overnight, and primary antibodies were then diluted in StartingBlock and incubated for 1 h at room temperature with the blots and were washed five times in 1× Tris-Buffered Saline with 0.1% (v/v) Tween (TBST). Then, the blots were incubated for 1 h with secondary antibodies at room temperature. The blots were washed five times in TBST and imaged using SuperSignal™ West Pico PLUS Chemiluminescent Substrate (Sigma, 34580).

## Fluorescence anisotropy

A DNA probe designed to mimic a 24-base long dsDNA hairpin from a CpG island of the *CDKN2B* gene (termed CpG24 DNA, see below for DNA sequence) was synthesised by Integrated DNA Technologies, Inc. CpG24 DNA stored at −80 °C as a small single-use aliquot and before each experiment was incubated for 3 min at 95 °C in 10 mM Tris-HCl pH 7.5 (at 25 °C) in a thermocycler with a heated lid, and was then immediately snap-cooled on ice for 5 min. Next, CpG24 DNA was diluted to 10 nM in FA binding buffer (50 mM Tris-HCl pH 7.5 at 25 °C, 0.05% Nonidet P40 (Roche, no. 11754599001), 0.1 mg/mL bovine serum albumin (NEB, #B9000S), 2 mM 2-mercaptoethanol and either 25, 50 or 100 mM KCl as indicated) and was allowed to fold for 30 min at 37 °C in a closed incubator. 1:2 serial dilutions of protein were prepared with a volume of 20 µL each, with 8 µM of protein as the maximum concentration and FA binding buffer as the diluent. Twenty-one wells for each row of a 384-well plate were used for each series, with the remaining three wells at the end of the row containing only buffer to use as a background measurement for that series. Next, 20 µL of 10 nM CpG24 DNA was added to each of the serially diluted protein solutions to make the volume up to 40 µL, and the contents mixed by pipetting very gently up and down six times. The protein and DNA were allowed to bind at 30 °C for 30 min. Fluorescence anisotropy data were collected using a PHERAstar plate reader (BMG Labtech) at 30 °C (excitation wavelength $\lambda_{ex} = 485$ nm, emission wavelength $\lambda_{em} = 520$ nm). For calculating $K_d$ and Hill, polarisation values were divided by the $B_{max}$, which was set to 1.0 as a constant. The background was subtracted from the average of the three protein-free "blank" wells at the end of each row. $K_d$, Hill and standard deviation values were calculated with GraphPad Prism software (Version 9.4.0) using nonlinear regression for specific binding with Hill slope function. Independent replicates of all fluorescent anisotropy experiments were performed on different days.

CpG24 DNA sequence (3′ fluorescein tag, synthesised by Integrated DNA Technologies (IDT)):

5′-CGCCCTGCCCCGCCTCGCTCTGGCGCTAGCCAGAGC-GAGGCGGGGCAGGGCG-3′.

The fluorescence anisotropy DNA binding assay for the PRC2.2 complexes was carried out as described above, except the binding buffer and the probes were replaced to account for the different complexes. Specifically, the binding buffer was 20 mM Tris pH 8.0, 100 mM NaCl, 0.1 mM ZnCl₂, 0.2 mg/ml BSA, 0.002% (v/v) NP40 and 2 mM 2-mercaptoethanol, and the probe was made of a 96 bases long DNA that was designed to form a DNA hairpin of 46 bp with a sequence originated from a CpG island of the human *CDKN2B* locus. This probe, termed 'CpG46 DNA', was synthesised by IDT with a 3′ fluorescein and the following sequence:

5′-GGCGCCCTGCCCCGCCTCGCTCTGGCAGAGTGGG-GAGCCAGCCGGCGCTAGCCGGCTGGCTCCCCACTCTGC-CAGAGCGAGGCGGGGCAGGGCGCC-3′.

## Electrophoretic mobility shift assay (EMSA)

CpG24 DNA was heated and cooled as described for fluorescence anisotropy. DNA was then allowed to fold at 10 nM concentration for 30 min at 37 °C in EMSA binding buffer (50 mM Tris-HCl pH 7.5 at 25 °C, 100 mM KCl, 0.05% v/v Nonidet P40 (Roche, no.

11754599001), 0.1 mg/ml BSA (NEB B9000S), 5% v/v glycerol, 2 mM 2-mercaptoethanol). 1:2 serial dilutions of protein were prepared with a volume of 10 µL with 8 µM of protein as the maximum concentration and EMSA binding buffer as the diluent. Next, 10 µL of CpG24 DNA was added to each of the serially diluted protein solutions to make the volume up to 20 µL, and the samples were mixed gently by swirling the pipette tip, and the protein and DNA allowed to bind at 30 °C for 30 min in a thermocycler. Samples were spun down at 2000 rpm for 1 min in a microcentrifuge before being loaded on a 0.7% agarose gel buffered with 0.5X TBE at 4 °C. Gel electrophoresis was carried out for 90 min at 6.6 V/cm in an ice box in a 4 °C cold room. Signal acquisition was performed with a Typhoon Trio phosphorimager (GE Healthcare) using fluorescence with a 488 nm laser and Cy2 filter. Densitometry was carried out with ImageQuant TL software (Version 8.1, GE Healthcare). $K_d$, Hill and standard deviation values were calculated with GraphPad Prism software (Version 9.4.0) using nonlinear regression for specific binding with Hill slope function. Data ranges for both dissociation constants and Hill coefficients were calculated based on three independent experiments that were carried out on different days.

For EMSA using nucleosome probes, a Cy5-labelled nucleosome probe was first diluted to 10 nM in 2× EMSA Buffer (100 mM Tris pH 8.0 at 25 °C, 5 mM $MgCl_2$, 0.2 mM $ZnCl_2$, 10% glycerol, 0.1% NP-40 Alternative (Millipore, 492016), 4 mM 2-mercaptoethanol, 0.2 mg/ml recombinant BSA (NEB, B9200S) and 1 ng/µl salmon sperm DNA (Sigma, 15632011)) to form a 2× Probe Mix. For EMSA with the PRC2-AEBP2$^{L(TEV)}$-JARID2 complex, protein was diluted to 1 µM in protein dilution buffer (TBS200 supplemented with 1 mM 2-mercaptoethanol and 0.01 mg/ml BSA) and incubated with 0.2 mg/ml TEV protease at room temperature for 30 min and further serial diluted with protein dilution buffer to form a 2× Protein Mix. Binding carried out by mixing of 5 µL of 2× Probe mix and 5 µL of 2× Protein Mix with final concentrations as indicated in the figure legends, such that the final EMSA binding buffer was designed to include final concentrations of 75 mM Tris pH 8.0 at 25 °C, 100 mM NaCl, 2.5 mM $MgCl_2$, 0.1 mM $ZnCl_2$, 5% Glycerol, 0.05% NP-40 Alternative (Millipore, 492016), 2.5 mM 2-mercaptoethanol, 0.1 mg/ml recombinant BSA (NEB, B9200S), 0.5 ng/µl salmon sperm DNA (Sigma, 15632011)), 5 nM nucleosome probe, and protein as indicated. For EMSA using the PRC2-AEBP2$^L$-JARID2 complexes, the same process was carried out, except for the incubation with TEV. The other steps were as described for the DNA binding EMSA above, with the exception that the fluorescence signal was recorded using the Typhoon 5 (Cytiva) with an LD635 laser and an 670BP30 filter.

## Single-particle cryo-EM sample preparation

Prior to cryo-EM sample preparation, complexes were PEGylated (Zhang et al, 2021b) at 0.9 mg/mL with 5 mM MS(PEG)$_4$ Methyl-PEG-NHS-Ester (Thermo Fisher Scientific) for 2 h at 4 °C. The Sample was buffer exchanged into 200 mM NaCl, 20 mM HEPES pH 7.5, 1 mM TCEP, and concentrated. NP-40 was added to the sample to 0.01%. 4.5 µL of the PEGylated complex at 2.4 mg/mL in 200 mM NaCl, 20 mM HEPES pH 7.5, 1 mM TCEP, 0.01% NP-40 was applied to a freshly glow-discharged Quantifoil (R1.2/1.3, Cu 200 mesh) grid. Samples were vitrified after blotting for 3 s at 4 °C, 100% humidity using a Vitrobot Mark IV (FEI).

## Single-particle cryo-EM data acquisition

Automated data acquisition was performed using a Titan Krios electron microscope (FEI) at 300 kV, equipped with a K3 Summit direct detector (Gatan) and a GIF Quantum energy filter (Gatan). Cryo-EM imaging was performed using nanoprobe EFTEM zero-loss imaging mode with a 10-eV slit width. At a nominal magnification of ×105,000, a magnified pixel size of 0.86 Å was provided. In total, 7175 movies were recorded using a K3 Summit direct detector (Gatan) operated in counting mode. Each movie had a total accumulated dose of 60 e$^-$ Å$^{-2}$, which were fractionated into 50 frames. The EPU software package (Thermo Fisher Scientific) was used for automated data collection and autofocus was set to achieve a defocus range from −0.5 to −2.5 µm.

## Single-particle cryo-EM data processing and model building

Relion (version 3.1.2) was used for the early processing steps (Scheres, 2012). The exposure frames were aligned using Motion-Cor2 (Zheng et al, 2017) to correct for beam-induced motion, and the aligned summed images were used for further processing. The CTF parameters for the micrographs were determined using CTFFIND4 (Rohou and Grigorieff, 2015). crYOLO was used for automatic selection of particles from the micrographs (Wagner et al, 2019). In total, 847,067 particles were selected. Subsequent processing steps were done in CryoSPARC (Punjani et al, 2017). Particles were subjected to two-dimensional classification. After sorting, 104,087 particles were subjected to ab initio three-dimensional classification. This initial classification yielded one class with 45,024 particles that clearly corresponds to an intact complex. This subset was subjected to three-dimensional non-uniform refinement followed by global and local CTF refinement, yielding a 3.64 Å resolution cryo-EM map (Punjani et al, 2020). Subsequent background subtraction and masked refinements of the individual domains and regions did not lead to map improvements. Reported resolution is based on the gold standard FSC = 0.143 criterion.

Structures for PRC2–AEBP2–JARID2 and SUZ12–RBBP4–JARID2–AEBP2 (Chen et al, 2018b; Kasinath et al, 2018, 2021) were used as reference models. Structures were rigid-body docked in Chimera (Pettersen et al, 2004) and relevant subunits were compiled. The resulting model was then improved by iterative manual building in COOT (Emsley et al, 2010) and refinement in PHENIX (Afonine et al, 2018). Refinement was guided by MOLPROBITY statistics (Chen et al, 2010). All structural graphics were prepared using UCSF Chimera (Pettersen et al, 2004). All data collection and refinement statistics are summarised in Table 1.

## Bioinformatic analyses of ChIP-Seq/Rx and CUT & RUN datasets

ChIP-Rx reads were aligned to a metagenome of mm10 and hg38 using bowtie2, with "_hg38" appended to chromosome names prior to combining and indexing. Samtools was used to process alignment files and removing multimapping reads. Then, reads were split into those aligning to mm10 and hg38 and duplicate reads were removed using the MarkDuplicates.jar utility from the

Picard package available from the Broad Institute (http://broadinstitute.github.io/picard/). ChIP-Rx scale factors were calculated using the method described in (Orlando et al, 2014), of 1/(spike-in aligned counts/1e6), which were utilised in deeptools bamCoverage to generate scaled bigwig files (at 10 bp resolution) for visualisation and for further normalisation in downstream analyses. ChIP-Seq and CUT&RUN reads were processed as above, but aligned to mm10 genome only, and bigwigs generated using a flag --normalizeUsing CPM. Peak calling was performed using macs2 (Zhang et al, 2008) with a q-value cutoff of 0.01. PRC2 target promoters were defined by overlapping WT SUZ12 ChIP-Rx peaks with gene promoter regions ($+/- 2$ kb) as annotated in the mm10 build of the mouse genome using bedtools intersect. Intergenic regions were defined by excluding sites within $+/- 5$ kb from gene bodies, using bedtools intersect with –wa –v flags. DeepTools was further utilised to generate average and tornado plots using the defined peaksets. Boxplots and violin plots were generated from normalised counts calculated with multiBigWigSummary, whose output was imported into Rstudio and visualised using ggplot2. Statistical significance was calculated in R using "wilcox.test" with y = WT values, and alternative = "greater" (for JARID2, SUZ12 and H3K27me3) or "less" (for AEBP2).

## QuantSeq and RNA-Seq library preparation

Total RNA was isolated from D0 Aebp2 WT and KO ESCs, and EpiLC following 48 h (2 days) or differentiation. The quality of extracted RNA was confirmed using the TapeStation (Agilent) with the RNA ScreenTape assay reagents (Agilent; 5067-5576). Total RNA (500 ng) was used/sampled as library preparation input. Libraries were generated using the QuantSeq 3′ mRNA-Seq Library Prep Kit FWD for Illumina (Lexogen; 015.24) in accordance with the manufacturer's instructions. Library DNA was quantified using the Qubit, and size distributions were ascertained on a TapeStation (Agilent) using the D1000 ScreenTape assay reagents (Agilent; 5067–5583). This information was used to calculate pooling ratios for multiplex library sequencing. Pooled libraries were diluted and processed for 75-bp single-end sequencing on an Illumina NextSeq instrument using the NextSeq 500 High Output v2 kit (75 cycles) (Illumina; FC-404–2005) in accordance with the manufacturer's instructions. RNA-Seq library from WT ESCs used in Fig. 3E was prepared and sequenced by Novogene.

## QuantSeq and RNA-Seq analysis

For Quant-seq, reads were trimmed using BBDUK to remove the first 11 nucleotides from reads, polyA tails, TruSeq adaptors, low-quality nucleotides with a $q$ below 10, and reads smaller than 20 nucleotides in length. Reads were aligned to the mm10 genome using the STAR aligner. Gene abundance was quantified using htseq- count and DESeq2 was utilised to identify differentially expressed genes. Gene abundance values were CPM normalised using edgeR and compared using a Wilcoxon rank-sum test. Significantly differentially expressed genes were visualised using Seaborn or pheatmap (Fig. EV5C). RNA-Seq fastqs of WT ESCs were aligned to mm10 using STAR and bigwigs were generated using deepTools bamCoverage as above for visualisation in UCSC Genome Browser.

## Tissue-wide expression analysis of AEBP2 promoters

Data was extracted from pre-processed datasets from CAGE-sequencing data collected and processed by the Fantom5 consortium (https://fantom.gsc.riken.jp/5) for human data (hg19.cage_peak_phase1and2combined_counts_ann.osc.txt.gz) and mouse data (mm9.cage_peak_phase1and2combined_counts_ann.osc.txt.gz) (Kawaji et al, 2017). Data were extracted for AEBP2 using grep and peaks combined based on proximity to annotated promoters. Datasets were grouped into the following tissues groups for human (blood, bladder, lung, testis, neuronal, embryo, muscle, bone, kidney, adipocyte, prostate, placenta, macrophages, ESC, heart, fibroblast, T cells, early embryo and leukaemia) and for mouse (neuronal, placenta, trophoblast, tracheal epithelial, neonate, mesenchymal stem cells, natural helper cells, glial, hepatic, B cells, T cells, macrophage, lung, liver, kidney, intestine, HSC, heart, ESC, muscle, cerebellum and embryo) based on Fantom5 annotation. TPM values for each CAGE peak associated with annotated promoters of AEBP2 were combined and plotted as a boxplot using ggplot2 in R. Promoter annotations and classification, and sample information are provided in Dataset EV2.

## AEBP2 charge and intrinsic disorder prediction

hAEBP2 (Q9Z248-1) disorder plots were generated via https://iupred2a.elte.hu/ using the IUPred2 short disorder settings (Erdős and Dosztányi, 2020; Mészáros et al, 2018). Residue charge calculations were generated via https://www.bioinformatics.nl/cgi-bin/emboss/charge using window length = 5 and the charge heatmap was then generated in Microsoft Excel.

## Analysis of AEBP2 in TetO reporter ESCs

All TetO reporter ESCs were cultivated without feeders in high-glucose-DMEM (Corning 10-013-CV) supplemented with 13.5% foetal bovine serum (Corning 35-015-CV), 10 mM HEPES pH 7.4 (Corning, 25-060-CI), 2 mM GlutaMAX (Gibco, 35050-061), 1 mM Sodium Pyruvate (Corning 25-000-Cl), 1% Penicillin/Streptomycin (Sigma, P0781), 1× non-essential amino acids (Gibco, 11140-050), 50 mM β-mercaptoethanol (Gibco, 21985-023) and recombinant LIF. Cells were incubated at 37 °C and 5% $CO_2$ and were passaged every 48 h by trypsinisation in 0.25% 1× Trypsin-EDTA (Gibco, 25200-056). All flow cytometry analyses were conducted on an Attune NxT Cytometer (Thermo Fisher) and FlowJo software (BD Biosciences).

Twenty million ESCs were dissociated with 0.25% 1× Trypsin-EDTA (Gibco, 25200-056). Single-cell suspension was pelleted by centrifugation with $500 \times g$ for 5 min at room temperature (RT) and washed once with 1× PBS. The cell pellet was incubated with 5 mL cold Swelling Buffer (10 mM Tris pH 7.5, 2 mM $MgCl_2$, 3 mM $CaCl_2$, 1 mM EDTA with freshly added 1× Halt™ Protease Inhibitor Cocktail (Thermo Scientific™, 78430) and rotated at 4 °C for 20 min. Swollen cell suspension was centrifuged at $500 \times g$ for 5 min at 4 °C and supernatant aspirated. To isolate nuclei from swollen cells, the pellet was incubated with 5 mL cold Gro-Lysis Buffer (10 mM Tris pH7.5, 2 mM $MgCl_2$, 3 mM $CaCl_2$, 0.5% Igepal, 10% glycerol, 1 mM EDTA with freshly added 1× Halt™ Protease Inhibitor Cocktail (Thermo Scientific™, 78430) and rotated at 4 °C for 10 min. Isolated

nuclei were pelleted down by centrifuging the cell suspension with $700 \times g$ for 5 min at 4 °C. The nuclei pellet was lysed with 300 µl cold RIPA Buffer (150 mM NaCl, 5 mM EDTA pH 8, 50 mM Tris, pH 8.0, 1% NP-40 (IGEPAL CA-630), 0.5% sodium deoxycholate, 0.1% SDS with freshly added 1 mM PMSF, 1 mM DTT and 1× Halt™ Protease Inhibitor Cocktail (Thermo Scientific™, 78430) and incubated on ice for 10 min. Chromatin in lysed nuclei extract was sheared by sonication in 1.5 mL Bioruptorâ tubes (Diagenode, C30010016) for four cycles (15 s on/30 s off) on a Bioruptorâ Pico sonicator (Diagenode). Nuclear extract was spun down at 20,000 $\times g$ for 10 min at 4 °C, and the supernatant was transferred into 1.5-mL Eppendorfâ tubes (21008-959). Protein concentration was quantified by using the Pierce Detergent Compatible Bradford Assay Kit (Thermo Scientific™, 23246). In all, 4× NuPAGE™ LDS Sample Buffer (Invitrogen™, NP0007) and 2-mercaptoethanol (10% final concentration in 4× LDS Sample Buffer) were added to the nuclear extract and boiled at 95 °C for 5 min. 1× sample reducing agent (Invitrogen™, NP0004) added into samples and were separated on NuPAGE™ 4–12% Bis-Tris gels (Invitrogen™, NP0335BOX) in NuPAGE™ MES SDS running buffer (Invitrogen™, NP0002) with NuPAGE™ antioxidants added (Invitrogen™, NP0005). The gel was transferred on a Merck Chemicals Immobilon-FL Membrane (PVDF 0.45 µm, IPVH15150). After blocking the membranes with 5% non-fat dry milk in 1× TBS and 0.1% Tween-20 for an hour at room temperature, the blots were incubated overnight with the primary antibodies in 5% non-fat dry milk in 1× TBS and 0.1% Tween-20. The list of antibodies used is in the Reagents and Tools table. The blots were washed three times for 15 min with 1× TBS and 0.1% Tween-20 and incubated with corresponding secondary antibodies for 45 min at RT. The blots were washed three times for 15 min with 1× TBS and 0.1% Tween-20 and imaged on an Odyssey CLx Near-Infrared Imaging System (LICOR).

For chromatin immunoprecipitation with quantitative PCR (ChIP-qPCR), $30 \times 10^6$ TetO reporter ESCs were collected, washed once in 1× PBS and crosslinked for 7 min in 1% formaldehyde. The crosslinking was quenched by the addition of 125 mM glycine and incubated on ice. The crosslinked cells were pelleted by centrifugation for 5 min at $1200 \times g$ at 4 °C. Nuclei were prepared by washes with NP-Rinse buffer 1 (10 mM Tris pH 8.0, 10 mM EDTA pH 8.0, 0.5 mM EGTA, 0.25% Triton X-100) followed by NP-Rinse buffer 2 (10 mM Tris pH 8.0, 1 mM EDTA, 0.5 mM EGTA, 200 mM NaCl). Afterwards, the nuclei were washed twice with shearing buffer (1 mM EDTA pH 8.0, 10 mM Tris-HCl pH 8.0, 0.1% SDS) and subsequently resuspended in 900 µL shearing buffer with added 1× Halt™ Protease Inhibitor Cocktail (Thermo Scientific, 78430). Chromatin was sheared by sonication in 15 ml Bioruptor® tubes (Diagenode, C01020031) with 437.5 mg sonication beads (Diagenode, C03070001) for six cycles (1 min on/1 min off) on a Bioruptor® Pico sonicator (Diagenode). For each ChIP reaction, ESC lysate was incubated in 1× IP buffer (50 mM HEPES/KOH pH 7.5, 300 mM NaCl, 1 mM EDTA, 1% Triton X-100, 0.1% DOC, 0.1% SDS) with the respective antibodies at 4 °C overnight on a rotating wheel. A list of antibodies used is provided in the Reagents and Tools table. Antibody-bound chromatin was captured using Dynabeads protein G beads (Thermo Scientific 10004D) for 4 h at 4 °C. Beads were washed 5 times with 1× IP buffer, followed by 3 washes with DOC buffer (10 mM Tris pH 8, 0.25 mM LiCl, 1 mM EDTA, 0.5% NP40, 0.5% DOC) and one wash with TE with 50 mM NaCl. ChIP DNA was eluted twice in elution buffer (1% SDS, 0.1 M

NaHCO$_3$) at 65 °C for 20 min and then RNase A treatment was carried out for 30 min at 37 °C. Proteinase K treatment was carried out for 3 h at 55 °C and crosslinks were reversed overnight at 65 °C. The following day, IP and corresponding input DNA were purified by PCI extraction and DNA precipitation. For ChIP DNA quantification, qPCR was performed using a CFX Connect Real-Time PCR Detection System (Bio-Rad Laboratories). The list of primers is available in Dataset EV1.

## Data availability

Quant-Seq, ChIP-Rx, CUT&RUN and RNA-Seq datasets from this paper are available for download and can be accessed from the Gene Expression Omnibus (GEO) database via accession GSE217538. Coordinates were deposited with the Protein Data Bank with accession numbers PDB 8EQV (Cryo-EM structure of PRC2 in complex with the long isoform of AEBP2). The 3D cryo-EM density map was deposited with the Electron Microscopy Data Bank under the accession number EMD-28547. The mass spectrometry proteomics data have been deposited to the ProteomeXchange Consortium via the PRIDE partner repository (Perez-Riverol et al, 2022) with the dataset identifier PXD053693.

The source data of this paper are collected in the following database record: biostudies:S-SCDT-10_1038-S44318-025-00616-9.

## Peer review information

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

## Acknowledgements

We thank past and present members of the Bracken and Davidovich lab members for helpful discussions and critical reading of the manuscript. We are grateful to the Genomics Core Facility at University College Dublin for expertise and help with next-generation sequencing. The authors acknowledge the use of instruments and assistance at the Monash Ramaciotti Centre for Cryo-Electron Microscopy, a Node of Microscopy Australia. We thank the staff of the Monash Animal Research Platform (MARP) and Animal Research Facility (ARL) at Monash University. We thank Bailey Richardson for helping with CRISPR genetic analysis of reporter gene silencing by TetR-AEBP2 long and short isoforms. EG was in part supported by a grant from Science Foundation Ireland (SFI) under Grant Number 15/IA/3104. Work in the Bracken Lab was supported by Science Foundation Ireland under the SFI Investigators Programme (SFI/16/IA/4562) and the BBSRC-SFI (SFI/17/BBSRC/3415), and an Irish Research Council Advanced Laureate Award (IRCLA/2019/21). NJM is the Isabella and Marcus Foundation Charlee Ferrar Scholar and is also supported through an Australian Government Research Training Program (RTP) Scholarship. CD is an EMBL-Australia Group Leader and acknowledges the support of the Sylvia and Charles Viertel Senior Medical Research Fellowship, the Australian Research Council (grant number DP190103407, FT240100821, DP250104894) and National Health and Medical Research Council (NHMRC; grant numbers APP1162921, APP1184637, APP2011767, APP2020900) and the Victoria State Government through mRNA Victoria. QZ holds an NHMRC Investigator EL1 grant (APP1196365). OB and ZEA were supported by NIH-NIMH (R01MH122565) and Start-up funding from the Norris Comprehensive Cancer Center at Keck School of Medicine of USC. The Vermeulen lab is part of the Oncode institute, which is partly funded by the Dutch Cancer Society (KWF).

## Author contributions

**Marlena Mucha**: Conceptualisation; Data curation; Formal analysis; Validation; Investigation; Methodology; Writing—original draft; Project administration; Writing—review and editing. **Zhihao Lai**: Data curation; Validation; Investigation; Visualisation; Methodology; Project administration; Writing—review and editing. **Nicholas J McKenzie**: Conceptualisation; Data curation; Funding acquisition; Validation; Investigation; Visualisation; Methodology; Writing—original draft. **Francesca Matrà**: Data curation; Validation; Investigation; Visualisation; Methodology; Project administration; Writing—review and editing. **Marion Boudes**: Investigation; Methodology; Project administration. **Sarena F Flanigan**: Investigation. **Maria Teresa Alejo-Vinogradova**: Investigation. **Craig Monger**: Formal analysis. **Qi Zhang**: Investigation. **Darragh Nimmo**: Formal analysis. **Evan Healy**: Formal analysis. **Ademar J Silva**: Investigation. **Daniel Angelov**: Investigation. **David M Reck**: Investigation; Visualisation. **Gráinne Holland**: Investigation. **Zeynep Eda Atmaca**: Investigation; Visualisation. **Helen E King**: Investigation. **Maeve Hamilton**: Investigation. **Eleanor Glancy**: Methodology. **James Nolan**: Investigation. **Robert J Weatheritt**: Investigation. **Oliver Bell**: Resources; Supervision; Funding acquisition; Methodology. **Michiel Vermeulen**: Resources; Supervision; Funding acquisition; Investigation. **Chen Davidovich**: Conceptualisation; Resources; Supervision; Funding acquisition; Methodology; Writing—original draft; Project administration; Writing—review and editing. **Adrian P Bracken**: Conceptualisation; Resources; Supervision; Funding acquisition; Investigation; Methodology; Writing—original draft; Project administration; Writing—review and editing.

Source data underlying figure panels in this paper may have individual authorship assigned. Where available, figure panel/source data authorship is listed in the following database record: biostudies:S-SCDT-10_1038-S44318-025-00616-9.

## Disclosure and competing interests statement

The authors declare no competing interests.

# Expanded View Figures

**Figure EV1.  AEBP2$^L$ inhibits the DNA-binding activity of PRC2 in vitro.**

(**A**) Multiple sequence alignment of human AEBP2 isoforms AEBP2$^{L(Iso1)}$ (Uniprot: Q6ZN18-1), AEBP2$^{L(Iso2)}$ (Uniprot: Q6ZN18-2), and AEBP2$^S$ (Uniprot: Q6ZN18-3), performed using Clustal 0 (1.2.4). (**B**) EMSA performed with 0.7% agarose and used to quantify the affinity of the indicated PRC2 complexes for fluorescein-labelled DNA in Fig. 1C. (**C**) Fluorescence anisotropy assays used to quantify the affinity of the indicated PRC2 complexes for the same fluorescein-labelled DNA probe used for EMSA in Fig. 1C. Three sets of assays were performed using the same binding buffer but with differing levels of KCl in the binding buffer as indicated above each graph (25/50/ 100 mM KCl). Data represent the mean of three independent experiments that were carried out on different days and error bars represent standard deviation.

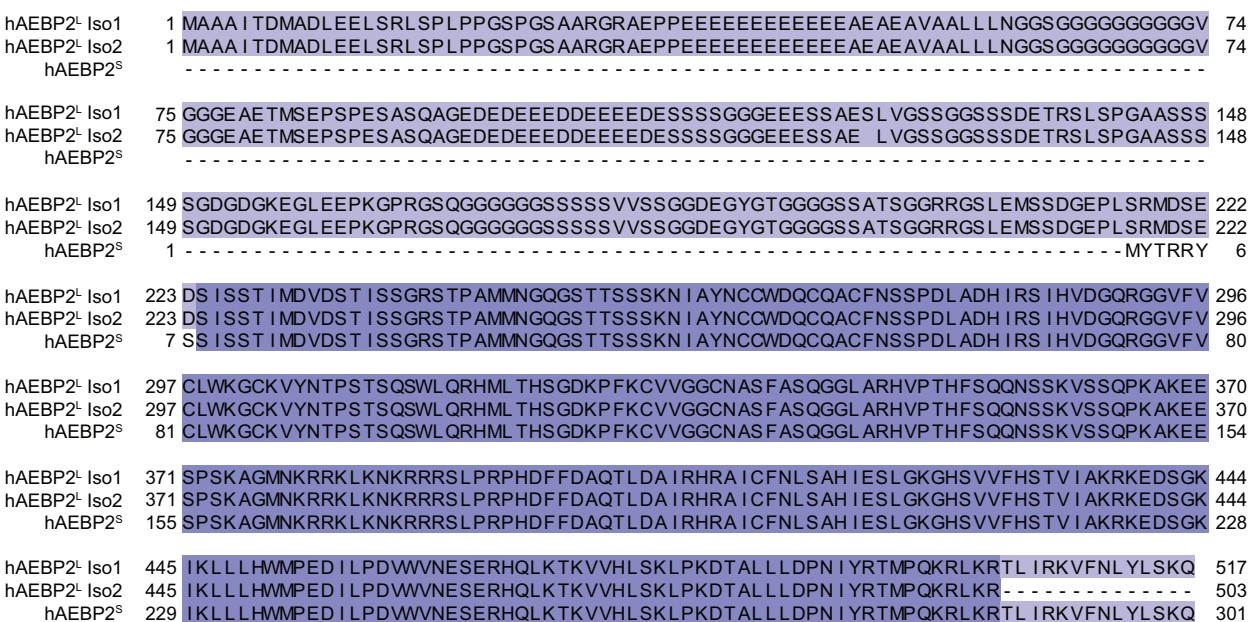

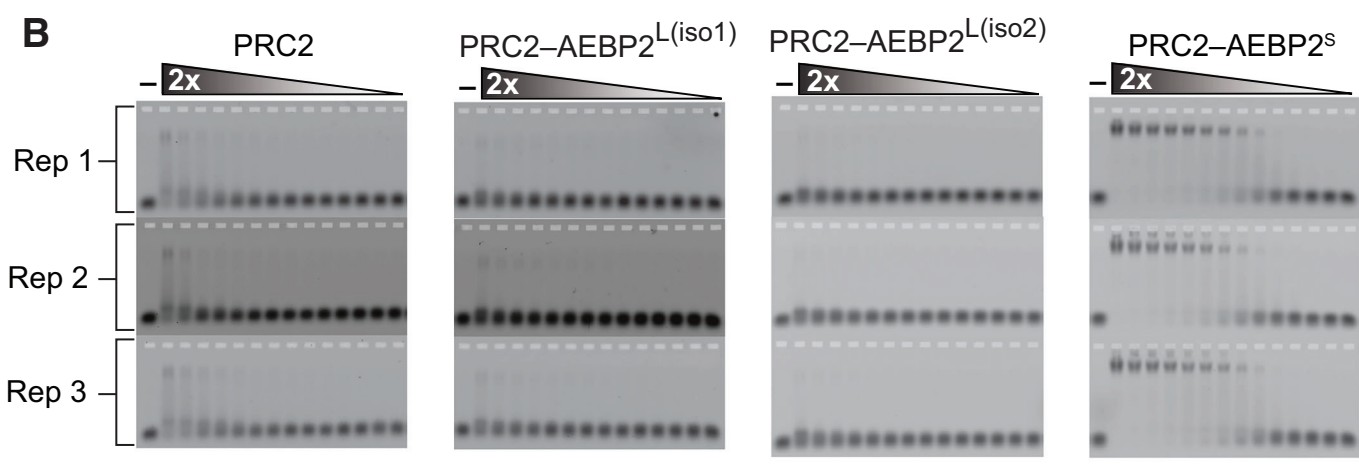

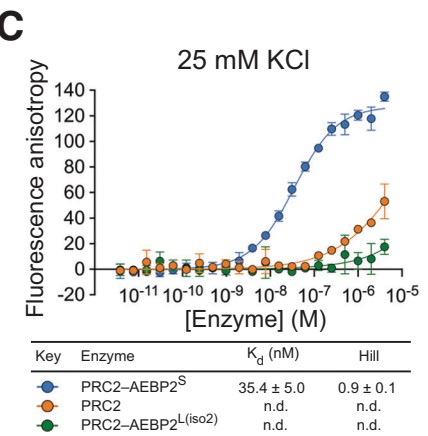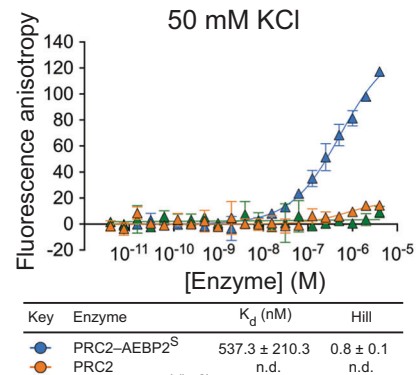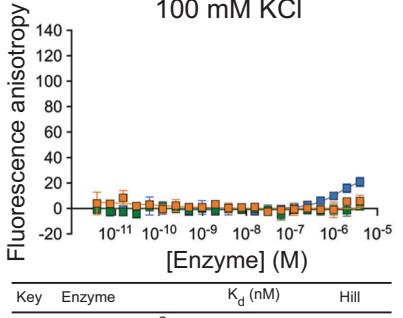

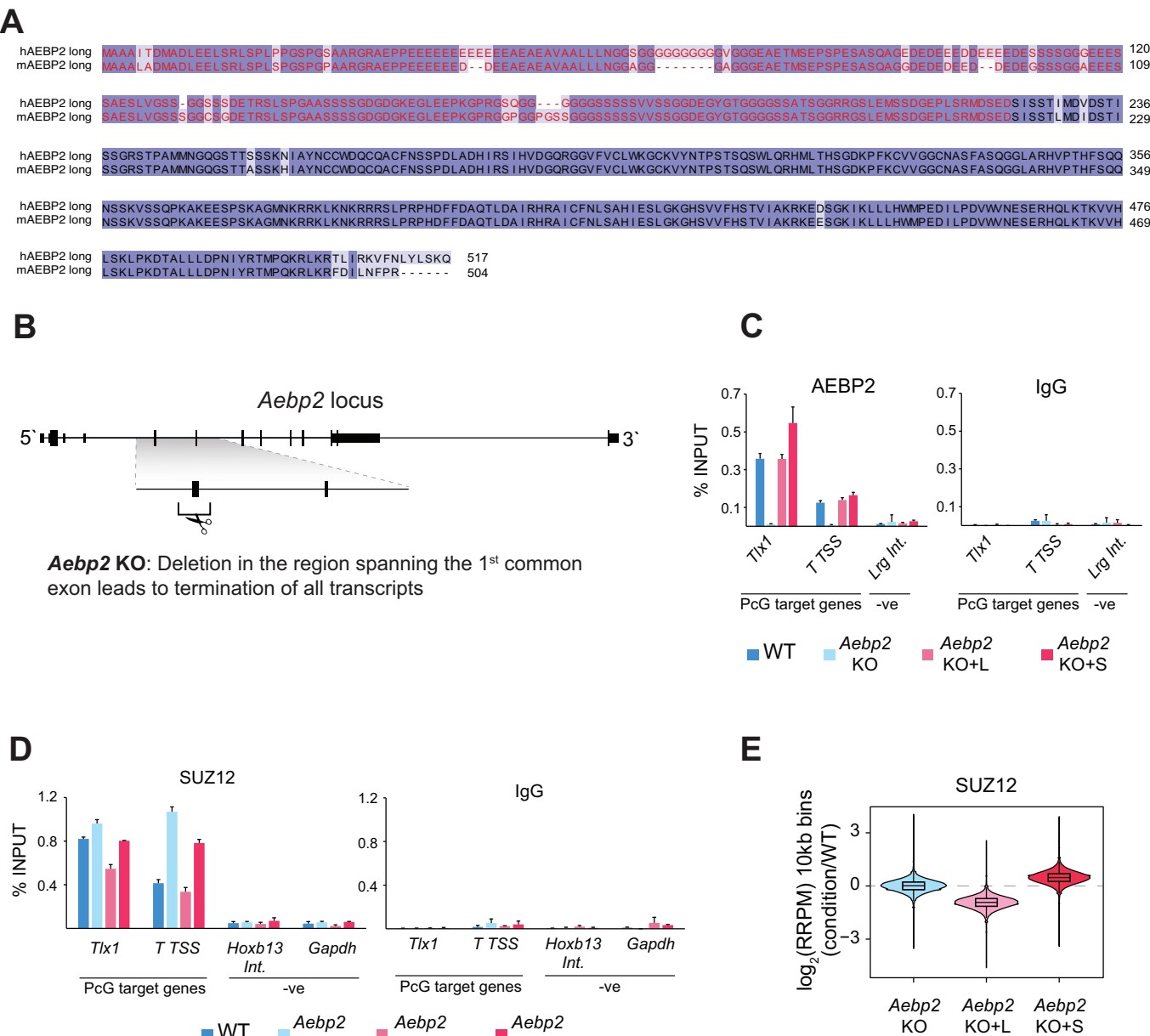

**Figure EV2.   Ectopic expression of AEBP2[L] leads to reduced PRC2 binding to target genes in mouse embryonic stem cells.**

(A) Multiple sequence alignment of human AEBP2 long isoform (Uniprot: Q6ZN18-1) and mouse AEBP2 long isoform (Uniprot: Q9Z248-1), performed using Clustal 0 (1.2.4). (B) Schematic representation of the *Aebp2* gene locus depicting the positions of the sgRNAs used to generate the *Aebp2* KO cell line. See Dataset EV1 for sgRNA sequences. (C) Quantitative chromatin immunoprecipitation (ChIP-qPCR) analyses using the indicated antibodies in *Aebp2* KO, *Aebp2* KO + L and *Aebp2* KO + S ESCs. Each experiment was performed at least three times on separate days. A representative experiment is shown. Error bars and the height of the bar show the standard deviation and the mean of three technical replicates, respectively. (D) Quantitative chromatin immunoprecipitation (ChIP-qPCR) analyses using the indicated antibodies in *Aebp2* KO, *Aebp2* KO + L and *Aebp2* KO + S ESCs. Each experiment was performed at least three times on separate days. A representative experiment is shown. Error bars and the height of the bar show the standard deviation and the mean of three technical replicates, respectively. (E) Violin plots representing the abundance of SUZ12 Rx-normalised reads per million (log_2RRPM) within 10 kb bins genome-wide (272,477 bins). Inside boxes represent the top quartile, the median and the bottom quartile.

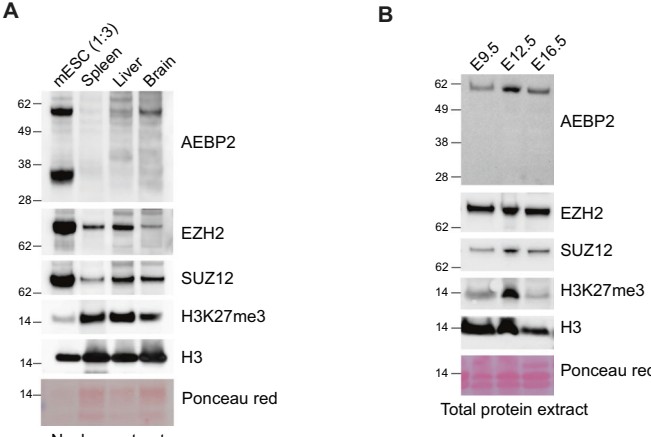

**Figure EV3.  Expression of Aebp2$^L$ and Aebp2$^S$ in adults and during development.**

(A) Western blot analyses of proteins in nuclear extracts of mouse embryonic stem cells (mESCs), spleen, liver and brain, using the indicated antibodies. (B) Western blot analyses of proteins in whole-cell extracts from mice embryos from the indicated developmental stages, using the indicated antibodies.

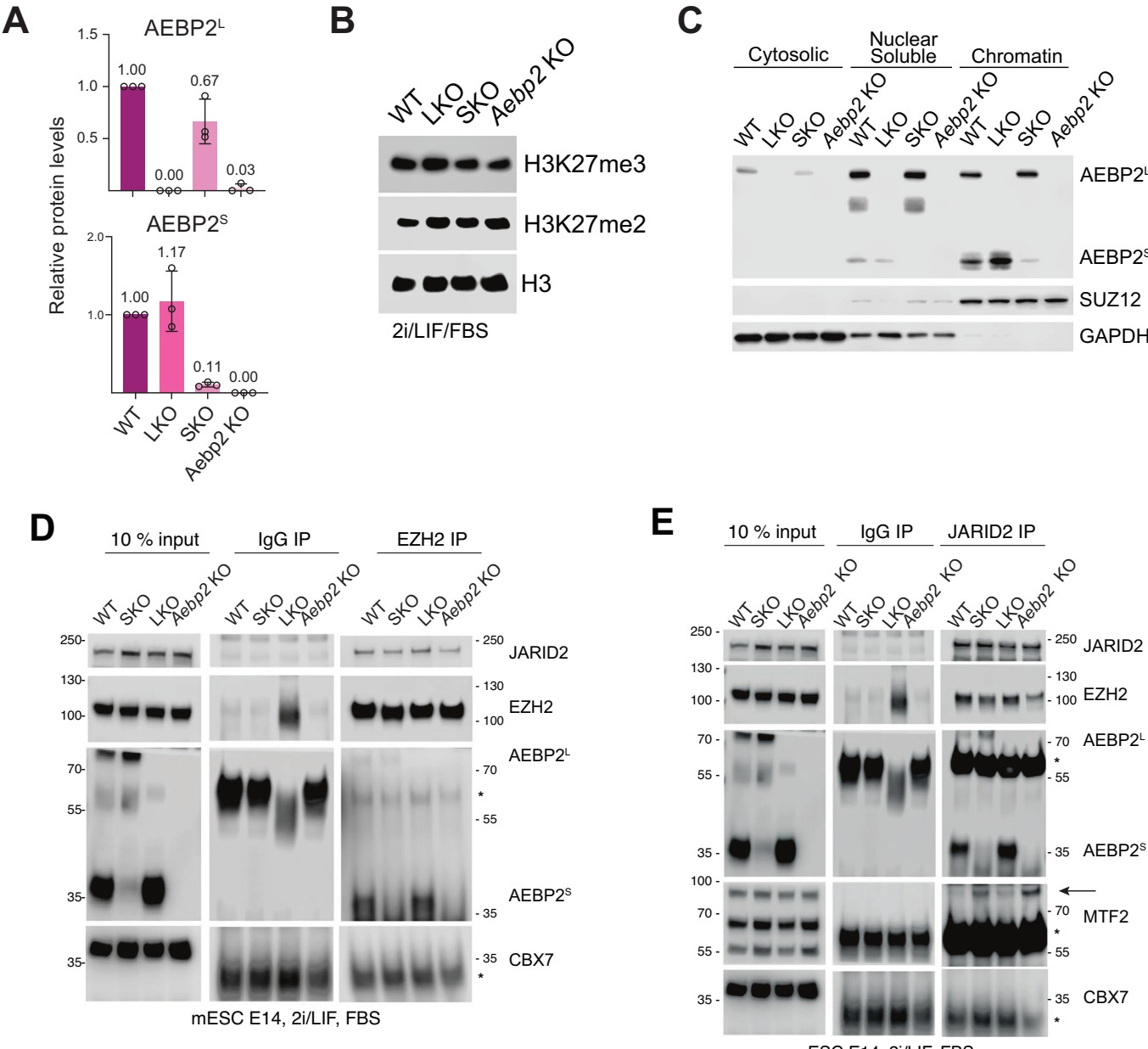

**Figure EV4. Characterisation of the composition of PRC2 in *Aebp2* KO ESC lines.**

(A) Quantification of AEBP2 expression in western blot analysis of whole-cell lysates. The ratio of AEBP2$^L$ and AEBP2$^S$ to normaliser band intensity in WT was set to 1. $n = 3$ biological independent samples. Data are shown as mean ± s.d. (B) Western blot analyses of H3K27me2 and H3K27me3 on whole-cell lysates from the indicated WT, LKO, SKO and *Aebp2* KO ESC lines. (C) Cellular fractionation of LKO, SKO, *Aebp2* KO, and matched WT ESCs, followed by western blot analyses, using the indicated antibodies. (D) Endogenous co-IPs of EZH2 in WT, LKO, SKO and *Aebp2* KO ESC lines, followed by Western blot analysis with the indicated antibodies. The 10% input and IgG IP lanes are identical to the ones from (E). (E) Endogenous co-IPs of JARID2 in WT, LKO, SKO and *Aebp2* KO ESC lines, followed by Western blot analysis with the indicated antibodies. The EZH2 and JARID2 co-IPs in (D, E) were simultaneously performed as part of one experiment. The 10% input and IgG IP lanes are identical to the ones from (D).

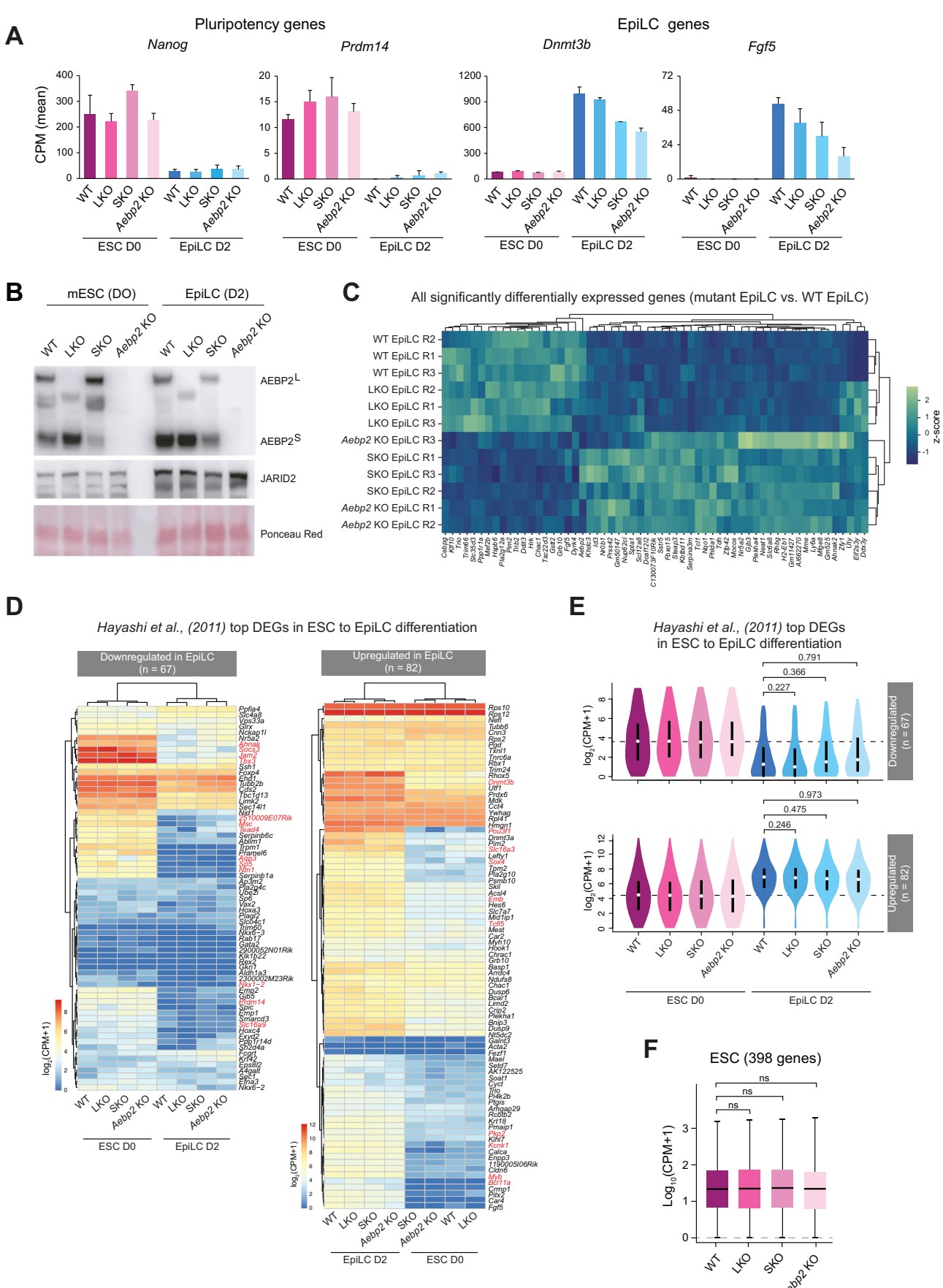

◀  **Figure EV5.  Consequences of loss of AEBP2$^S$ or AEBP2$^L$ on EpiLC differentiation.**

(A) Bar plots showing the mean and standard deviation of relative expression of two representative pluripotency genes (*Nanog* and *Prdm14*) and two representative genes highly expressed at the EpiLC stage (*Fgf5* and *Dnmt3b*) at Day 0 and Day 2 timepoints, measured by Quant-Seq (CPM), based on three independent biological replicates performed on different days. (B) Western blot analyses using the indicated antibodies on whole-cell lysates from WT, LKO, SKO and *Aebp2* KO ESC lines, either before (mESCs) or after differentiation to EpiLC. (C) Heatmap showing unsupervised hierarchical clustering of the mRNA levels of significantly differentially expressed genes (DEG) at the EpiLC differentiation stage, between LKO vs. WT, SKO vs. WT, and *Aebp2* KO vs. WT, as measured by Quant-Seq. Scale: z-score. The gene lists can be found in Dataset EV4. (D) Heatmaps showing unsupervised hierarchical clustering of the mRNA levels of genes previously shown to be significantly differentially expressed during EpiLC differentiation (Hayashi et al, 2011), in WT, LKO, SKO and *Aebp2* KO cell lines at the two stages of differentiation, as measured by Quant-Seq. Data shown represent log$_2$(mean CPM + 1) values of three independent biological triplicates. (E) Violin plots showing the mRNA levels of the genes from panel d. Shown is the mean of three independent biological triplicates. Wilcoxon test was used to measure statistical significance, with *P* values shown. The inside boxplots represent the interquartile range (Q1 to Q3), with a median indicated by the white dot. (F) Boxplots representing the mRNA levels of the 398 PRC2 de novo recruited genes at the ESC stage, measured by Quant-Seq. Shown is the mean of three independent biological replicates. Wilcoxon test was used to measure statistical significance. ns = not significant. The boxplots represent the interquartile range (Q1 to Q3), with a median indicated by the thick line, and the minimum and maximum values indicated by the whiskers.

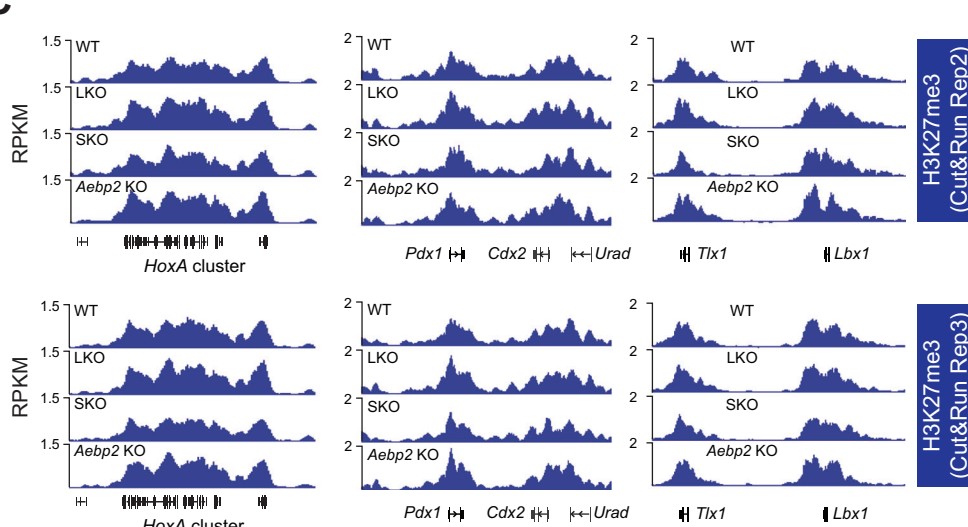

A

AEBP2 R2

SUZ12 R2

JARID2 R2

AEBP2 R3

SUZ12 R3

JARID2 R3

■ WT
■ LKO
■ SKO
■ *Aebp2* KO

B

H3K27me3 (Cut&Run Rep2)

WT    LKO    SKO    *Aebp2* KO

H3K27me3 (Cut&Run Rep3)

WT    LKO    SKO    *Aebp2* KO

WT SUZ12 ChIP-Rx peaks (n=6,787)

-10kb   10kb

-10kb   10kb

C

HoxA cluster

Pdx1   Cdx2   ←←Urad

Tlx1   Lbx1

H3K27me3 (Cut&Run Rep2)

HoxA cluster

Pdx1   Cdx2   ←←Urad

Tlx1   Lbx1

H3K27me3 (Cut&Run Rep3)

**Figure EV6.  Loss of AEBP2$^L$ but not AEBP2$^S$ leads to increased PRC2 and H3K27me3 on Polycomb target genes in mouse ESCs.**

(A) Quantitative chromatin immunoprecipitation (ChIP) analyses using the indicated antibodies in WT, LKO, SKO and *Aebp2* KO ESCs. Two biological replicates are shown, independent of the ChIP-Rx replicate from Fig. 4. Error bars are representative of technical triplicates. (B) Average plot and heatmap representations of H3K27me3 CUT&RUN Replicates 2 and 3 RPKM values at WT SUZ12 peaks ($n = 6787$) in WT, LKO, SKO and *Aebp2* KO ESCs. Plots are centred on region midpoint $+-10$ kb. Relative intensities are indicated. (C) UCSC genome browser representations of H3K27me3 CUT&RUN Replicates 2 and 3 RPKM values at three representative PcG target loci (*HoxA, Pdx1/Cdx2, and Tlx1/Lbx1*) in WT, LKO, SKO and *Aebp2* KO cell lines.

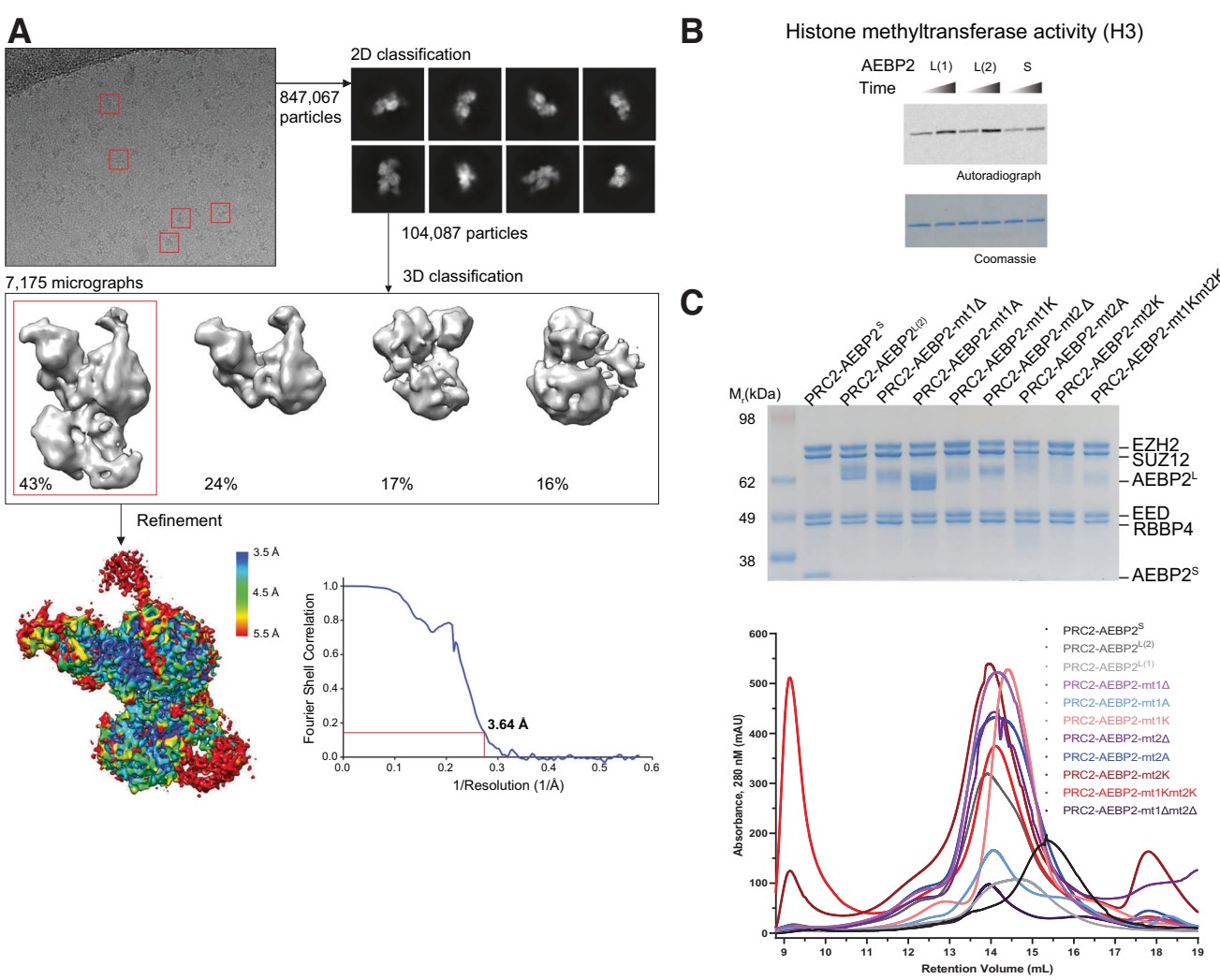

**Figure EV7.  The N-terminal region of AEBP2$^L$ is disordered and utilises mammalian-specific acidic tracts to inhibit PRC2.**

(A) Cryo-EM processing workflow for PRC2–AEBP2$^L$. 847,067 particles were picked and sorted using two-dimensional classification. The resulting 104,087 particles were subjected to ab initio three-dimensional classification. This initial classification yielded one class with 45,024 particles that clearly corresponds to an intact complex. This subset was subjected to three-dimensional non-uniform refinement followed by global and local CTF refinement, yielding a 3.64 Å resolution cryo-EM map (gold-standard FSC = 0.143 criterion). Subsequent background subtraction and masked refinements of the individual domains and regions did not lead to map improvements. (B) A representative Coomassie blue-stained SDS–PAGE and the corresponding radiogram of the HMTase assay of AEBP2$^{L/S}$ with a H3.1-only substrate. (C) Coomassie blue-stained SDS–PAGE gel showing the purity of PRC2–AEBP2 complexes used for Fig. 5G and (B), and well as representative gel filtration chromatograms (Superose 6 Increase 10/300 GL) of proteins used for Fig. 5, and this figure.

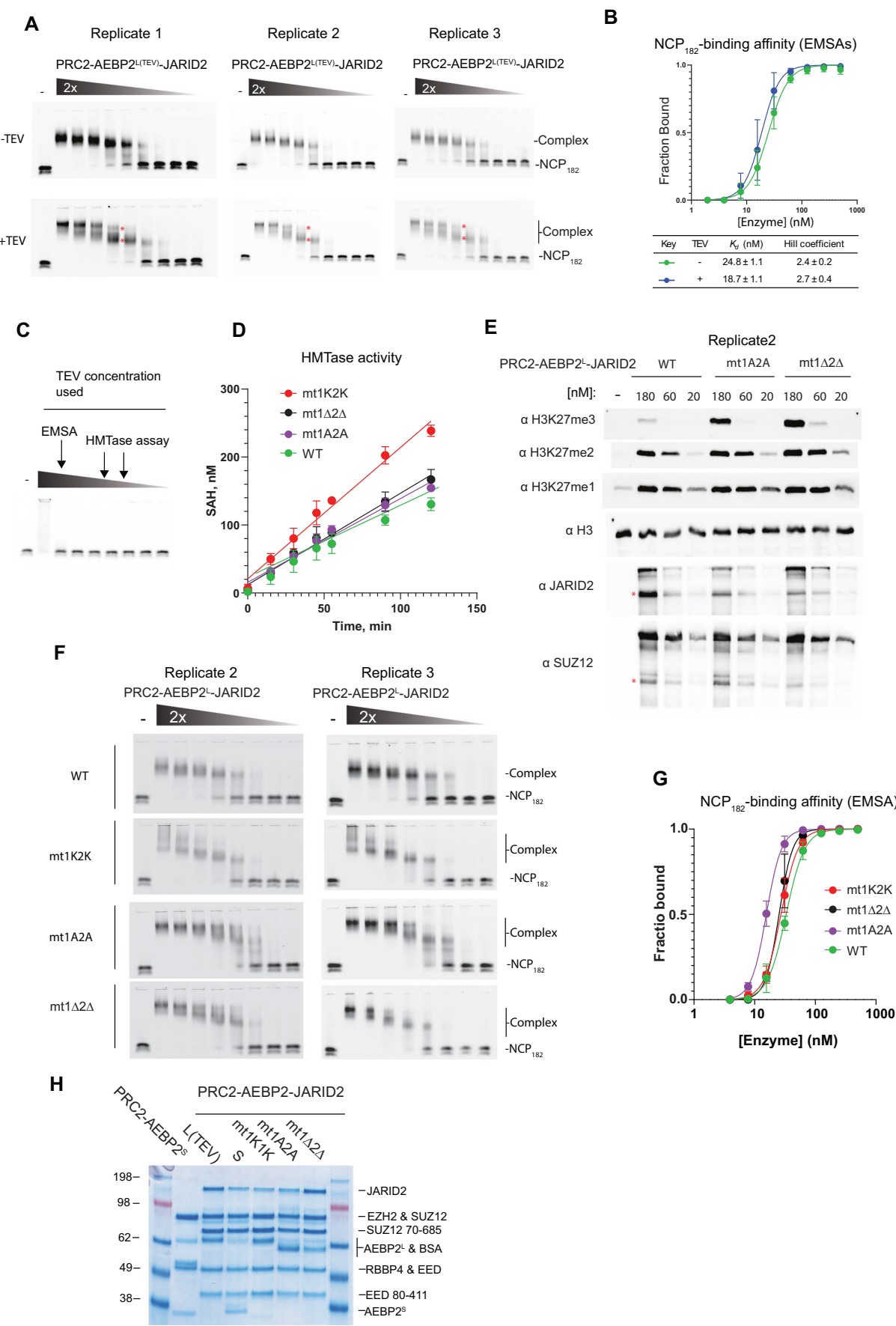

◄ **Figure EV8.  AEBP2^L antagonises PRC2.2.**

(**A**) PRC2-AEBP2^{L(TEV)}-JARID2 was treated in the presence or absence of TEV protease before were subjected to EMSA with 5 nM Cy5-NCP_{182} probe. The protein was subjected to a twofold serial dilution starting from 500 nM. Red asterisks indicate multiple bound complexes that are migrated with different velocities. (**B**) Top: binding curves of the EMSA from (**A**). Bottom: The derived $K_d$ and Hill coefficients. Means and error bars represent the average SAH concentration and the standard error, respectively, based on three independent replicates performed on 3 different days. (**C**) To identify a TEV concentration that would not interfere with a binding assay, EMSA was carried out using 5 nM Cy5-NCP_{182} probe and twofold serially diluted TEV protease, starting from 0.4 mg/ml. "TEV concentration used" indicated the TEV concentration that was selected for EMSA experiments (0.2 mg/ml) in (**A**) and final TEV concentrations in the HMTase reactions in Fig. 6C that were carried out in the presence of 50 nM and 20 nM PRC2-AEBP2^{L(TEV)}-JARID2 (TEV concentration of 0.02 mg/ml and 0.008 mg/ml, respectively). (**D**) Progress curves were carried out under the same conditions used for the Michaelis–Menten kinetic analysis in Fig. 6F, except that 1 μM chromatin (NCP equivalent) and 25 nM PRC2-AEBP2^L-JARID2 complexes as indicated were used. The reaction was stopped at 15 min, 20 min, 45 min, 55 min, 90 min and 120 min before the produced SAH was quantified. Mean are derived from two independent experiments that were carried out on different days and the error bars represent standard deviation. A linear regression was used to fit a linear line across the means. (**E**) Second replicate of the immunoblotting from Fig. 6F, using antibodies as indicated. Red asterisks indicate degradation products of SUZ12 and JARID2. (**F**) Second and third replicates of the EMSA shown in Fig. 6G. (**G**) Binding curves of different PRC2-AEBP2^L-JARID2 complexes, as indicated, and NCP_{182} nucleosomal probe. Means represent the fraction of the bound probe as quantified from the three independent EMSA experiments presented in Fig. 6G and in (**F**), and the error bars represent standard deviation. (**H**) SDS–PAGE analysis of the sample used for fluorescence anisotropy in Fig. 6J. 8 μL of the binding reaction from the highest protein concentration (500 nM) was mixed with 4 μL of 4X LDS buffer and subjected to SDS–PAGE. "S" indicates a TEV-free PRC2-AEBP2^S-JARID2, generated by the TEV cleavage of PRC2-AEBP2^{L(TEV)}-JARID2 followed by the removal of the TEV enzyme using size exclusion chromatography.

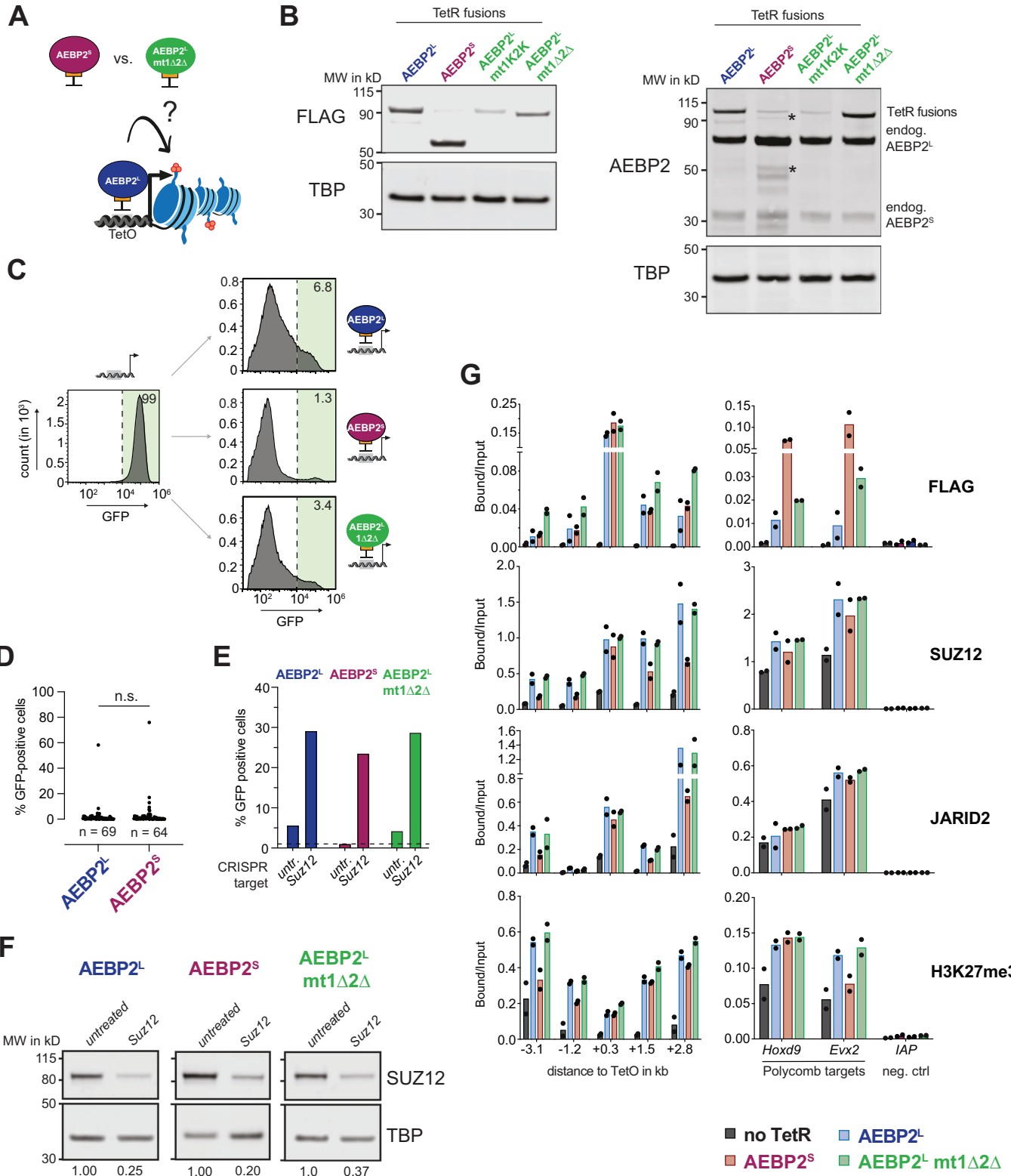

◄ **Figure EV9.  The antagonistic effect of AEBP2$^L$ can be rescued by heterologous tethering to chromatin.**

(A) Scheme of experimental design. TetR fusions facilitate forced recruitment of wild-type and mutant AEBP2 isoforms to Tet Operator sites (TetO) upstream of a reporter gene to compare the impact on chromatin modifications and gene expression. (B) Immunoblots show expression levels of TetR fusion proteins relative to endogenous AEBP2 isoforms. TBP serves as loading controls. Asterisks indicate residual uncleaved mCherry-P2A-TetR fusion proteins (C) Flow cytometry histograms of GFP expression in the absence (left) and presence of TetR fusions with AEBP2 isoforms (right). Inserted numbers indicate the percentage of GFP-positive reporter ESCs. (D) Quantification of percentage of GFP-positive cells in clonal reporter ESC lines in expressing AEBP2$^L$ or AEBP2$^S$. (E) Bar plot shows the percentage of GFP-positive TetR-AEBP2$^L$ or TetR-AEBP2$^S$ reporter ESCs following transduction with Cas9/sgRNAs targeting *Suz12*. Untransduced reporter ESCs serve as controls ("untr."). (F) Immunoblotting of the mESC reporter cell lines that were assayed in (E) using antibodies as indicated. (G) ChIP-qPCR analysis shows relative enrichments of FLAG-TetR fusions, Polycomb proteins and histone modifications upstream and downstream of TetO (left) and at endogenous loci (right) in reporter ESCs. Shown are data of two independent experimental replicates.

