## [Peer Review File · The EMBO Journal]

Auto-inhibition of PRC2 by the broadly expressed long isoform of AEBP2

Marlena Mucha, Zhihao Lai, Nicholas McKenzie, Francesca Matra, Marion Boudes, Sarena Flanigan, Maria Alejo Vinogradova, Craig Monger, Qi Zhang, Darragh Nimmo, Evan Healy, Ademar Silva, Daniel Angelov, David Reck, Gráinne Holland, Zeynep Atmaca, Helen King, Maeve Hamilton, Eleanor Glancy, James Nolan, Robert Weatheritt, Oliver Bell, Michiel Vermeulen, Chen Davidovich, and Adrian Bracken

Corresponding author(s): Adrian Bracken (adrian.bracken@tcd.ie) , Chen Davidovich (chen.davidovich@monash.edu)

Review Timeline:

Submission Date:	25th Jun 25
Editorial Decision:	12th Aug 25
Revision Received:	2nd Sep 25
Accepted:	1st Oct 25

Editor: Cornelius Schneider

Transaction Report:

This manuscript was transferred to The EMBO JOURNAL following peer review at another journal.

We thank the reviewers for dedicating their time to review our manuscript. We believe that their comments allowed us to improve it substantially. This includes two new main figures: The new Figure 4 demonstrates that CRISPR-mediated loss of AEBP2^L, but not loss of AEBP2^S, causes increased PRC2 binding and H3K27me3 deposition on Polycomb target genes. The new Figure 6 demonstrates that the N-terminal region of AEBP2^L antagonizes PRC2 binding to nucleosomes and HMTase activity, also in the presence of JARID2. Together, with several other additions and improvements, our revised manuscript characterizes the opposite roles of the short and long isoforms of AEBP2. We show how the AEBP2^S isoform is restricted to very early development and germ cells and has a Polycomb-related role in the *de novo* repression of stem cell genes during the differentiation of pluripotent cells. In contrast, we show that AEBP2^L is present in most cell types during development and adulthood and antagonises H3K27me3 deposition by interfering with PRC2-chromatin interactions.

Reviewers' Comments:

Reviewer #1:

Remarks to the Author:

Mucha and colleagues have explored the role of the short and long isoforms of AEBP2 in the regulation of PRC2. They first reconstituted both complexes in vitro (PRC2-AEBP2^L and PRC2-AEBP2^S) and analyzed their properties. Then, they expressed each isoform in ES cells AEBP2-KO and performed ChIP-seq for SUZ12 and AEBP2. They conclude from those experiments that in contrast to the short isoform, AEBP2-long does not have the ability to bind DNA and could have an inhibitory effect on PRC2 by limiting its tethering to chromatin. Then, instead of overexpressing the isoforms in a KO context, they selectively targeted the promoter of each isoform to maintain the other isoform controlled by its endogenous promoter and analyzed ES differentiation into EpiLC. The authors propose that AEBP2-short is implicated in de-novo gene targeting and transcriptional repression during early development while AEBP2-long has another function in adult tissues. They finally argue with structure prediction and Cryo-EM structure of PRC2/AEBP2-long that the N-terminal part which is specific to AEBP2-long is highly

flexible. The authors further hypothesize that two acidic tracts within this N-terminal region are key to the negative regulation of PRC2 and test this hypothesis with a mutagenesis in-vitro.

Investigating the role of AEBP2 isoforms is an interesting effort and the authors have all the tools to carefully tackle this question however the current version of the study appears very preliminary. Consequently, most experiments that are key to support the main conclusions lack crucial controls and/or necessary replicates. Besides, we do not think that the presented data support a model in which the acidic tracts are necessary and sufficient for PRC2 inhibition.

For these reasons, major revisions are required to be considered for publication.

Please find below a list of our comments and suggestions for experiments:

Figure 1:

As AEBP2 is part of a complex including JARID2 in cells, the EMSA and HMT assays presented in figure 1 need to be performed in presence of JARID2. Without this information, one cannot make any conclusion regarding the relevance of the described enzymatic assay in vivo.

Also, in panel (e) it is unexpected that AEBP2-S does not appreciably enhance the activity of PRC2 in this assay. Do the authors have an explanation? See for instance Lee et al., 2018 (Mol. Cell) where Aebp2 has a substantial effect.

We thank this reviewer for the important suggestions - we now performed HMTase assays and nucleosome binding assays using a PRC2-AEBP2-JARID2 complexes (new Fig. 6 and new Extended Data Fig. 8).

In Fig 6a-b (pasted below), we demonstrated that we developed a new system for the study of PRC2 and the full-length JARID2 together with AEBP2^L and AEBP2^S. Briefly, we found it very challenging to co-purify recombinant PRC2 together with the short isoform of AEBP2 and the full-length JARID2. To our knowledge, such a recombinant complex was never produced in previous studies. Since we show that AEBP2^S immunoprecipitated with JARID2 from cells (Extended Data Fig. 4e), we believe that our difficulty co-purifying

PRC2 together with JARID2 and AEBP2^S is simply a technical issue, likely related to the positioning of tag on the shorter AEBP2 isoform within the context of the holo-complex. To solve this technical problem, we introduced a TEV site downstream of the N-terminal region of AEBP2^L, to produce a long isoform of AEBP2 (AEBP2^{L(TEV)}) that can be converted into a short AEBP2 (AEBP2^S) isoform by a TEV cleavage. Such TEV cleavage can take place after the PRC2-AEBP2^{L(TEV)}-JARID2 complex has been purified, thereby converting it to PRC2-AEBP2^S-JARID2:

In Fig. 6c, we demonstrated that the HMTase activity of PRC2-AEBP2^{L(TEV)}-JARID2 on chromatinized substrate is substantially increased after the N-terminal region of AEBP2^L is removed by TEV cleavage:

To further support the statement that the N-terminal region of AEBP2^L antagonises the interactions of PRC2 with chromatin, we now show that the inhibition of its HMTase activity only occurs against a chromatinized substrate (above), but not against histone tail peptide substrate (below, Fig. 6d):

Furthermore, we also performed HMTase assays using PRC2-AEBP2^L-JARID2, with wild-type and mutant AEBP2^L, demonstrating that mutations in the acidic tracts of the N-terminal region of AEBP2^L increase the HMTase activity of PRC2-AEBP2^L-JARID2 (Fig. 6e-f):

We also performed EMSA experiments with all the PRC2-AEBP2-JARID2 protein complexes that were assayed above (Fig. 6h-i and Extended Data Fig. 8a,b,f,g). We find that PRC2-AEBP2^L-JARID2 exhibits a modest reduction in the affinity for nucleosomes, with respect to PRC2-AEBP2^S-JARID2 and PRC2-AEBP2^L-JARID2 containing N-terminal mutations in AEBP2^L (Fig. 6i and Extended Data Fig. 8b). When interpreting this data, one should bear in mind that PRC2.2 has multiple nucleosome binding sites, so even if one site is blocked by AEBP2^L, the probe may still bind to another site during an EMSA experiment (more on this below, in response to the last point made by this reviewer). Accordingly, the modest increment of the affinity of PRC2-AEBP2^S-JARID2 to the nucleosome probe is accompanied by the appearance of a prominent additional band (relevant bands are marked in red asterisks in Extended Data Fig. 8b):

These data, together with the new HMTase assays, which form the new Fig. 6, indicate that the acidic tracts of AEBP2^L reduce the HMTase activity of PRC2.2 on chromatin substrate in a mechanism that involves reduced affinity to chromatin or altered binding interactions. This new Fig. 6 is discussed in a new section of the Results, entitled “AEBP2^L antagonises PRC2.2”.

Regarding the HMTase experiments in Lee *et al.*, 2018, *Mol. Cell*¹: It is impossible to directly compare the HMTase activities that Lee *et al* reported to the HMTase activities here. This is because of the differences between the HMTase buffers that were used. Specifically, Lee *et al* carried out HMTase assays in a buffer that did not include a monovalent salt (their HMT buffer was: 50 mM Tris-HCl, pH 8.5, 5 mM MgCl₂, and 4 mM DTT). Whereas, we performed our HMTase assays in a buffer that included a near-physiological concentration of monovalent salt: 100 mM KCl (HMTase buffer that was used here: 50 mM Tris-HCl pH 8.15 at 25°C, 100 mM KCl, 0.5 mM MgCl₂, 0.1% Tween-20 and 5 mM DTT). This is a very important difference between Lee *et al* and our study, as we recently demonstrated that the monovalent salt concentration is extremely important in HMTase assays of PRC2 when nucleosomal substrates are used. For instance, in Fig. 1f from Gail *et al* *Nature Genetics*, 2024², one can see that the different concentrations of KCl (0, 50 and 100 mM) have almost no effect on the HMTase activity of PRC2 for H3 substrate, but yet KCl strongly inhibits the activity of PRC2 for a chromatinized substrate:

Fig. 1f from Gail *et al* Nature Genetics, 2024².

One of our labs (C.D.) also showed that the monovalent salt concentration has a strong effect on the PRC2-PALI1 complex, whereas KCl has the opposite effect on the HMTase activity for a chromatinized substrate. Specifically, near-physiological monovalent salt concentration (100 mM KCl) increases the activity of PRC2-PALI1, while low monovalent salt concentration (30 mM KCl) reduces the activity of PRC2-PALI1 (from Fig. 7c in Zhang *et al* 2021, Nature Comms³):

Hence, the huge difference between the concentration of the monovalent salt used by us here (100 mM KCl) and by Lee *et al* (0 mM KCl/NaCl) is likely contributing to variations in observations between these works. We believe that our experimental system more closely resembles physiological monovalent salt concentration, which is commonly considered as 100-150 mM KCl (reviewed in⁴⁻⁶).

Interestingly, regarding the relevance of the described enzymatic assays without JARID2, while we have now performed the relevant experiments both in the presence (Fig. 6) and absence of JARID2 (Fig. 1), we know that AEBP2-PRC2.2 can exist in cells that do not express full-length JARID2. The full-length form of JARID2 that can bind to PRC2 is not

present in several tissue types, including lymphocytes and keratinocytes⁷. Supporting this, in two independent studies from one of our labs (A.P.B.), we were unable to detect JARID2 protein peptides in mass spectrometry of either endogenous immunoprecipitations of SUZ12 or AEBP2 in the WSU-DLCL2 Diffuse large B-cell lymphoma cell line (Reviewer Fig. 1, shown below) or in endogenous immunoprecipitations of EED in human diploid fibroblasts (Reviewer Fig. 2, shown below). While these data are not included in this revised manuscript, we explain the relevance of studying AEBP2-PRC2 without JARID2 when describing Figure 1.

Specifically, we modified the text of the results section related to Figure 1 to cite the Al-Raawi *et al.*, 2018 paper⁷ and highlight the fact that AEBP2 can exist within PRC2.2 in all cells that do not express full-length JARID2.

Reviewer Fig. 1. Unpublished IP-MS data from a separate study showing JARID2 is not present in immunoprecipitation experiments of either endogenous SUZ12 or AEBP2 in the WSU-DLCL2 Diffuse large B-cell lymphoma cell line.

Reviewer Fig. 2. Unpublished IP-MS data from an additional separate study showing JARID2 only very weakly immunoprecipitates with EED in a human diploid fibroblast (HDF) cell line.

Figure 2

The ChIP-RX data presented Figure 2 lack several conditions to support the author's model.

- 1) H3K27me3 ChIP-seq needs to be included. Since the authors' hypothesis is that AEBP2-long inhibits PRC2 activity, the relevant readout is H3K27me3.
- 2) All experiments should include the WT control. This is particularly important considering the figure S3G middle panel showing that deletion of either AEBP2L or AEBP2S impairs SUZ12 recruitment to a similar extent to the knockout of both isoforms.

Here, we firstly address Point 2. We added the WT control to all relevant panels in the updated Figure 2 and Extended Data Figure 2, as requested. Importantly, the Western blot analyses revealed that the AEBP2 isoforms are 2-4 fold overexpressed compared to the levels of endogenous AEBP2^S and AEBP2^L in the matched parental control E14 ESCs (Fig. 2b):

In terms of the ChIP-Rx analyses in Figure 2, as shown below, we also modified these panels to now include the WT parental ES controls throughout (Fig. 2c-f):

Importantly please note, these exogenous expression experiments had levels of AEBP2^S and AEBP2^L at about 3-4 fold greater than their endogenous levels, which we felt could lead to artifacts in assays of H3K27me3 and transcription. Therefore, we address this by studying the consequences of loss of the endogenous AEBP2 isoforms in CRISPR engineered ESC lines in the updated Figures 3 and 4.

Nonetheless, Figure 2 does establish an important key difference between AEBP2^S and AEBP2^L, which is that AEBP2^S is better than AEBP2^L at binding to chromatin in cells (Fig. 2c-d). This experiment also establishes that ectopic overexpression of AEBP2^L causes reduced SUZ12 binding, representative of core PRC2, on Polycomb target genes (Fig. 2e-f). This supports our *in vitro* observations in Figure 1 in which we establish that AEBP2^S-PRC2 binds to nucleosomes better than AEBP2^L-PRC2. Taken together, we conclude in the revised paper that AEBP2^L competes with other accessory proteins, including MTF2 and AEBP2^S, for binding to a limited amount of core PRC2, and consequently, ectopic overexpression of AEBP2^L leads to an overall reduced binding of PRC2 to Polycomb target genes.

In terms of the Reviewer's Point 1, as mentioned, the above experiment in Figure 2, while informative for comparing the relative abilities of ectopic forms of the AEBP2^S and

AEBP2^L isoforms to bind to chromatin and Polycomb target genes in cells, is limited in its ability to inform on the biology of either AEBP2 isoform. Therefore, we instead focused all our subsequent analyses, including ChIP/Cut&Run of PRC2 components and H3K27me3 (new Fig. 4 and Extended Data Fig. 6), RNA-seq and proteomics (Fig. 3 and Extended Data Fig. 4-5) on our CRISPR engineered ESC lines. These lines have either knockout of the endogenous AEBP2^S isoform (SKO), the AEBP2^L isoform (LKO) or both isoforms (*Aebp2* KO). While these various experiments are described in more detail below, the take home message is that loss of AEBP2^L, but not loss of AEBP2^S, leads to increased PRC2 binding and H3K27me3 deposition on Polycomb target genes (see new Fig. 4 and Extended data Fig. 6):

New Fig. 4. Loss of AEBP2^L but not AEBP2^S leads to increased PRC2 and H3K27me3 on Polycomb target genes in mouse ESCs.

3) The appropriate models for those experiments are the cell lines described in figure 3 instead of the rescue by overexpression.

We entirely agree and thank the reviewer for highlighting this. As mentioned above, we performed all subsequent ChIP-Rx experiments for SUZ12, AEBP2 and JARID2 and CUT&RUN for H3K27me3 (new Fig. 4 – shown above - and Extended Data Fig. 6), as

well as endogenous IP-MS proteomic analyses (updated Fig. 3) in the isogenic parental (WT) and CRISPR engineered KO of AEBP2^S (SKO) or AEBP2^L (LKO) isoforms, as well as the full *Aebp2* KO ESC line. These latter IP-MS analyses are discussed in more detail below.

4) In order to enable appropriate interpretation of the data, the ChIP-seq for H3K27me3 should be completed by western blot showing the global levels of H3K27me3 & me2 in each cell lines.

To address this, we performed H3K27me3 and H3K27me2 Western blot analysis of the CRISPR KO ESC lines and their matched wild type in Extended Data Fig. 4b, as shown below:

Fig. 3g

Extended Data Fig. 4b

This revealed no significant changes in global levels of H3K27me3 or H3K27me2 upon loss of either one or both AEBP2 isoforms (Fig. 3g and Extended Data Fig. 4b). This result is consistent with previous reports in which loss of as many as six PRC2 accessory proteins (AEBP2, JARID2, PHF1, MTF2, PHF19, and EPOP) did not affect the global levels of H3K27me2/3⁸.

5) More details should be provided in the material and methods regarding the bioinformatics pipeline (i.e. parameters for peak calling?, Spike-in correction for each sample). More importantly, duplicates are required for ChIP-seq.

We thank the Reviewer for highlighting this. We updated the materials and methods section regarding the bioinformatics pipeline used for the spike-in ChIP-Rx analysis. It now includes these extended details: “*Reads were aligned to a metagenome of mm10*

and hg38 using bowtie2, with “_hg38” appended to chromosome names prior to combining and indexing. Samtools was used to process alignment files removing multimapping reads. Then, reads were split into those aligning to mm10 and hg38 and duplicate reads were removed using the MarkDuplicates.jar utility from the Picard package available from the Broad Institute (<http://broadinstitute.github.io/picard/>). ChIP-Rx scale factors were calculated using the method described in Orlando et al., 2014⁹, of $1/(\text{spike-in aligned counts}/1\text{e}6)$ which were utilised in deeptools bamCoverage to generate scaled bigwig files (at 10bp resolution) for visualisation and for further normalisation in downstream analyses. Peak calling was performed using macs2¹⁰ with a q-value cutoff of 0.01. PRC2 target promoters were defined by overlapping WT SUZ12 ChIP-Rx peaks with gene promoter regions (+/- 2kb) as annotated in the mm10 build of the mouse genome using bedtools intersect. Intergenic regions were defined by excluding sites within +/- 5kb from gene bodies, using bedtools intersect with -wa -v flags. DeepTools was further utilised to generate average plots using the defined peaksets. Box plots, violin plots and chromosome-wide plots were generated from normalised counts calculated with featureCounts and visualised with BoxPlotR (<http://shiny.chemgrid.org/boxplotr/>).

In terms of replicates related to ChIP-Rx analyses, the Bracken lab routinely performs at least 3 independent biological replicates for every ChIP and CUT&RUN assay. Specifically, for ChIP-Rx analyses, we generate libraries and sequence at least one replicate and perform quantitative qPCRs on at least two independent replicates, checking at several relevant genomic loci. Our experience is that the ChIP-qPCR method is the more reliable quantitative measure of enrichment. This is because the library preparation step, which involves a subsequent PCR amplification can sometimes introduce artefacts and other issues. Therefore, due to both the cost implications and enduring value of the quantitative ChIP-qPCR approach, since 2018, we have routinely always carefully ensured the reliability of all our ChIP-Rx analyses by comparing changes in any experiment to those observed in ChIP-qPCRs of at least two independent experiments. Therefore, in this paper, we present ChIP-Rx analyses of SUZ12, JARID2 and AEBP2 in the new Fig. 4a-c and ChIP-qPCRs of two independent biological replicates in Extended Data Fig. 6a. Specifically, for CUT&RUN analyses of H3K27me3, we

sequenced libraries from three independent biological replicates, which are presented in the new Fig. 4a-c and Extended Data Fig. 6b-c.

Figure 3

- Fig 3a and b, comparable values should be provided for the figure 3a and 3b. This should be further supported by western blot to support the presence of different isoforms during development and in adult tissues.

To address this, the expression of mRNA transcripts encoding human and mouse AEBP2^L and AEBP2^S across various cell types and tissues was analyzed using CAGE-sequencing data collected and processed by the Fantom5 consortium¹¹ (Fig. 3a-b):

This established that, in mice, AEBP2^S is expressed in ESCs, placenta, and trophoblasts, but is largely not expressed in most other tissues (Fig. 3a). In human cells, AEBP2^S is primarily expressed in the testis and is otherwise not expressed in other tissues (Fig. 3b). In contrast, AEBP2^L is expressed at substantial levels in the majority of tissues in both mice and humans (Fig. 3a-b). These observations points to the broad relevance of studying the mechanism of action of AEBP2^L, as we do in this paper.

As also suggested, we performed Western blot analyses for AEBP2 isoforms during mouse development (Extended data Fig. 3d) and in adult mouse tissues (Extended data Fig. 3c), as shown below:

These results are consistent with the above mRNA analyses in that the AEBP2^L protein is widely expressed, whereas the AEBP2^S protein is more restricted, being expressed in pluripotent mouse ESCs. As mentioned just above, this data further strengthens the biological relevance of our mechanistic analysis of AEBP2^L function, as it is the prevalent isoform at least from as early as mid-gestation (E9.5 embryos) and throughout adulthood.

- The differentiation experiments presented in Figure 3 were barely analyzed.

To address this, we expanded the analyses of the RNA-seq of the four ESC lines directed to differentiate to EpiLCs in the updated Fig. 3k-m, as shown below. We firstly increased the number of example genes which gain Polycomb binding in EpiLC compared to ESCs¹² from 2 to 6 (Fig. 3l). Please note that our now published Glancy *et al.* paper identified 398 genes that become *de novo* bound by PRC2 and repressed upon differentiation of mouse naïve pluripotent ESCs to primed pluripotent epiblast-like-cells (EpiLCs). Importantly, in that paper, we showed that loss of JARID2 or Polycomb-like proteins led to loss of PRC2 recruitment and impaired repression of these 398 genes. In the revised Fig. 3l, we show that a representative six of these genes also display impaired repression in ESCs that lack AEBP2^S (SKO and *Aebp2* KO), but not ESCs that only lack AEBP2^L (LKO), as measured by Quant-Seq of three independent biological replicates (Fig. 3l). Furthermore, we performed new statistical analysis (Wilcoxon rank sum test) of all 398 *de novo* PRC2 recruited genes in Fig. 3m, which shows that the loss of AEBP2^S (in the SKO and *Aebp2* KO ESC lines), but not loss of only AEBP2^L (LKO), leads to a statistically significant impairment of *de novo* repression of these genes upon differentiation of the ESCs to EpiLCs (Fig. 3m):

Updated Fig. 3k-m. AEBP2^S but not AEBP2^L is required for repression of *de novo* Polycomb target genes during EpiLC differentiation.

Additionally, in the revised manuscript, in the new Extended Data Fig. 5d, we provide additional analyses of the above described ESC to EpiLC differentiation Quant-Seq experiment. Using DEseq2 coupled with unsupervised hierarchical clustering, we show that the loss of AEBP2^S in AEBP2^S KO (SKO) and *Aebp2* KO EpiLCs leads to the largest amount of gene expression changes compared to matched WT and AEBP2^L KO (LKO) cells (Extended Data Fig. 5d). Taken together, these data indicate that AEBP2^S has a more important role than AEBP2^L during the differentiation of ESCs to EpiLCs.

1) All experiments should be replicated and statistical analysis performed. We do not consider qPCR technical replicates to be sufficient for any statistical analysis and thus to draw any conclusions (Figure S3e).

We agree and clarified in the Figure legend description of the updated Figure (previously Fig. S3e, now Extended Data Fig. 5a) that this panel shows the mean mRNA levels from

a Quant-Seq experiment, measured as mean counts per million (CPM). Importantly, the updated Figure legend also details how the error bars represent the standard deviations calculated from three independent biological replicates:

2) RNA-seq (with replicates) should be provided

We modified the Figure legends text of Fig. 3l-m and Extended Data Fig. 5c-d in the revised manuscript to more clearly describe that we performed Quant-seq, not RNA-seq, and that the data are based on three independent replicates.

3) Western blot showing the expression of AEBP2 isoforms during differentiation in each cell line should be provided. The authors should also check JARID2 expression.

To address this, we performed western blot analysis AEBP2 and JARID2 on the CRISPR generated KO ESC lines (WT, *Aebp2* KO, SKO and LKO) before and after 2 days directed differentiation of ESCs to EpiLCs. As shown below, both AEBP2 and JARID2 proteins are relatively stable and not decreased in this two day directed differentiation (Extended Data Fig. 5b):

New Extended Data Fig. 5b.

Figure 4

The authors propose that the two acidic tracts within the N-terminal part of AEBP2-long are responsible for PRC2 inhibition via a reduced interaction with DNA. However, the presented data do not support this mechanism.

Indeed, a simple deletion of these tracts should already rescue partially the HMTase activity of PRC2/AEBP2 but this is not the case neither for mt1 Δ nor for mt2 Δ (and the data for mt1 Δ 2 Δ are not presented). The addition of exogenous positively charged residues to replace the acidic tracts, has an effect but still does not fully recapitulate the activity of PRC2/AEBP2-short and the structure of this mutant form of AEBP2 is now so artificial that the result is difficult to interpret.

Also EMSA experiments with the presented mutants would have supported the hypothesis that acidic tracts inhibit DNA binding.

As described above, in response to comments related to Fig. 1 made by this reviewer, we performed new HMTase assays using PRC2-AEBP2^L-JARID2 wildtype and its acidic tract mutants, including mt1 Δ 2 Δ , and they all demonstrated substantial increase in HMTase compared to PRC2-AEBP2^L-JARID2 wildtype (Fig. 6f):

We now added EMSA experiments using a fluorescently labelled mononucleosome probe and PRC2-AEBP2^L-JARID2 wildtype and its acidic tract mutants, including the mt1Δ2Δ mutant in Fig. 6h (plus two additional replicates in Extended Fig. 8f):

Quantitative analysis confirmed that all the mutants slightly increase the affinity of PRC2 to mononucleosomes (up to ~2-fold *K_d*; see Fig. 6i for *K_d* values; binding curves are from Extended Data Fig. 6g):

All the *K_d* values that we quantified are in the range of 33.0 ± 0.8 nM (wildtype PRC2-AEBP2^L-JARID2) to 15.6 ± 0.4 nM (mt1A2A; Fig. 6i). These dissociation constants are close to the concentration of the nucleosome probe (5 nM), which limits the possibility of quantifying large differences between dissociation constants (specifically, this is because no dissociation constant lower than the probe concentration can be quantified in such an

EMSA binding assay). Hence, it is possible that the real dissociation constants of some of the mutants are slightly lower than what is quantified. Nevertheless, we now toned down one of the conclusions that we made regarding the AEBP2^L-specific, acidic N-terminal domains in the Discussion section. The original text described the AEBP2^L-specific, acidic N-terminal domains as “**responsible for**” the ability of AEBP2^L to inhibit PRC2, whereas the revised text describes them as “**contributing to**” that. We believe this statement is well supported.

While interpreting this data, one should also consider that PRC2-EZH2 has multiple binding surfaces for nucleosomes^{13–15}. Hence, it is possible that when AEBP2^L interferes with interactions between a nucleosome and PRC2, another site on PRC2 may be available for binding to the nucleosome probe. In such a case, the apparent affinity of PRC2 for a nucleosome, as measured by EMSA, may not change substantially if the probe is still shifted through interactions with another site. This agrees with the presence of a single band during nucleosome EMSA of PRC2-AEBP2^{L(TEV)}-JARID2, but two bands in the case of a nucleosome EMSA of PRC2-AEBP2^{L(S)}-JARID2 (Extended Data Fig. 8a and pasted above, in responding to the first point by this Reviewer).

Most critically, there are previous reports of mutations in the substrate-nucleosome binding site in PRC2 that lead to only a small reduction in the apparent affinity of PRC2 for nucleosomes, despite a large reduction in HMTase activity^{2,16}. These studies exemplify the complexity of interpreting PRC2-nucleosome EMSA results. However, we believe that our additional analyses herein, including the extensive HMTase assays using chromatin and H3 peptide substrates (Fig. 6), now better support our proposed model.

Reviewer #2:

Remarks to the Author:

The conserved AEBP2 subunit of PRC2 stimulates H3K27 methylation through enhanced nucleosome binding. Different isoforms of AEBP2 are known to exist in mammalian cells, and this study explored different functions of them.

The authors find that core PRC2 with the long AEBP2 isoform 2 exhibit lower DNA-binding activity as compared to the short isoform in EMSA experiments. Additionally, PRC2 containing the long AEBP2 isoform exhibited relatively low methyltransferase activity on nucleosome substrates, as compared to the core PRC2 and short isoform. In contrast to previous studies, the authors don't observe a stimulation of HMT activity with PRC2 containing short AEBP2, as compared to the core PRC2. In mESCs, the short AEBP2 isoform led to higher AEBP2 and SUZ12 ChIP-seq signals at known sites of PRC2 localization. Loss of AEBP2 led to impaired EpiLC differentiation and aberrant gene repression. The authors performed structural analysis using cryo-EM, however, these efforts were unable to gain insight into the difference between the short and long AEBP2 isoform. The N-terminus that's unique to the long isoform contains poly-acidic and poly-E tracts in an unstructured region. Lysine substitutions in these two domains led to enhanced HMT activity of PRC2 by in vitro assays. Overall, the authors proposed that this poly-E/acidic domains inhibit binding of PRC2 to DNA and lead to loss of PRC2 activity.

Overall, the manuscript presents some interesting observations about differences between the two dominant isoforms of AEBP2. However, the study is incomplete and many important/critical experiments for properly evaluating their proposed model are missing.

It's unclear if the presumed differences of the AEBP2-long vs -short isoforms are caused by activities intrinsic to the sequence (e.g. the poly acidic stretch in the long isoform) or possible differences in the abilities to contact non-core regulatory subunits (e.g. JARID2). The possibility that the long isoform may obstruct interactions between the core PRC2 subunits and accessory proteins and/or regulatory RNAs has not been evaluated.

Thus, the manuscript in its current form is not appropriate for publication in "Journal X".

1) As mentioned above, the different cell phenotypes observed with the short vs long isoforms could be due to altered holo-PRC2.2 composition. To assess this possible difference, the authors need to assess the proteomic differences in the long vs short AEBP2 isoforms.

We thank the Reviewer for this excellent and important suggestion. We performed endogenous immunoprecipitations of IgG, SUZ12 and JARID2, coupled with mass spectrometry (IP-MS) on the four ES cell lines, WT, AEBP2^S KO (SKO), AEBP2^L KO (LKO) and full *Aebp2* KO, as shown below (Fig. 3h-j). Importantly, all iBAQ values represented below in panels 3i and 3j are the average of three independent biological replicates:

This revealed that the overall stoichiometry of core PRC2 was not altered upon KO of either AEBP2^S or AEBP2^L isoform alone or KO of both, as indicated by the fact that SUZ12 IP-MS detected similar amounts of EZH2 and EED in all 4 ESC lines (Fig. 3i). While decreased AEBP2 peptides were detected in the SUZ12 and JARID2 IP-MS of the KO lines, as expected, there were also some indications that less JARID2 was immunoprecipitated in SUZ12 IPs upon loss of AEBP2 isoforms, and vice versa, that JARID2 immunoprecipitated less core PRC2 members in the absence of AEBP2 isoforms. Taken together, while not the main point of this work, these data indicate that the AEBP2 isoforms contribute to stabilising or promoting JARID2 association with PRC2.

We also performed western blot analyses of a fourth independent replicate, as shown below (Extended Data Fig. 4d-e):

While also not the main message of the paper, the MTF2 westerns of JARID2 IPs curiously found evidence that the hybrid form of PRC2, containing JARID2 and MTF2, increases upon loss of AEBP2^S - please see weak, but reproducible, MTF2 bands in Western blots of JARID2 IPs of AEBP2^S KO and full *Aebp2* KO ESCs (Extended Data Fig. 4d-e). Brockdorff and colleagues first reported that this hybrid form of PRC2, which contains both JARID2 and MTF2, is more detectable in *Aebp2* KO ESCs compared to WT control¹⁷. While not the purpose or main point of this work, our Western blot data above extend the work of Brockdorff and colleagues¹⁷ and suggest that AEBP2^S, but not AEBP2^L, might compete with MTF2 for binding to the C2 region SUZ12 within core PRC2. However, we could not expand on this possibility in the revised paper because we were unable to validate this observation in the IP-MS, most likely due to the technical limitations of sensitivity (Fig. 3j).

2) The experiments in figure 1: the EMSA and HMT assays were performed on different substrates (EMSA on DNA, while HMT on nucleosomes). The authors suggest that the low activity on nucleosomes is due to lower DNA-binding activity, however, this is an assumption and the authors should assess nucleosome EMSA of the long and short AEBP2 isoforms.

We thank the reviewer for this suggestion. In response to the first comment of Reviewer 1, we performed many of the experiments in the revision using the PRC2-AEBP2-JARID2

complex, which they perceived as more biologically relevant. We developed an AEBP2^{L(TEV)} construct that allows the removal of the N-terminal region of AEBP2^L using TEV protease in the context of PRC2-AEBP2-JARID2 (Fig. 6a-b). As expected, TEV cleavage converted PRC2-AEBP2^{L(TEV)}-JARID2 to PRC2-AEBP2^S-JARID2, which was accompanied by a substantial increase in HMTase activity on a chromatinized substrate (Fig 6a-c):

Next, we carried out EMSA using a nucleosome probe, in the presence and absence of TEV cleavage (Extended Data Fig. 8a):

We found that PRC2-AEBP2^{L(TEV)}-JARID2 exhibits a modest reduction in the affinity for nucleosomes (see Extended Data Fig. 8b for quantification), but we consistently observed two bands when PRC2-AEBP2^{L(TEV)}-JARID2 (top) is converted to PRC2-AEBP2^S-JARID2 (bottom, marked by red asterisks). This is in contrast to PRC2-AEBP2^{L(TEV)}-JARID2, which appears as a single band (top gels). This indicates a different nucleosome binding mode and is discussed in a new section of the results, entitled “AEBP2L antagonises PRC2.2”.

While interpreting this data, one should also consider that PRC2-EZH2 has multiple binding surfaces for nucleosomes¹³⁻¹⁵. Hence, it is possible that even if AEBP2^L blocks the substrate nucleosome-binding site, another site on PRC2 may still be available for binding to the nucleosome probe. In such a case, the apparent affinity of PRC2 for nucleosomes, as measured by EMSA, may not change substantially. As examples for that, the substrate nucleosome binding site in PRC2 was identified using cryoEM and, accordingly, mutations in this site lead to a severe defect in HMTase^{2,16}. Yet, the same mutations lead to only a moderate reduction in the apparent affinity of PRC2 for nucleosomes, as measured by EMSA^{2,16}.

Furthermore, this reviewer correctly pointed out above that “*In contrast to previous studies, the authors don’t observe a stimulation of HMT activity with PRC2 containing short AEBP2, as compared to the core PRC2*”. This is because some relevant previous studies carried out HMTase assays in monovalent salt concentration of 50 mM KCl or even below that, which is much lower than physiological concentration (commonly considered as 100-150 mM KCl in cells; reviewed in⁴⁻⁶). For instance, Lee et al., 2018, Mol. Cell¹ concluded that PRC2-AEBP2^S has a higher activity than PRC2 based on HMTase assays that were carried out without monovalent salt in the HMTase buffer (their HMT buffer was: 50 mM Tris-HCl, pH 8.5, 5 mM MgCl₂, and 4 mM DTT, with no KCl or NaCl indicated). Contrarily, we performed our HMTase assays in a buffer that included a near-physiological concentration of monovalent salt: 100 mM KCl (HMTase buffer that was used here: 50 mM Tris-HCl pH 8.15 at 25°C, 100 mM KCl, 0.5 mM MgCl₂, 0.1% Tween-20 and 5 mM DTT). This is a very important difference between our study to some previous studies, including Lee *et al.* This point is important, as we recently demonstrated that the monovalent salt concentration is extremely important in HMTase assays of PRC2 when nucleosomal substrates are used. For instance, in Fig. 1f from Gail *et al* Nature Genetics, 2024², one can see that the different concentrations of KCl (0, 50 and 100 mM) have almost no effect on the HMTase activity of PRC2 for H3 substrate, but yet KCl strongly inhibits the activity of PRC2 for a chromatinized substrate:

Fig. 1f from Gail *et al* Nature Genetics, 2024².

One of our labs (C.D.) also showed that the monovalent salt concentration has a strong effect on the PRC2-PALI1 complex, whereas KCl has the opposite effect on the HMTase activity for a chromatinized substrate. Specifically, near-physiological monovalent salt concentration (100 mM KCl) increases the activity of PRC2-PALI1, while low monovalent salt concentration (30 mM KCl) reduces the activity of PRC2-PALI1 (from Fig. 7c in Zhang *et al* 2021, Nature Comms³):

Hence, the large difference between the concentration of the monovalent salt used by us here (100 mM KCl) and by Lee *et al* (0 mM KCl/NaCl) and some other previous works is likely contributing to variations in observations. We believe that our experimental system more closely resembles physiological monovalent salt concentration, which is commonly considered as 100-150 mM KCl (reviewed in⁴⁻⁶).

3) The authors need to assess the methyltransferase activity on histone H3 peptide substrates. The authors find that long AEBP2 PRC2 has a lower activity than short AEBP-

PRC2 or simply core PRC2. This experiment will help determine if the lower activity of long AEBP2 is due to lower substrate binding (may be favored considering EMSA data) or something intrinsic to the PRC2 with long AEBP2.

We thank the reviewer for this great suggestion. We performed the same HMTase experiment that was described above (for point 2 of this Reviewer) with the exception that we replaced the chromatinized substrate with H3 histone tail peptide substrate. As the reviewer predicted, the N-terminal region of AEBP2^L substantially inhibits HMTase of chromatinized substrate (Fig. 6c, discussed above), but not of H3 peptide (Fig. 6d):

This data indicates that the N-terminal region of AEBP2^L does not inhibit the active site of PRC2 directly, but rather interferes with chromatin substrate binding. This is discussed in a new section of the results, entitled “AEBP2^L antagonises PRC2.2”.

4) Additionally, it's unclear which degree of lysine-methylation is affected by the isoforms and Poly-E mutants. Immunoblotting of the reaction products using antibodies against K27me1, K27me2, and K27me3 should be used to assess which degrees of methylation are primarily affected by the isoforms and Poly-E mutants.

We thank the reviewer for this excellent suggestion. We now did the experiment and found that H3K27me3 is affected (Fig. 6g; and see another replicate in Extended Data Fig. 8e):

It should be pointed out that this result does not exclude the possibility that H3K27me2 and H3K27me1 are affected too. However, it certainly demonstrates that H3K27me3 is affected. This is discussed in a new section, entitled “AEBP2^L antagonises PRC2.2”.

5) Figure 1E uses an indirect measurement (SAH production) as a readout for PRC2 methyltransferase activity, while figure 4G uses tritiated SAM and autoradiography. For consistency, the authors should use the same assay to compare the methyltransferase activity of the different isoforms (Figure 1) and the Poly-E mutants (figure 4).

We now added additional HMTase assays that measure SAH production also for the two isoforms (Fig. 6c, discussed above) and the different mutants (Extended Data Fig. 8d):

These results are qualitatively in agreement with the radioactive HMTase assays that were done before, although we did this assay using the PRC2-AEBP2^L-JARID2 complex (see first comment of Reviewer 1). This is discussed in a new section, entitled “AEBP2^L antagonises PRC2.2”.

6) Also, steady-state kinetics should be used to assess if the AEBP2 isoform difference in HMT activity primary affects the K_m and/or K_{cat} of the substrate. This is "Journal X", and performing the bare minimum of enzyme kinetics is well within the scope of expected experiments for this study.

We now carried out a Michaelis-Menten kinetics analysis for the PRC2-AEBP2^L-JARID2 and its different mutants (Fig. 6f,i):

In agreement with the proposed model, all the PRC2-AEBP2^L-JARID2 mutants increase the catalytic efficiency of PRC2 (k_{cat}/K_m). This is discussed in a new section, entitled “AEBP2^L antagonises PRC2.2”.

7) Western blots are needed for H3K27me1/2/3 levels in mESCs with short and long AEBP2 transgenes. Does the long vs short AEBP2 lead to different levels of H3K27me in mESCs? Also, it's striking that CHIP seq for H3K27me3 (required, at the very least) was not performed in the transgenic cells. RNA-seq should also be performed in the mESCs with the isoforms.

To address this, we first performed Western blot analysis of H3K27me3 and H3K27me2 on the CRISPR KO and the matched wild type ESC lines, as shown below (Fig. 3g, and Extended Data Fig. 4b):

Fig. 3g**Extended Data Fig. 4b**
This revealed no significant changes in the global levels of H3K27me3 and H3K27me2 upon loss of either one or both AEBP2 isoforms. This is consistent with previous reports in which loss of as many as six PRC2 accessory proteins (AEBP2, JARID2, PHF1, MTF2, PHF19, and EPOP) did not affect the global levels of H3K27me2/3⁸.

We further performed ChIP-Rx of AEBP2, SUZ12, JARID2 and CUT&RUN of H3K27me3 on the four CRISPR KO ESC lines, either lacking AEBP2^S (SKO), AEBP2^L (LKO) or both isoforms (*Aebp2* KO). This is included in the new Fig. 4, as shown below, and Extended Data Fig. 6. This established that SUZ12 binding (representative of core PRC2) and H3K27me3 levels increased on Polycomb target genes in the absence of AEBP2^L (in LKO and *Aebp2* KO ESCs), but not in the absence of only AEBP2^S (SKO ESCs):

New Fig. 4. Loss of AEBP2^L but not AEBP2^S leads to increased PRC2 and H3K27me3 on Polycomb target genes in mouse ESCs.

Finally, we expanded our genome-wide RNA Quant-seq data analyses in the revised Fig. 3 and Extended Data Fig. 5. This shows that while no significant gene expression changes were observed upon KO of one or both AEBP2 isoforms in ESCs, we did observe impaired *de novo* repression of a cohort of 398 Polycomb target genes upon directed differentiation to become EpiLCs (Revised Fig. 3k-m and Extended Data Fig. 5). Since our original submission of this AEBP2 focused paper, we published a separate paper in which we identified these 398 genes that have *de novo* recruitment of Polycombs during directed differentiation of ESCs to EpiLCs¹². Importantly, we showed that loss of JARID2 or Polycomb-like proteins led to loss of PRC2 recruitment and impaired repression of these genes. In the revised Fig. 3l, shown below, we show that a representative six of these *de novo* PRC2 repressed genes display impaired repression in ESCs that lack AEBP2^S (SKO and *Aebp2* KO), but not ESCs that only lack AEBP2^L (LKO), as measured by Quant-Seq of three independent biological replicates (Fig. 3l). Furthermore, we

performed new statistical analysis (Wilcoxon rank sum test) of all 398 *de novo* PRC2 recruited genes in Fig. 3m, which shows that the loss of AEBP2^S (in the SKO and *Aebp2* KO ESC lines), but not loss of only AEBP2^L (LKO), leads to a statistically significant impairment of *de novo* repression of these genes upon differentiation to EpiLCs (Fig. 3m):

Updated Figure 3k-m. AEBP2^S but not AEBP2^L is required for repression of *de novo* Polycomb target genes during EpiLC differentiation.

Additionally, in the revised manuscript, in the new Extended Data Fig. 5d, we provide additional analyses of this ESC to EpiLC differentiation Quant-Seq experiment. Using DEseq2, we identified significantly deregulated genes ($\text{padj} < 0.05$, $\log_2\text{FC} > 1.5$), and show that AEBP2^S KO (SKO) and *Aebp2* KO ESCs display the largest amount of gene expression changes compared to matched WT ESCs (Extended Data Fig. 5d). Taken together, these data support the role of AEBP2^S being more important than AEBP2^L during ESC to EpiLC differentiation.

8) How do the levels of the transgenic AEBP2 compare to the endogenous AEBP2 levels? While the authors compare the different isoforms of AEBP2 to each other, it's important that the levels of transgenic AEBP2 are similar to the endogenous AEBP2.

This is an excellent point. We added the WT control to all relevant panels in the updated Fig. 2, as requested (Fig. 2b-g). Importantly, Western blot analyses revealed that the AEBP2 isoforms are ~2-4 fold overexpressed compared to endogenous AEBP2^S and AEBP2^L protein levels in the matched parental control ESCs, as shown below (Fig. 2b):

In terms of the ChIP-Rx analyses in Fig. 2, as shown below, we also modified these panels to now include the WT parental ESC controls throughout (Fig. 2c-f):

This established that ectopically expressed AEBP2^S binds to Polycomb target genes better than ectopically expressed AEBP2^L, while ectopically expressed AEBP2^L leads to reduced overall binding of SUZ12 (representative of core PRC2) on Polycomb target genes. This experiment in Fig. 2 was therefore valuable to clearly establish a key difference between both isoforms, which is that AEBP2^S is better than AEBP2^L at binding to Polycomb target genes in cells, supporting our *in vitro* observations in Fig. 1. Furthermore, we conclude in the revised paper that ectopic overexpression of AEBP2^L leads to an overall reduced binding of PRC2 to Polycomb target genes, presumably because it competes with other accessory proteins for binding to a limited amount of PRC2.

Importantly please note that since these exogenous expression experiments had levels of AEBP2^S and AEBP2^L at about 3-4 fold greater than their endogenous levels, we felt they could lead to artifacts in any assays of H3K27me3 or transcription. Therefore, we later explore the respective contributions of AEBP2^S and AEBP2^L to PRC2 mediated H3K27me3 deposition and transcription control by studying the consequences of loss of

their endogenous isoforms in CRISPR engineered ESC lines in the updated Figures 3 and 4.

9) Using unique gene knockout strategies, the authors found impaired EpiLC differentiation upon loss of short isoform. The authors should use the long and short AEBP2 isoform transgenes to directly test if the short isoform can rescue the EpiLC differentiation phenotype caused by the supposed loss of the short isoform. Additionally, it would be interesting to know if over expression of the long isoform could act as a dominant negative and lead to the same gene expression phenotypes observed in the short isoform knockout cells.

As described above, in the updated Fig. 2, we did observe a dominant negative effect on PRC2 binding to Polycomb target genes upon ectopic overexpression of AEBP2^L about 3-4 fold above endogenous AEBP2^L levels in ESCs. This dominant negative effect was that AEBP2^L caused a partial displacement of SUZ12 (representative of core PRC2) from Polycomb target genes (Fig. 2). In contrast, ectopic overexpression of AEBP2^S led to increased SUZ12 (again representative of core PRC2) binding on Polycomb target genes (Fig. 2e-f). However, due to this 3-4 fold overexpression of the AEBP2 isoforms, we did not feel that our ectopic expression experimental strategy was suitable for comparing them during EpiLC differentiation as we were unable to closely balance their ectopic and endogenous levels.

10) The authors need to determine if the PolyE to PolyK substitution in figure 4 in the EMSA experiment performed in figure 1. Additionally, the authors should determine if the mt1K2K transgene phenocopies the short AEBP2 isoform in the EpiLC experiments. Additionally, it's unclear why one transgene listed in figure 4F (deletion of M1 and M2) was not included in the HMT assay in figure 4G. These experiments are needed to determine if the E to K substitutions really do in fact suppress the observed differences the long AEBP2 isoform.

We now performed Michaelis-Menten kinetic analysis of PRC2-AEBP2^L-JARID2 wildtype and mutants, including mt1 Δ 2 Δ , using a chromatinized substrate (Fig. 6f,i):

i

PRC2-AEBP2 ^L -JARID2	HMTase Activity			Nucleosome Binding	
	k_{cat} [10^{-4}S^{-1}]	K_M [nM]	k_{cat}/K_M [$10^{-5}\text{nM}^{-1}\text{S}^{-1}$]	K_d [nM]	Hill coefficient
WT	11.8 ± 0.3	37.7 ± 3.4	3.1 ± 0.3	33.0 ± 0.8	2.9 ± 0.2
mt1K2K	23.9 ± 0.5	34.8 ± 2.4	6.9 ± 0.5	27.2 ± 0.7	3.1 ± 0.2
mt1A2A	19.5 ± 0.5	24.9 ± 2.4	7.8 ± 0.8	15.6 ± 0.3	3.5 ± 0.2
mt1Δ2Δ	25.1 ± 0.4	30.6 ± 2.0	8.2 ± 0.5	25.3 ± 0.8	3.9 ± 0.4

This data indicates that all the mutants increase the HMTase activity of PRC2-AEBP2^L-JARID2.

We also added EMSA experiments using a fluorescently labelled mononucleosome probe and PRC2-AEBP2^L-JARID2 wildtype and its acidic tract mutants mutant in Fig. 6h (an additional two replicates are in Extended Fig. 8f):

Furthermore, densitometry analysis of these results are in Extended Data Fig. 6g:

This quantitative analysis confirms that all the mutants slightly increase the affinity of PRC2 to mononucleosomes (up to ~2-fold K_d , see Fig. 6i for K_d values). All the K_d values that we quantified are in the range of 33.0 ± 0.8 nM (wildtype PRC2-AEBP2^L-JARID2) to 15.6 ± 0.4 nM (mt1A2A; Fig. 6i). These dissociation constants are close to the concentration of the nucleosome probe (5 nM), which limits the possibility of quantifying large differences between dissociation constants (specifically, no dissociation constant lower than the probe concentration can be quantified in such an EMSA assay). Hence, it is formally possible that the real dissociation constants of some of the mutants are slightly lower than what is quantified. This data indicates that the acidic tracts of AEBP2^L interfere with chromatin binding, at least to some extent. Nevertheless, we now toned down one of the conclusions that were made regarding the AEBP2^L-specific acidic N-terminal domains in the Discussion section. The original text described the AEBP2^L-specific, acidic N-terminal domains as “**responsible for**” the ability of AEBP2^L to inhibit PRC2, whereas the revised text describes them as “**contributing to**” that. We believe this statement is well supported.

While interpreting the EMSA data, one should also consider that PRC2-EZH2 has multiple binding surfaces for nucleosomes¹³⁻¹⁵. Hence, it is possible that while AEBP2^L blocks one substrate-nucleosome-binding site in PRC2, another nucleosome-binding site may be available for binding to the nucleosome probe. In such a case, the apparent affinity of PRC2 for nucleosomes, as measured by EMSA, may not change substantially. As examples for that: the substrate nucleosome binding site in PRC2 was identified using cryoEM and, accordingly, mutations in this site lead to severe defects in HMTase^{2,16}. Yet, the same mutations led to only a moderate reduction in the apparent affinity of PRC2 for nucleosomes, as quantified using EMSA^{2,16}.

Beyond that, we agree with the reviewer that it could be useful to evaluate if the mt1K2K mutant phenocopies the short AEBP2 isoform in cells. As mentioned above, since our ectopic rescue assay had a 3-4 fold overexpression AEBP2 isoforms, it was not suitable to use this system in a differentiation assay for comparing WT and mutant forms of AEBP2 in cells. Furthermore, we were unable to express AEBP2^L mt1K2K in mECS to a similar expression level to the other AEBP2 constructs in (Extended Data Fig. 9b):

Therefore, we had to exclude AEBP2^L mt1K2K from experiments in ESCs, but we did assay AEBP2^L mt1Δ2Δ in ESCs. Specifically, we fused a TetR protein to AEBP2^S, AEBP2^L and AEBP2^L mt1Δ2Δ and targeted them to a chromosomal-integrated reporter that includes a TetO array (Extended Data Fig 9a,c):

These results indicate that all the different AEBP2 proteins that we tested are active in gene repression when are forced onto the reporter through the TetO:TetR interactions. Specifically, the FACS analysis (above) shows that the reporter is repressed in >90% of the cells regardless of the AEBP2 construct that was used. Accordingly, ChIP-qPCR identified no difference in H3K27me3 deposition on the TetO array between any of the reporter cell lines (Extended Data Fig. 9g):

Hence, tethered AEBP2^L and AEBP2^S promote similar PRC2 enzymatic activity in cells (Extended Data Fig. 9), whereas the untethered AEBP2^L causes lower PRC2 enzymatic activity than AEBP2^S in cells (Fig. 4) and *in vitro* (Fig. 1, 5 and 6). This data implies that the heterologous tethering of AEBP2^L to chromatin serves to rescue or nullify the inhibitory activity of the N-terminal region of AEBP2^L. This agrees with a model where

AEBP2^L inhibits the chromatin-binding activity of PRC2. This point is discussed in a new section, entitled “AEBP2^L antagonises PRC2.2”.

Other comments:

There are two long isoforms in figure 1A, however, isoform 1 is not studied in the subsequent experiments in the study. The authors need to explain why only long isoform 2 was used for the additional studies.

We focused on long isoform 1 in Figure 2 because our work in Fig. 1 established that long isoforms 1 and 2 behaved equivalently. Of note, Figure 3 of the revised manuscript achieves KO of both long isoforms and shows that their loss correlates with increased PRC2 binding and H3K27me3 deposition on Polycomb target genes.

A sequence alignment of the three different AEBP2 informs studied in the manuscript should be included in supplemental figure 1.

We thank Reviewer 2 for this helpful suggestion. As shown below, a sequence alignment of the human AEBP2 isoforms AEBP2^L Iso1, AEBP2^L Iso2 and AEBP2^S is now included in Extended Data Fig.1a:

a

hAEBP2 ^L Iso1	1	MAAAITDMADLEELSRLSPLPPGSPGSAARGRAEPPEEEEEEEEEAEAEAVAALLLNGGSGGGGGGGGGV	74
hAEBP2 ^L Iso2	1	MAAAITDMADLEELSRLSPLPPGSPGSAARGRAEPPEEEEEEEEEAEAEAVAALLLNGGSGGGGGGGGGV	74
hAEBP2 ^S		-----	
hAEBP2 ^L Iso1	75	GGGEAETMSEPSPEASQAGEDEDEEEDDEEEDDESSSSGGGEEESSAESLVGSSGGSSSDETRSLSPGAASSS	148
hAEBP2 ^L Iso2	75	GGGEAETMSEPSPEASQAGEDEDEEEDDEEEDDESSSSGGGEEESSAE L VGSSGGSSSDETRSLSPGAASSS	148
hAEBP2 ^S		-----	
hAEBP2 ^L Iso1	149	SGDGDGKEGLEEPKGRPGSQGGGGGSSSSVVSSGGDEGYGTGGGSSATSGGRRGSLMSSDGEPLSRMDSE	222
hAEBP2 ^L Iso2	149	SGDGDGKEGLEEPKGRPGSQGGGGGSSSSVVSSGGDEGYGTGGGSSATSGGRRGSLMSSDGEPLSRMDSE	222
hAEBP2 ^S	1	-----MYTRY	6
hAEBP2 ^L Iso1	223	DSISSTIMDVSTISSGRSTPAMNGQGSTTSSSKNIAYNCWQCCQACFNSSPDLADHIRSIHVDGQRGGVFV	296
hAEBP2 ^L Iso2	223	DSISSTIMDVSTISSGRSTPAMNGQGSTTSSSKNIAYNCWQCCQACFNSSPDLADHIRSIHVDGQRGGVFV	296
hAEBP2 ^S	7	SSISSTIMDVSTISSGRSTPAMNGQGSTTSSSKNIAYNCWQCCQACFNSSPDLADHIRSIHVDGQRGGVFV	80
hAEBP2 ^L Iso1	297	CLWKGCKVYNTPSTSQSWLQRHMLTHSGDKPFKCVVGGCNASFASQGLARHVP THFSQQNSSKVSSQPKAKEE	370
hAEBP2 ^L Iso2	297	CLWKGCKVYNTPSTSQSWLQRHMLTHSGDKPFKCVVGGCNASFASQGLARHVP THFSQQNSSKVSSQPKAKEE	370
hAEBP2 ^S	81	CLWKGCKVYNTPSTSQSWLQRHMLTHSGDKPFKCVVGGCNASFASQGLARHVP THFSQQNSSKVSSQPKAKEE	154
hAEBP2 ^L Iso1	371	SPSKAGMNRRLKKNKRRRSLPRPHDFFDAQTLDAIRHRAICFNLSAHIESLGKGHSVVFHSTVIAKRKEDSGK	444
hAEBP2 ^L Iso2	371	SPSKAGMNRRLKKNKRRRSLPRPHDFFDAQTLDAIRHRAICFNLSAHIESLGKGHSVVFHSTVIAKRKEDSGK	444
hAEBP2 ^S	155	SPSKAGMNRRLKKNKRRRSLPRPHDFFDAQTLDAIRHRAICFNLSAHIESLGKGHSVVFHSTVIAKRKEDSGK	228
hAEBP2 ^L Iso1	445	IKLLLHMMPEDILPDWVWNERHQLKTKVVHLSKLPKDTALLLDPNIYRTMPQKRLKRTLIRKVFNL YLSKQ	517
hAEBP2 ^L Iso2	445	IKLLLHMMPEDILPDWVWNERHQLKTKVVHLSKLPKDTALLLDPNIYRTMPQKRLKRLR-----	503
hAEBP2 ^S	229	IKLLLHMMPEDILPDWVWNERHQLKTKVVHLSKLPKDTALLLDPNIYRTMPQKRLKRTLIRKVFNL YLSKQ	301

Figure 2— better loading control for the western blot is needed to assess if the cells have similar levels of the different AEBP2 transgenic proteins.

We agree that ideally we would have performed the usual loading controls of GAPDH or ACTIN for this particular experiment. We went back and checked what else we blotted on this experiment, which was the control for the subsequent ChIP-Rx analyses in Fig. 2. We found that we also performed a Western blot for BAF155, a member of the BAF complex and therefore, a non-PRC2 protein. While it's not perfect - there's an issue with the band in the WT lane - it is clear that the loading for the two AKO-L and AKO-S lanes were about equivalent. Therefore, we believe that the two exogenous forms of AEBP2 were expressed at equivalent levels in this experiment, which was used for the ChIP-Rx analyses in the rest of Fig. 2. Of note perhaps, we subsequently confirmed that loss of AEBP2 isoforms does not affect the amount of EZH2 immunoprecipitated in IP-MS of endogenous SUZ12 (New Fig. 3i), thereby partially justifying the inclusion of EZH2 as a loading control in Fig. 2b. Finally, we also performed a Western of FLAG (also not shown in Fig. 2b) and this confirmed the result of the western with the endogenous AEBP2 antibody.

Reviewer Figure 3. Additional western blots for BAF155 and FLAG were performed on the experiment shown in Fig. 2b. This provides additional evidence that the loading for AKO-L and AKO-S lanes were about equal. Therefore, coupled with the Western blot using an AEBP2 antibody, we believe we can state that the two exogenous forms of AEBP2 were expressed at equivalent levels in the experiment in Fig. 2.

Figure 3H and 3I— it's unclear which statistical analysis was used to assess the significance of the expression differences in the four different genotypes. For example, figure 3I doesn't seem to show any significant difference in expression of PRC2-regulated genes. We thank the reviewer for pointing this out. To substantiate our point that *de novo* repression of the 398 PcG targets is impaired only in SKO and *Aebp2* KO, but not in LKO, we performed a Wilcoxon rank sum test and added this to the updated Fig. 3m (previously Fig. 3l), as shown below. This confirmed that the expression of these 398 genes in EpiLCs are statistically significant different between WT compared to SKO and WT compared to *Aebp2* KO, but not between WT compared to LKO:

Reviewer #3:

Remarks to the Author:

In the manuscript entitled “Auto-inhibitory function of AEBP2 controls PRC2 activity during development through isoform switching”, the authors describe a potential new role of two AEBP2 isoforms in regulating PRC2 activity in mouse ESC and their role in epiblast differentiation. Moreover, they show evidence that acidic tracts at the N-terminal region of the long isoform of AEBP2 is responsible for inhibition of PRC2 *in vitro*. The manuscript is well written, but the data presented in this version is very limited and not particularly strong. Most importantly, this reviewer does not believe that this study represents a major advance in our understanding on Polycomb biology, therefore I cannot recommend publication of this manuscript.

Our new data indicates that the long isoform of AEBP2 (AEBP2^L) is effectively the only AEBP2 isoform that expressed in most tissues from early embryogenesis (new Fig 3a-b

and new Extended Data Fig 3c-d; discussed in point 1 below). In addition to this revelation, we provide here a mechanistic characterisation to explain how AEBP2^L regulates PRC2.2 (new Fig 6; discussed below). Hence, our study represents a major advance in understanding Polycomb biology, by showing that AEBP2^L is broadly expressed and revealing its mechanism of action. While an inhibition of PRC2 by the germline-specific protein EZHIP was already identified, our work implies that the autoinhibition of PRC2 is a much broader phenomenon, relevant in most cell types during the majority of embryonic development and adulthood by AEBP2^L.

Major comments:

1. According to figure 3a/b, AEBP-L is highly expressed in adult tissues, while AEBP-S expression is more restricted to early development. This is a nice observation. The problem is that in mouse ESC, which is the model system they used throughout the manuscript, AEBP-L is more expressed than the short version (figure 3e). Moreover, the authors wrote that “Consistent with the reduced chromatin binding ability of AEBP2-L, this was the predominant isoform in the nuclear soluble fraction. This suggests that the majority of endogenous AEBP2-L is not stably bound to chromatin.” This statement is problematic because in figure S3C, the chromatin bound AEBP-L is actually higher than AEBP-S. Thus, in mESC, both isoforms are highly expressed and associated to chromatin with very similar affinity. Thus, it seems that ESC was not the best model to use. Finally, it is not clear why the authors did not include cells from adult tissues to investigate the role AEBP-S.

We are grateful for the opportunity to clarify the text on this point. Firstly, we agree that AEBP2^L is also abundant in ESCs and therefore modified the above mentioned sentence to be more clear of our intended meaning. Our intended meaning was to compare the relative levels of AEBP2^L in the nuclear soluble fraction versus the chromatin bound fraction. As shown below, there is marginally more AEBP2^L in the nuclear soluble fraction (now Extended Data Fig. 4c). Therefore, the revised text in the results section reads as follows, “*Consistent with the reduced chromatin binding ability of AEBP2^L demonstrated in Fig. 1 and Fig. 2, AEBP2^L relative levels were higher in the nuclear soluble fraction*

compared to the chromatin bound fraction. In contrast, the majority of AEBP2^S was bound to chromatin, consistent with its greater chromatin binding ability’.

Furthermore, the updated Fig. 3 better establish that mRNA for AEBP2^S is restricted to very early development, whereas AEBP2^L is more broadly expressed, as shown below:

Furthermore, we also performed Western blot analyses for AEBP2 isoforms during mouse development (Extended Data Fig. 3d) and in adult mouse tissues (Extended Data Fig. 3c), as shown below. Consistent with the RNA analyses, AEBP2^L protein is widely expressed, whereas AEBP2^S protein is more restricted, being only detected in pluripotent mouse ESCs, but not in E9.5-16.5 embryonic tissues or in adult spleen, liver or brain:

Based on these updated findings, we explain in the revised manuscript that differentiation of naïve pluripotent ESCs to primed pluripotent EpiLCs is an ideal model system to study the function of AEBP2^S, and to compare it to AEBP2^L, since both isoforms are present. Supporting this, we also establish in the new Extended Data Fig. 5b that both AEBP2 protein isoforms are expressed in EpiLCs. The revised Fig. 3 establishes that AEBP2^S, but not AEBP2^L, contributes to the *de novo* repression of a cohort of 398 Polycomb target genes that are expressed in ESCs and become repressed in EpiLCs. Notably, this model system is helpful to show that AEBP2^L does not have a role in this PRC2 mediated *de novo* repression despite being present in both ESCs and EpiLCs. We already know from work by the Brockdorff lab that loss of both isoforms of AEBP2 in mice causes a Trithorax phenotype. Based on our work in the new Fig. 4, in which we establish that SUZ12, JARID2 and H3K27me3 increase on Polycomb target genes in ESCs lacking AEBP2^L (LKO), but not in ESCs lacking only AEBP2^S (SKO), we conclude in our revised abstract by proposing AEBP2^S is a Polycomb protein, whereas AEBP2^L acts as a Trithorax-like protein and antagonist of PRC2 in most cells of the developing and adult body.

We are however studying AEBP2^L function in an adult cancer context in a separate study which is in preparation for submission elsewhere. In that study, we show that AEBP2^S is not expressed in germinal centre lymphomas and that loss of AEBP2^L leads to elevated levels of H3K27me3, consistent with what we observe here in ESCs. In conclusion,

AEBP2^L acts as an antagonist of the ability of PRC2 to deposit H3K27me3 in both ESCs (this paper) and in adult cells.

2. This reviewer does not agree with the statement that “AEBP2-L inhibits the DNA-binding and HMTase activities of PRC2 in vitro.” While it is clear that AEBP-L inhibits PRC2 activity, the data presented in figure 1 does not support that AEBP-L also inhibit DNA-binding of the complex because PRC2 without AEBP-L also does not bind to DNA. The DNA-binding activity of the core PRC2 complex, in the absence of accessory subunits, is very low (reviewed by Owen and Davidovich, NAR 2022¹⁸). Therefore, it is expected that the PRC2 core complex will not bind to DNA with high affinity. Our data in Fig. 1 indicate that PRC2-AEBP2^S binds to DNA, but PRC2-AEBP2^L does not. Hence the comparison between AEBP2^S to AEBP2^L is important for determining the mechanism. With that in mind, AEBP2^L is very similar to AEBP2^S, except for the long N-terminal region that is unique to AEBP2^L. Hence, the simplest explanation for the reduced affinity of PRC2-AEBP2^L for DNA is the presence of the N-terminal region, which interferes with DNA binding.

To further support this model, we now added an additional experiment: we generated a PRC2-AEBP2-JARID2 complex using an AEBP2 construct that included a cleavable TEV sequence in between the N-terminal region of AEBP2 to the rest of the protein. This AEBP2 construct (AEBP2^{L(TEV)}) is almost identical to AEBP2^L, except that upon TEV cleavage it is converted into AEBP2^S (Fig. 6a-b):

We then used PRC2-AEBP2^{L(TEV)}-JARID2 (PRC2.2) in an HMTase assay that was performed after TEV cleavage or without TEV cleavage. This was done to determine how

the N-terminal region of AEBP2^L affects PRC2 HMTase activity. Importantly, we performed this experiment using two different substrates: In the first HMTase assay, we used a chromatinized substrate. As expected, the N-terminal region of AEBP2^L had a substantial inhibitory effect on the HMTase activity of PRC2.2 on the chromatinized substrate (Fig. 6c):

Then, we performed exactly the same assay, with the exception that instead of the chromatinized substrate, we used an H3 peptide substrate. Here, the N-terminal region of AEBP2^L had no effect on the HMTase activity of PRC2 for the H3 peptides (Fig. 6d):

This observation that the N-terminal region of AEBP2^L inhibits PRC2 HMTase activity on a chromatinized substrate, but not on an H3 peptide substrate, reaffirms our original conclusion: The N-terminal region of AEBP2^L inhibits PRC2 HMTase activity by interfering

with the interactions between PRC2 and chromatin. This is now discussed in a new section, entitled “AEBP2^L antagonises PRC2.2”.

3. It is not clear why the authors decided to use human AEBP2 versions, instead of mouse AEBP2, in the mouse ESC experiments.

This is an important point. We added this line in the results section to clarify why we stick with the human ORFs for the experiments in Fig. 2 in which we ectopically expressed human AEBP2^S and AEBP2^L in mouse ESCs: “..we decided to ectopically express human AEBP2 isoforms to allow for a more direct comparison with the in vitro experiments in Fig. 1, while being reassured by the fact that the mouse and human proteins are highly conserved (Extended Data Fig. 2a)”. Importantly, as shown below, and now in the revised Extended Data Fig. 2a, we show the very high levels of sequence homology. This suggested to us that the functions of mouse and human ORFs would most likely be similar:

4. Figure 2 lacks an important control, which is AEBP2 and SUZ12 ChIP-seq in WT cells. Without these experiments, the conclusions are not well supported by the data. Moreover, it is surprising that H3K27me3 ChIP-seq was not performed in this figure.

This point was also raised by the other reviewers. To address it, we added the WT control to all relevant panels in the updated Fig. 2 and Extended Data Fig. 2, as requested. Importantly, the Western blot analyses revealed that the AEBP2 isoforms are 2-4 fold overexpressed relative to the levels of endogenous AEBP2^S and AEBP2^L in the matched parental control E14 ESCs (Fig. 2b):

In terms of the ChIP-Rx analyses in Fig 2, as shown below, we also modified these panels to now include the WT parental ES controls throughout (Fig 2c-f):

This Fig. 2 is useful in that it establishes an important key difference between AEBP2^S and AEBP2^L, which is that AEBP2^S is better than AEBP2^L at binding to Polycomb target genes in ESC cells (Fig. 2c-d). It also establishes that ectopic overexpression of AEBP2^L causes reduced SUZ12 binding, representative of core PRC2, on Polycomb target genes (Fig. 2e-f). This supports our *in vitro* observations in Figure 1 in which we establish that AEBP2^S-PRC2 binds to nucleosomes better than AEBP2^L-PRC2. Taken together, we conclude in the revised paper that AEBP2^L competes with other accessory proteins, including MTF2 and AEBP2^S, for binding to a limited amount of core PRC2, and

consequently, ectopic overexpression of AEBP2^L leads to an overall reduced binding of PRC2 to Polycomb target genes.

The above experiment, while informative for comparing the relative abilities of ectopic forms of the AEBP2^S and AEBP2^L isoforms to bind to chromatin and Polycomb target genes, it was not useful or informative the relative contributions of either AEBP2 isoform to H3K27me3 deposition or PRC2 mediated repression because their exogenous levels were 3-4 fold greater than their endogenous levels. Therefore, we instead focused all our subsequent ChIP and Cut&Run of PRC2 components and H3K27me3 (new Fig. 4 and Extended Data Fig. 6), RNA-seq and proteomics (Fig. 3 and Extended Data Fig. 4-5) experiments on our CRISPR engineered ESC lines with either knockout of the endogenous AEBP2^S isoform (SKO), the AEBP2^L isoform (LKO) or both isoforms (*Aebp2* KO). This included generating multiple replicates, to address the consequences to PRC2 binding on Polycomb target genes, transcription and H3K27me3 deposition. These various experiments are described in more detail below. However, the take home message is that loss of AEBP2^L, but not loss of AEBP2^S, leads to increased PRC2 binding and H3K27me3 deposition on Polycomb target genes (see new Fig. 4 and Extended data Fig. 6):

New Fig. 4. Loss of AEBP2^L but not AEBP2^S leads to increased PRC2 and H3K27me3 on Polycomb target genes in mouse ESCs.

5. The manuscript has very little mechanistic insight. Few obvious questions and experiments that surprisingly were not yet addressed: 1) is the architecture of PRC2.2 affected in cells expressing only one of the two isoforms? 2) what are the levels of H3K27me3 in AEBP2 KO+L and AEBP2 KO+S? 3) I assume that ncPRC1 recruitment should not be affected in AEBP2 KO+L and AEBP2 KO+S cells, but what does the recruitment of cPRC1 and PRC2.1 look like in AEBP2 KO+L and AEBP2 KO+S cells? Given the complexity of the Polycomb-mediated repression mechanisms, it is important to investigate the impact of PRC1 and PRC2.1.

We agree that these are important questions and we have addressed them in the context of our CRISPR engineered ESC lines, either lacking AEBP2^S (SKO), AEBP2^L (LKO) or both (AEBP2 KO), as detailed below.

1) Firstly, in terms of addressing the architecture of PRC2.2, in the revised Fig. 3, we performed endogenous immunoprecipitations of IgG, SUZ12 and JARID2, coupled with mass spectrometry (IP-MS) on the four ES cell lines, WT, AEBP2^S KO (SKO), AEBP2^L KO (LKO) and full AEBP2 KO, as shown below (Fig. 3h-j). Importantly, all iBAQ values represented below in panels 3i and 3h are the average of three independent biological replicates:

This above experiment revealed that the overall stoichiometry of core PRC2 was not altered upon KO of either AEBP2^S or AEBP2^L isoform alone or KO of both, as indicated by the fact that SUZ12 IP-MS detected similar amounts of EZH2 and EED in all 4 ES cell lines (Fig. 3i). Furthermore, the proportion of PRC2.1 specific EPOP and MTF2 peptides, was not significantly impacted in the SUZ12 IP-MS upon the loss of either one or both AEBP2 isoforms. This suggests that the composition of PRC2.1 is not impacted upon the loss of AEBP2 isoforms. In terms of the impact of the composition of PRC2.2, while decreased AEBP2 peptides were detected in the SUZ12 and JARID2 IP-MS of the KO lines, as expected, there were also some indications that less JARID2 was immunoprecipitated in SUZ12 IPs upon loss of AEBP2 isoforms. Vice versa - JARID2 immunoprecipitated less core PRC2 members in the absence of AEBP2 isoforms. Taken together, while not the main point of this work, these data indicate that the AEBP2 isoforms contribute to stabilising or promoting JARID2 association within PRC2.2.

We also performed western blot analyses of a fourth independent replicate, as shown below (Extended Data Fig. 4d-e):

While also not the main message of the paper, the MTF2 westerns of JARID2 IPs curiously found evidence that the hybrid form of PRC2, containing JARID2 and MTF2, increases upon loss of AEBP2^S - please see weak, but reproducible, MTF2 bands in Westerns blots of JARID2 IPs of AEBP2^S KO and full Aebp2 KO ESCs (Extended Data Fig. 4d-e). Brockdorff and colleagues first reported that this hybrid form of PRC2, which contains both JARID2 and MTF2, was more detectable in AEBP2 full KO ESCs compared to WT control¹⁷. While not the purpose or main point of this work, our Western blot data above extend the discovery of Brockdorff and colleagues¹⁷ and suggest that AEBP2^S, but not AEBP2^L, might compete with MTF2, and perhaps other Polycomb-like proteins, for binding to C2 region of SUZ12 within core PRC2. However, we were not able to validate this observation in the IP-MS, most likely due to insufficient sensitivity (Fig. 3j). The biological and mechanistic roles of this hybrid form of PRC2 could be the subject of subsequent studies, but are beyond the scope and focus on this paper.

2) Secondly, in terms of addressing the levels of H3K27me3 in the absence of one or both AEBP2 isoforms - in the revised Fig. 4 (shown above) and Extended Data Fig. 6, we performed CHIP-Rx of SUZ12, AEBP2 and JARID2 and CUT&RUN of H3K27me3 in the four ESC lines. This revealed that SUZ12, JARID2 and H3K27me3 increased on Polycomb target genes in the absence of AEBP2^L (LKO and Aebp2 KO), but not in the absence of AEBP2^S alone (SKO). This is consistent with the main message of this paper

– that is that AEBP2^L is relatively poor at binding chromatin and its presence competes with other accessory proteins, including AEBP2^S and MTF2, which are all better at bringing PRC2 to its target genes. Therefore the loss of AEBP2^L coincides with increased Polycomb binding on target genes.

3) Thirdly, while we agree it would be interesting, we felt the analyses of ncPRC1 and cPRC1 were beyond the scope of this particular study. However, to address this, we added a section to the discussion in which we speculate that due to the role of ncPRC1 upstream of PRC2, its recruitment would not be expected to be affected by the increased PRC2 and H3K27me3 levels at Polycomb target genes in ESCs lacking AEBP2^L (LKO and full *Aebp2* KO). In contrast, we speculate that we'd expect to see increased cPRC1 recruitment on Polycomb target genes with elevated PRC2 binding and H3K27me3 enrichments in the ESC lines lacking AEBP2^L (LKO and full *Aebp2* KO). We write as follows: “*While the increased H3K27me3 in the absence of AEBP2^L would not be predicted to affect upstream non-canonical PRC1, it would be expected to lead to increased canonical PRC1 recruitment and gene repression*”. Importantly, we further speculate that this increased cPRC1 recruitment would confer increased repression of Polycomb target genes in the absence of AEBP2^L and conclude by proposing that it is the loss of AEBP2^L, not AEBP2^S, that most likely contributes to the Trithorax phenotype observed upon knockout of both *Aebp2* isoforms in mice¹⁷.

6. In contrast to the authors' conclusion, the ChIP-qPCR results shown in figure S3D do not confirm the ChIP-seq results. See T TSS.

To clarify on this point, we performed ChIP-qPCR and ChIP-Rx analyses of SUZ12 in two separate contexts. In Fig. 2 and Extended Data Fig. 2, we performed SUZ12 ChIP-Rx (Fig. 2e-f) and ChIP-qPCRs of an independent replicate (Extended Data Fig. 2e), which both show that ectopic expression of AEBP2^L reduced SUZ12 (representative of core PRC2) binding at Polycomb target genes, whereas ectopic expression AEBP2^S increased SUZ12 binding at Polycomb target genes. In the new Fig. 4 and Extended Data Fig. 6, we performed SUZ12 ChIP-Rx (Fig. 4a-c) and ChIP-qPCRs from two independent replicates (Extended Data Fig. 6a), which show that loss of AEBP2^L (LKO and *Aebp2* KO) led to increased SUZ12 binding at Polycomb target genes, whereas loss of AEBP2^S on

its own (SKO) did not lead to any change in SUZ12 binding at Polycomb target genes. Taken all together, these data from Fig. 2 and 4 are central to supporting our core model that the presence of AEBP2^L, due to its relatively weak ability to bind to chromatin, acts to restrict overall binding of PRC2 to Polycomb target genes.

7. There is not a good rationale that supports the differentiation of ESC to EpiLC, other than they found 398 de novo PRC2 targets in EpiLC (not shown). Which isoform is expressed in EpiLC? Most importantly, the differentiation results of the SKO cells are very weak and indeed suggest that both isoforms may play a synergetic role in EpiLC. The characterization of the EpiLC differentiation assays is extremely poor. The authors only analyzed by RT-qPCR two EpiLC and ESC specific genes. Also, the conclusions are not supported by the data because only 1 out 2 EpiLC genes analyzed show some difference in expression in the SKO cells, and both Nanog and Prdm14 are strongly downregulated in all the cell lines. Overall, this set of experiments feel very preliminary and not particularly strong.

We addressed this by expanding on the rationale to study AEBP2 isoform function in a model of directed differentiation of naïve pluripotent ES cells (ESCs) to post-implantation pre-gastrulation epiblast-like cells (EpiLCs) in the revised manuscript. Firstly, in the revised Fig. 3a-b, we show that while AEBP2^L is widely expressed, AEBP2^S expression is more restricted to ESCs, placenta, trophoblast and testis:

Based on this restricted expression of AEBP2^S to very early development, we decided that a model of naïve (ESC) to primed (EpiLC) pluripotent differentiation was the most appropriate model to compare AEBP2^S function to AEBP2^L. Furthermore, we validated in

the new Extended Data Fig. 5b that both AEBP2^S and AEBP2^L proteins are expressed in both ESCs and ESCs differentiated to EpiLCs, as shown below:

Our since published study, using this ESC to EpiLC differentiation system, established that 398 genes expressed in naïve ESCs are downregulated upon directed differentiation to primed pluripotent EpiLCs and have accumulation of PRC2 and PRC1 when repressed in EpiLCs¹². Importantly, this study also established that Polycomb-like proteins and JARID2 are required to recruit PRC2.1 and PRC2.2 to these gene promoters and stabilise repression, respectively. In Fig. 3m (shown below), we show that while loss of AEBP2^S impairs the repression of these *de novo* repressed 398 genes, the loss of AEBP2^L does not cause any defect:

This result is important because it implicates AEBP2^S as acting as a Polycomb group protein during very early development, whereas our other data (Fig. 4) and that of others¹⁷ implicate AEBP2^L as acting as a Trithorax-like protein.

In addition to expanding on the rationale and significance of these data, we also extended the analyses of the Quant-seq of the four ESC lines directed to differentiate to EpiLCs in

the updated Fig. 3l-m, as shown above. We firstly increased the number of examples of genes which gain Polycomb binding in EpiLC compared to ESCs from 2 to 6 (Fig. 3l). In this revised Fig. 3l, we show that these representative 6 genes display impaired repression in EpiLCs that lack AEBP2^S (SKO and *Aebp2* KO), but not EpiLCs that lack AEBP2^L (LKO), as measured by Quant-Seq of three independent biological replicates. Furthermore, we performed new statistical analysis (Wilcoxon rank sum test) of all 398 *de novo* PRC2 recruited genes in Fig. 3m, which shows that the loss of AEBP2^S (in the SKO and *Aebp2* KO ESCs), but not loss of only AEBP2^L (LKO), leads to a statistically significant impairment of *de novo* repression of these genes upon differentiation of the ESCs to EpiLCs (Fig. 3m). Additionally, in the new Extended Data Fig. 5d, we provide additional analyses of this ESC to EpiLC differentiation Quant-Seq experiment. Using DEseq2, we show that AEBP2^S KO (SKO) and *Aebp2* KO ESCs display the largest amount of gene expression changes compared to matched WT cells (Extended Data Fig. 5d):

Taken together, these data support the role of AEBP2^S being more important than AEBP2^L during ESC to EpiLC differentiation.

I cannot make specific comments on figure 5 because I am not a structural biologist.

References:

1. Lee, C. H. *et al.* Distinct Stimulatory Mechanisms Regulate the Catalytic Activity of Polycomb Repressive Complex 2. *Mol. Cell* **70**, 435-448.e5 (2018).
2. Gail, E. H. *et al.* Inseparable RNA binding and chromatin modification activities of a nucleosome-interacting surface in EZH2. *Nat. Genet.* 2024 566 **56**, 1193–1202 (2024).
3. Zhang, Q. *et al.* PALI1 facilitates DNA and nucleosome binding by PRC2 and triggers an allosteric activation of catalysis. *Nat. Commun.* **12**, (2021).
4. Melkikh, A. V. & Sutormina, M. I. Model of active transport of ions in cardiac cell. *J. Theor. Biol.* **252**, 247–254 (2008).
5. Thier, S. O. Potassium physiology. *Am. J. Med.* **80**, 3–7 (1986).
6. Zacchia, M., Abategiovanni, M. L., Stratigis, S. & Capasso, G. Potassium: From Physiology to Clinical Implications. *Kidney Dis.* **2**, 72–79 (2016).
7. Al- Raawi, D. *et al.* A novel form of JARID2 is required for differentiation in lineage-committed cells. *EMBO J.* **38**, (2019).
8. Højfeldt, J. W. *et al.* Non-core Subunits of the PRC2 Complex Are Collectively Required for Its Target-Site Specificity. *Mol. Cell* **76**, 423-436.e3 (2019).
9. Orlando, D. A. *et al.* Quantitative ChIP-Seq normalization reveals global modulation of the epigenome. *Cell Rep.* **9**, 1163–1170 (2014).
10. Zhang, Y. *et al.* Model-based Analysis of ChIP-Seq (MACS). *Genome Biol.* **9**, R137 (2008).
11. Kawaji, H., Kasukawa, T., Forrest, A., Carninci, P. & Hayashizaki, Y. The FANTOM5 collection, a data series underpinning mammalian transcriptome atlases in diverse cell types. *Sci. Data* **4**, 2016–2018 (2017).
12. Glancy, E. *et al.* PRC2.1- and PRC2.2-specific accessory proteins drive recruitment of different forms of canonical PRC1. *Mol. Cell* **83**, 1393-1411.e7 (2023).
13. Poepsel, S., Kasinath, V. & Nogales, E. Cryo-EM structures of PRC2 simultaneously engaged with two functionally distinct nucleosomes. *Nat. Struct. Mol. Biol.* 2018 252 **25**, 154–162 (2018).
14. Kasinath, V. *et al.* JARID2 and AEBP2 regulate PRC2 in the presence of

- H2AK119ub1 and other histone modifications. *Science* (80-.). **371**, (2021).
15. Sauer, P. V. *et al.* Activation of automethylated PRC2 by dimerization on chromatin. *bioRxiv* 2023.10.12.562141 (2023). doi:10.1101/2023.10.12.562141
 16. Finogenova, K. *et al.* Structural basis for PRC2 decoding of active histone methylation marks H3K36me2/3. *Elife* **9**, 1–30 (2020).
 17. Grijzenhout, A. *et al.* Functional analysis of AEBP2, a PRC2 Polycomb protein, reveals a Trithorax phenotype in embryonic development and in ESCs. *Development* **143**, 2716–2723 (2016).
 18. Owen, B. M. & Davidovich, C. DNA binding by polycomb-group proteins: searching for the link to CpG islands. *Nucleic Acids Res.* **50**, 4813–4839 (2022).

We sincerely thank the reviewers for their time and valuable feedback, which have greatly improved our manuscript.

Reviewer #1

In the revised work “Auto-inhibition of PRC2 by the broadly expressed long isoform of AEBP2,” Davidovich, Bracken and colleagues investigate the biochemical and functional differences between two isoforms of the PRC2 accessory factor AEBP2, a short form AEBP2S whose expression is mainly restricted to early embryonic development, and a long form AEBP2L expressed throughout life. They find that AEBP2S specifically promotes PRC2 binding to DNA, and is required for de novo repression of target genes upon differentiation from naïve to primed pluripotency, while AEBP2L inhibits the DNA binding and enzymatic activity of PRC2 in vitro and PRC2 binding to target genes in cells (as indicated by the title). Combining cryo-electron microscopy and biochemical assays, they ascribe the inhibitory function of AEBP2L to its recently evolved, negatively charged N-terminus, which is absent in AEBP2S.

The manuscript, whose first version was quite preliminary, has been substantially revised with new experimental data. While more thorough, the new experiments give a rather unclear picture with data that are difficult to conciliate and some effects that are unexpectedly subtle. This raises some doubts about the details of the mechanism. We thank Reviewer 1 for their thorough review of our revised manuscript and for recognising the significant revisions and new experimental data. We also appreciate their feedback about the need to more clearly integrate and explain some of the new data, as well as some of the more subtle effects observed. While these effects are expected, we have further clarified their significance in the revised manuscript. Below, we respond to each of the Reviewer’s comments point by point, addressing any ambiguities and providing additional context where needed.

Specific comments:

1. Lines 123-126: the argument is misleading. The cells described in Ref 29 cannot “lack” the full-length isoform of JARID2 since the short isoform is a cleaved product of the long one.

To address this point, we revised the corresponding lines 125-130 in the revised manuscript. The edited text now reads: “*It is important to note that the PRC2.2 complex containing AEBP2, but lacking JARID2, remains biologically relevant. This is partly because in cell types such as lymphocytes, keratinocytes, and endothelial cells, JARID2 is cleaved into a shorter isoform that cannot bind to PRC2.²⁹ Additionally, in line with the variable expression of JARID2 across cell types, endogenous proteomics of PRC2 in human diploid fibroblasts detected AEBP2, but not JARID2³⁰*”. The latter McCole et al. reference refers to our recent IP-MS of EED, EZH1 and EZH2 in human diploid fibroblasts, which showed the presence of PRC2-AEBP2 complex without JARID2 in these cells. The EED IP-MS from this paper is shown below in Reviewer Fig. 1.

Endogenous EED IP-MS in human diploid fibroblasts

Reviewer Fig. 1. IP-MS data from our separate study shows that JARID2 does not immunoprecipitate with EED in a proliferating (cycling) human diploid fibroblast (HDF) cell line¹.

Furthermore, we also provide an additional unpublished experiment below to further highlight the intriguing fact that AEBP2 can exist within PRC2.2 without JARID2 in some other cell types. Below, we performed endogenous IP-MS of both SUZ12 and AEBP2 on the WSU-DLCL2 Lymphoma cell line (Reviewer Figure 2). This shows that the JARID2 protein is not detected in IP-MS of SUZ12 and AEBP2:

Reviewer Fig. 2. Unpublished IP-MS data from a separate study showing JARID2 is not present in immunoprecipitations of either endogenous SUZ12 or AEBP2 in the WSU-DLCL2 Diffuse large B-cell lymphoma cell line.

Collectively, these lines of evidence strongly indicate that a PRC2-AEBP2 complex lacking JARID2 is present in various biological contexts.

2. Line 156: On what basis do the authors state that the ectopically expressed AEBP2 proteins are “2-4-fold above the endogenous levels”? Since they used chemiluminescence, which produces a non-linear signal, it is not possible to determine relative quantities with such precision. Besides, AEBP2-KO and OE appear to modulate JARID2 protein accumulation. This has to be mentioned in the text and discussed as it could contribute to the phenotypes.

This is true — the fold increase mentioned was an approximate estimate, and as the Reviewer correctly notes, chemiluminescence generates a non-linear signal. Therefore, to address this, we removed “2-4 fold” from the relevant sentence (now lines 166-168), which now reads as “We then ectopically expressed either human AEBP2^l(iso1) or AEBP2^s in Aebp2 KO ESCs (Fig. 2a) and confirmed their comparable protein levels by Western blot, albeit above the endogenous levels (Fig. 2b)”. In terms of potential modulations of JARID2 levels, we agree that this is important to consider. Therefore, we thoroughly examined all our Western blots of JARID2 across multiple similar ectopic experiments, as well as AEBP2^s and AEBP2^l knockout experiments. However, we found

no consistent trend in JARID2 protein levels. Therefore, any minor variations in JARID2 protein levels observed between experiments lacked a consistent trend, can be attributed to random variation, and therefore are unlikely to contribute to a phenotype across the manuscript, neither substantially nor consistently.

3. Figure 2f and g: It is troublesome that most of the effect of AEBP2 OE occurs at the intergenic region and not at the PcG promoters where the effect shown is modest. This should be discussed.

We thank Reviewer 1 for bringing this to our attention. As they know, PRC2 binding primarily occurs at CpG islands at Polycomb-enriched genes rather than at intergenic regions. This is shown in our ChIP-qPCRs of SUZ12 in Extended Data Figure 2d (shown below). While SUZ12 is enriched at Polycomb-bound CpG islands, near the *Tlx1* and *T* promoters, no detectable SUZ12 binding was identified at an intergenic region near the *Hoxb13* gene, consistent with previous genome-wide ChIP-based studies of PRC2 binding in mESCs^{2,3}:

Therefore, to avoid any misinterpretation in Figure 2f, we adjusted the Y-axis scale for the plot of SUZ12 binding at intergenic regions to match the scale as shown for SUZ12 binding at Polycomb bound promoters (please see below). Importantly, this is the same scaling approach we used in Figure 2c for the AEBP2 ChIP-Rx. As shown below, this correction confirms again that most of the SUZ12/PRC2 binding occurs at Polycomb target genes, and not at intergenic regions, which are close to background levels. This correction underscores the importance of the SUZ12 binding at Polycomb target promoter sites, which varies depending on the AEBP2 isoform used for rescue:

4. The authors' comments on the Figure 2 are confusing: on the one hand they make strong claims regarding the role of AEBP2 isoforms in PRC2 recruitment and on the other

hand they acknowledge that the level of overexpression, which could not be evaluated in the previous version since they did provide the WT control, “(...) could lead to artifacts in assays of H3K27me3 and transcription”. Either they trust their model and should provide H3K27me3 ChIP-seq or they don’t and should remove the figure altogether.

We appreciate Reviewer 1’s feedback on Figure 2 and understand why the original explanation might have been confusing. To clarify, we modified and extended the explanation of Figure 2 in the relevant results section (lines 192-200) as follows: “Taken together, these data **suggest that AEBP2^S and AEBP2^L perform opposing functions in regulating PRC2 localisation on chromatin: ectopic expression of AEBP2^S increases the chromatin occupancy of PRC2, whereas AEBP2^L reduces it. However, while this assay is valuable for investigating the relative chromatin binding abilities of AEBP2 isoforms in cells, the striking opposite effects we see are likely exaggerated due to their overexpression. Therefore, to explore the effects on H3K27me3 deposition or transcription, we instead focused on developing a system to specifically knock out each isoform and study their individual functions (Figures 3-4)**”. In short, the data presented in Figure 2 are informative on the relative chromatin binding ability of AEBP2 isoforms and therefore highly valuable to the paper. However, we agree that the above additional explanation (in red) should help provide greater clarity and explanation as to why we advance to the knockout strategy in Figures 3 and 4.

To illustrate why overexpression of AEBP2 isoforms could cause artifacts in terms of monitoring H3K27me3 levels, we show below an early experiment in which we monitored H3K27me3 at Polycomb target genes and control sites (Reviewer Figure 3). This shows that while H3K27me3 levels increase slightly on Polycomb target genes (*Six3* and *Pou4f3*) in *AEBP2* KO ESCs (as shown in Figure 4 and reported previously^{6,7}), its levels were reduced upon overexpression of either AEBP2 isoform. While we didn’t further explore this, these H3K27me3 reductions likely occur because both ectopically expressed AEBP2 isoforms outcompete Polycomb-like protein MTF2 for binding to SUZ12. Supporting this, PRC2.1 has been reported to drive more H3K27me3 deposition than PRC2.2 in mESCs^{2,5}.

Reviewer Fig. 3. ChIP-qPCR analysis of WT, *Aebp2* KO, *Aebp2* KO + L, and *Aebp2* KO + S, generated in the C129 ESC background, using the indicated antibodies.

Taken all together, the results in Figure 2 show that ectopic AEBP2^S binds to Polycomb target genes more effectively than AEBP2^L, which supports our initial *in vitro* findings in Figure 1, where AEBP2^S-PRC2 exhibited stronger DNA binding than AEBP2^L-PRC2. The results in Figure 2 also show that overexpression of AEBP2^L can partially displace PRC2, which is particularly significant as it is the direct opposite of the phenotype observed when we later specifically knockout AEBP2^L in ESCs (Figure 4). Importantly, when we take the results of Figures 2 and 4 together, we are later in the paper able to propose a model

in which AEBP2^L impairs PRC2 binding to Polycomb target genes, potentially by reducing the pool of core PRC2 available to interact with other accessory proteins, such as MTF2, which we and others previously established are more effective at targeting PRC2 to Polycomb target genes³⁻⁵.

5. The single knockout of AEBP2S in mouse ES cells is not complete, as can be seen in Figure 3g and Extended Data Figure 4c. This does not undermine the message, since the effects of the SKO are clear, and one can reasonably assume that they would be even more pronounced if the knockout were complete. But it should be acknowledged somewhere. Also, a shorter exposure of Figure 3g top should be provided to evaluate how much AEBP2S is stabilized in LKO as suggested by Extended Figure 5b.

To address this, we provided quantifications of the levels of each isoform for 3 independent replicates. This shows that the residual AEBP2^S was down to about 11% (Extended Data Fig. 4a):

As requested, to acknowledge this, we edited the following line to the results section (lines 234-236) accordingly: “We confirmed that the complete knockout of AEBP2S or AEBP2^L, while AEBP2^S was reduced by approximately 89%, with the remaining AEBP2^S likely originating from the P2 or P3 promoters (Fig. 3g and Extended Data Fig. 4a). As requested, we replaced the AEBP2 Western blot at the top of Figure 3g with the shorter exposure, as shown here:

6. Lines 255 to 258: How do the authors conciliate their observation regarding the JARID2-MTF2 interaction in IP-WB (Extended Data Figure 4e) and the IP-mass spec (Figure 3j, MTF2 is not clearly co-IP'd in any condition)?

We believe this is a ‘false negative’ due to a weaker batch of JARID2 antibody used in the IP-MS experiment in Figure 3j. Supporting this, subsequent IP-MS of endogenous JARID2 and exogenous FLAG-JARID2 both detected the MTF2 protein, as shown below:

Reviewer Fig. 4. Stoichiometry of PRC2 proteins relative to SUZ12 from IP-MS of endogenous JARID2 and ectopically expressed FLAG-tagged JARID2-FL, based on IgG-adjusted iBAQ scores. Error bars represent the standard deviation from three biological replicates.

Based on this, we believe that MTF2 is associated in a hybrid PRC2 complex with JARID2, as first reported by Brockdorff in 2016⁶. They reported that this hybrid MTF2-JARID2-PRC2 complex increases in the absence of AEBP2. While not the focus of this paper, our Western blot analysis in Extended Data Figure 4e could indicate that it's the loss of AEBP2^s that leads to increased MTF2-JARID2-PRC2. However, we believe this would require additional characterisation. Therefore, while interesting, we do not place emphasis on this observation in the paper.

7. The statement in lines 278-280 describing AEBP2S as “the only isoform that functions in de novo repression of Polycomb target genes during early ESC to EpiLC differentiation” is too categorical. First the Western blot shown in Extended Data Figure 5b suggests that the effect of each KO on AEBP2 isoforms is more complicated than described by the authors. Specifically, in EpiLC SKO, both isoforms are destabilized whereas in LKO, the short isoform is stabilized. Besides, genes that are upregulated in LKO EpiLCs compared to WT are also upregulated in the full knockout, showing that loss of AEBP2L can impair a small subset of the de novo repression events and that AEBP2S is not “the only isoform that functions” in this process. A figure showing the relative levels of the different class of DEGs in all four conditions would have been more informative to rule out any threshold effect.

To address the first point, we revised the statement in lines 293-296 to “Taken together, these data suggest that AEBP2^s is the *only main AEBP2 isoform that functions in contributing, like the Polycomb-like and JARID2 accessory proteins, to de novo repression of Polycomb target genes during early ESC to EpiLC differentiation*”. To address the second point regarding potential ‘threshold effects’, we were careful to highlight in the Figure legends of the revised manuscript that the RNA-seq data was generated from three independent biological replicates. Furthermore, as requested, we replaced the previous Extended Data Figure 5d with a new panel (new Extended Data

Figure 5c). As shown below, this new panel shows the relative levels of all significantly differentially expressed genes in all four conditions:

New Extended Figure 5c. This shows that the mRNA level changes resulting from the loss of specific AEBP2 isoforms. This shows that the SKO and *Aebp2* KO EpiLCs exhibited a greater number of differentially expressed genes compared to WT and LKO EpiLCs

Finally, we also added a new Supplementary Table 4 containing the list of the genes represented in the above panel – that is, all significantly differentially expressed genes between the four EpiLC lines:

Significantly differentially expressed genes in EpiLC (D2) between condition/WT; Related to Extended Figure 5c		
LKO/WT	SKO/WT	Aebp2 KO/WT
Ddx3y	Al662270	Al662270
Eif2s3y	Ahnak2	Aebp2
Serpina3m	C130073F10Rik	C130073F10Rik
Uty	Cebpg	Ddit3
Zfy1	Chac1	Ddx3y
	Dcaf12l1	Eif2s3y
	Ddit3	Fbxo15
	Ddx3y	Fgf5
	Dyrk4	Gjb3
	Eif2s3y	Gm11427
	Fbxo15	Gm50147
	Gm11427	Gm525
	Gm50147	H2-Eb1
	Gm525	Klf10
	Grb10	Ly6a
	Gstt2	Mfge8
	H2-Eb1	Mme
	Hrk	Neat1
	Hspb6	Nqo1
	Id3	Nr5a2
	Kbtbd11	Pim2
	Khdc3	Pla2g12a
	Mef2b	Plekha4
	Mocos	Rhbg
	Nr0b1	Slc6a8
	Nup62cl	Sstr5
	Phlda1	Steap3
	Pim2	Tcl1
	Ppp1r1a	Tdh
	Prss42	Uty
	Slc12a8	Zfy1
	Slc35d3	
	Slc6a8	
	Spta1	
	Sstr5	
	Steap3	
	Tdh	
	Trib3	
	Trim66	
	Trio	
	Tsc22d3	
	Zfp42	

8. Lines 283 to 287: The authors' conclusion should be toned down considering the indirect effect of the isoform-specific KO mentioned above.

To address this point, we removed the text "*suggesting more AEBP2^S binds to Polycomb target genes compared to AEBP2^L*" at the end of the relevant line, which now read as follows (lines 299-303): "*Next, we evaluated the genome-wide localisation of AEBP2, SUZ12 and JARID2 by ChIP-seq/Rx and the deposition profiles of H3K27me3 by CUT&RUN in our Aebp2 knock-out ESC lines. The SKO ESCs displayed lower AEBP2 occupancy at Polycomb target genes, compared to LKO, ~~suggesting more AEBP2^S binds to Polycomb target genes compared to AEBP2^L~~ while the full Aebp2 KO had signal reduced to background levels (Fig. 4a-c and Extended Data Fig. 6a)*".

9. New Figure 4: The authors should explain how they conciliate the gene expression analysis in EpiLCs (LKO has a modest effect, and SKO is closest to full KO) and the ChIP-seq in ESCs (SKO for PRC2 core components and for H3K27me3 is close to WT whereas the LKO is close to full KO).

We understand why the original explanation could lead to a confusion. It is important to note that the gene sets used to evaluate the effects of our AEBP2 isoform knockouts on PcG binding in ESCs (Figure 4) and on gene repression in EpiLCs (Figure 3) do not overlap and therefore should not be directly compared. The data presented in Figure 4 represents transcriptionally repressed Polycomb target genes in steady-state ESCs, whereas the data presented in Figure 3l-m represents active genes in ESCs that gain *de novo* PcG binding in EpiLCs (i.e. upon the differentiation of ESCs to EpiLCs). To help better clarify this important point in the manuscript, we modified the concluding sentence (lines 315-318) in the results section related to the discussion of Figure 4, as follows: "*Collectively, this data supports a model in which AEBP2^L, but not AEBP2^S, functions to antagonise the binding of PRC2 to Polycomb target genes in embryonic stem cells, thereby restricting the deposition of H3K27me3.*" We also modified line 519-520 in the Discussion section, as follows: "*Supporting this, the loss of AEBP2^L, but not loss of AEBP2^S, led to increased PRC2 binding on Polycomb target genes in ESCs.*"

10. An important shortcoming of the model remains that the acidic patches only partially explain the inhibitory effect of the AEBP2L N-terminus. In Figure 5g, for instance, charge reversal of the entirety of both acidic tracts does not result in the same level of HMTase activity as observed with AEBP2S. No other experiment shows AEBP2S and acidic tract mutants of AEBP2L in parallel. While the role of these acidic tracts is no longer exaggerated (thanks to the change on line 481 for example), there is still a passage that needs to be corrected. Lines 360-361 state that "the deletion of either acidic tract (mt1Δ and mt2Δ) was insufficient to fully alleviate the inhibitory activity of AEBP2L" but in fact the deletion of either acidic tract, on its own, was not sufficient to alleviate the inhibitory activity of AEBP2L at all, let alone fully. (These single deletions are indistinguishable from the intact AEBP2L in Figure 5g.) By the same token, on line 363, "the second acidic tract in AEBP2L is certainly required for inhibiting PRC2" rings false since its deletion has no effect. "Required" needs to be replaced with "involved." Finally, on line 365, "the positive effect seen for all the charge-swap mutations (m1K, mt2K and mt1K2K)" is also an overstatement. Only the double-charge-swap mutation (mt1K2K) shows an appreciable increase, and incidentally this is the only increase that the authors report as statistically significant.

We now moderated the relevant statement to address all the concerns made here (now lines 377-383; see below, changes marked in red): “The deletion of either acidic tract (*mt1Δ* and *mt2Δ*) was insufficient to **fully** alleviate the inhibitory activity of AEBP2^L (Fig 5g). This could point to functional redundancy between the two motifs or for the involvement of additional determinants in inhibiting PRC2. Yet, the second acidic tract in AEBP2^L is **certainly required for** involved in inhibiting PRC2, given that *mt2A* increased the activity of PRC2 by >2-fold. This result, together with the positive effect seen for **all** the charge-swap mutations (~~*mt1K*, *mt2K* and *mt1K2K*~~), point to electrostatic interactions as a determinant.”

11. The assays to test the substrate affinity of PRC2.2 containing different forms of AEBP2 use nucleosome core particles. While more physiologically relevant than naked DNA, this strategy is plagued by the limits acknowledged by the authors in their rebuttal (i.e., the affinity is too close the probe concentration to be measured accurately). We can certainly buy that the variety of interactions between PRC2 and the nucleosome is such that the inhibition of binding by the AEBP2L N-terminus will have only a subtle effect in this context. But the problem is that AEBP2S and AEBP2L are nearly indistinguishable, which means we can't really evaluate the impact of the acidic tract mutations on substrate binding. In this situation it may be worthwhile to use a less physiological binding assay, namely the free-DNA-binding assay presented in Figure 1d, as a tool to assess the importance of the acidic tracts, with the caveat that things might be different inside the cell. There is at least a clear range of behaviors from AEBP2S to AEBP2L, which would allow a compelling test of whether the weakened inhibition of PRC2 HMTase activity by acidic tract mutants is indeed reflected in a weakened inhibition of its binding to DNA.

We now carried out the requested binding assays using fluorescence anisotropy experiments with wild-type and mutant PRC2.2 complexes. The results reaffirmed our original conclusions: either deletion of the two acidic tracks in AEBP2 (PRC2-AEBP2^{Lmt1Δ2Δ}-JARID2), or mutation of them to alanine (PRC2-AEBP2^{Lmt1A2A}-JARID2) or lysines (PRC2-AEBP2^{Lmt1K2K}-JARID2), increase the apparent affinity for a 46 bp CpG DNA probe (DNA CpG46) by >15-fold *K_d*. See data in new Fig 6j:

This new experiment is now summarised in lines 445-451: “Fluorescence anisotropy binding assays indicated that either deletion of the two acidic tracks in AEBP2^L (PRC2-AEBP2^{Lmt1Δ2Δ}-JARID2), or mutating them to alanine residues (PRC2-AEBP2^{Lmt1A2A}-JARID2) or lysines (PRC2-AEBP2^{Lmt1K2K}-JARID2), increase the affinity of PRC2.2 for a 46 bp CpG DNA probe by >15-fold *K_d* (Fig. 6j and Extended Data Fig. 8h). These observations indicate that the acidic tracts of AEBP2^L inhibit the DNA binding activity of PRC2.2.”

12. Since the authors did not purify the newly formed complex after TEV digestion, the right control would have been PRC2-AEBP2L(TEV)-JARID2 + TEV inactivated (e.g., heat-inactivated) to rule out any indirect effect.

We now converted a PRC2-AEBP2^{L(TEV)}-JARID2 complex to PRC2-AEBP2^S-JARID2 by TEV cleavage and only then purified the newly formed PRC2-AEBP2^S-JARID2 using size exclusion chromatography to remove the TEV enzyme. The TEV-free PRC2-AEBP2^S-JARID2 complex is marked “S” in the new Extended Data Fig 8h:

We next used this TEV-free PRC2-AEBP2^S-JARID2 complex for DNA binding assays, side-by-side with the PRC2-AEBP2^{L(TEV)}-JARID2 complex (marked as “L(TEV)” in the new Extended Data Fig 8h, see above). Importantly, the results indicate that the TEV-free PRC2-AEBP2^S-JARID2 complex binds to DNA with >7-fold higher affinity than the PRC2-AEBP2^{L(TEV)}-JARID2 complex (see Fig. 6j, which is presented in our response for the previous point). This observation is in agreement with other results in our manuscript: it is attributed to an increment of the intrinsic DNA-binding activity of PRC2.2 after the removal of its N-terminal domain, rather than the presence of the TEV enzyme in the assay.

13. On a related point, the new artificial tethering assays provided in Extended Data Fig. 9 can be argued to support the authors’ model to the extent that AEBP2L doesn’t negatively regulate PRC2 activity when it is forced to bind chromatin, consistent with the idea that the effects documented earlier were due to inhibition of PRC2 recruitment to chromatin. However, the inclusion of the acidic tract mutants in this context is not very informative, since AEBP2L doesn’t exert a PRC2-inhibitory effect here. In fact, the impact of the acidic tract mutations is only ever tested in vitro using binding assays and HMTase assays. We therefore don’t formally know whether these mutations result in any phenotype in cells, and thus whether they reveal a biologically meaningful aspect of AEBP2L function. If overexpressed in AEBP2-KO cells, for instance, would the mutants fail to impair SUZ12 recruitment to the same extent as overexpressed unmodified AEBP2L?

We thank the Reviewer for this helpful comment. To address it, we expanded the text describing Extended Data Figure 9g to include this line (476-478): “*As expected, outside the tethered site, TetR-AEBP2^S showed stronger binding to two Polycomb target genes compared with TetR-AEBP2^L, while TetR-AEBP2^L mt1Δ2Δ exhibited intermediate binding*”. We hope that this line helps clarify that we tested the binding of the AEBP2^L mt1Δ2Δ in comparison to wild-type AEBP2^L and AEBP2^S at two separate Polycomb target genes. As

expected, the AEBP2^L mt1Δ2Δ mutant exhibited stronger binding to these targets than wild-type AEBP2^L, though still weaker than AEBP2^S:

Extended Data Figure 9g. ChIP-qPCR analysis shows relative enrichments of FLAG-TetR AEBP2 binding upstream and downstream of TetO (left) and at endogenous loci (right) in reporter ESCs. Shown are data of two independent experimental replicates.

14. An alternative interpretation to the tethering assays is that in a system without any indirect interference (e.g., indirect effect on the other isoform's stability), the two isoforms and the mutant behave in an identical manner. This does not really support of model where AEBP2^L has any antagonistic role on PRC2.

We respectfully disagree with this interpretation, as this is a system with a strong interference: TetR:TetO tethering. These tethering assays were designed to artificially localize AEBP2^S-PRC2 and AEBP2^L-PRC2 to chromatin at the TetO site, effectively bypassing or neutralising the influence of the N-terminal region of AEBP2^L. Specifically, the high affinity and specificity of the TetR fusion protein for the TetO array on the reporter drive equivalent chromatin targeting of various AEBP2-containing PRC2.2 complexes in this assay, regardless of their respective DNA-binding activities. In that sense, this experiment asks a simple question: How would AEBP2^L function if we forced it to bind chromatin? The data indicates that when AEBP2^L is artificially tethered to chromatin, it behaves similarly to AEBP2^S. This is in agreement with our model, that the N-terminus of AEBP2^L antagonises AEBP2^L-PRC2 interactions with chromatin.

Reviewer #2

The revised manuscript and rebuttal letter, I acknowledge that the authors have addressed most of the critical concerns raised of their initial manuscript. The additions of new figures, specifically Figures 4 and 6, significantly improve the overall manuscript by providing further insights into the distinct roles of AEBP2L and AEBP2S isoforms in PRC2 regulation.

The inclusion of new experiments involving loss of AEBP2L and AEBP2S (Figure 4) as well as functional assays (Figure 6) investigating the N-terminal region of AEBP2L in regulating PRC2 activity are improvements. These new data support the authors' claims that AEBP2L antagonizes PRC2 activity, specifically by inhibiting its interaction with chromatin and HMTase activity.

The data now provide stronger evidence for the distinct roles of AEBP2 isoforms. The revised manuscript effectively shows that AEBP2S is implicated in de novo gene repression during early development, while AEBP2L, prevalent in most tissues during later stages, antagonizes PRC2 chromatin-binding ability and subsequent H3K27me3 deposition. The experiments using endogenous knockout of AEBP2S and AEBP2L (Figure 4) demonstrate that the loss of AEBP2L increases PRC2 binding and H3K27me3 deposition on Polycomb target genes, reinforcing the model of AEBP2L as a negative regulator of PRC2 activity.

The revised experiments involving PRC2-AEBP2-JARID2 complexes (Figure 6) effectively address the concerns regarding the role of JARID2 in modulating PRC2 activity. The introduction of a cleavable AEBP2L(TEV) construct is an innovative solution to demonstrate how the N-terminal region of AEBP2L affects HMTase activity and chromatin binding. The authors now provide critical H3K27me3 ChIP-seq and ChIP-qPCR data (Figures 2 and 4), including controls from WT cells, which validate the differences in chromatin binding between AEBP2S and AEBP2L. The inclusion of ESC lines (knockouts of AEBP2S, AEBP2L, or both isoforms) enhances the rigor of the conclusions.

There are still some limitations that should be addressed or at least acknowledged. While the authors have explored how AEBP2L and AEBP2S differentially regulate PRC2 interaction with JARID2, there remains limited exploration of how these isoforms influence interactions with other regulatory proteins or non-coding RNAs. The suggestion that AEBP2L might interfere with accessory subunit binding is intriguing but requires more direct evidence. This could be pursued in future studies. While the authors speculate on the potential impact of AEBP2 isoforms on PRC1 recruitment, this remains an important avenue that has not been directly addressed. The absence of experiments probing PRC1 recruitment in response to AEBP2 knockout limits our understanding of how PRC2 regulation by AEBP2L and AEBP2S influences the broader Polycomb repression.

We revised the manuscript to incorporate a discussion of these important points. For example, in the results section and discussion, we cited previous work which showed that AEBP2 and Polycomb-like proteins compete for binding to the same C2 region in SUZ12^{8,9}. As suggested, we modified and extended lines 564-568 of the Discussion to

indicate that further work would be required to explore how AEBP2 isoforms might differentially regulate the abundance of a hybrid PRC2 containing JARID2 and MTF2. This reads now as “*More broadly, while further research is needed to understand the functions of hybrid complexes containing JARID2 and MTF2, our findings highlight the importance of distinguishing PRC2 subtypes based on subunit isoforms, rather than solely relying on the previously established classification of PRC2.1 and PRC2.2⁶.*”

Furthermore, while we agree that there is a need for a comprehensive evaluation of the different forms of cPRC1 in ESCs in cells that lack the various AEBP2 isoforms, we feel that this goes beyond the scope of this study. We acknowledge this in the revised discussion with this line (576-580) predicting the consequences of loss of the AEBP2^L isoform in mice: “*While this would require further investigation, we speculate that loss of only the AEBP2^L isoform would lead to increased H3K27me3 and consequent cPRC1-mediated repression of Polycomb target genes, thereby phenocopying causing the Trithorax phenotype observed in Aebp2 KO mice*”.

The statistical analysis presented in the revised manuscript appears adequate, with appropriate use of replicates and statistical tests. But it would be helpful to see more extensive analysis of the differentiation experiments, particularly regarding the role of AEBP2S in the repression of key developmental genes during ESC to EpiLC differentiation. The current data (Figure 3) provide valuable insights, but additional replicates or more genes analyzed would strengthen the conclusions.

We thank the Reviewer for these suggestions. We were careful in the legends of Figure 3 and Extended Data Figure 5 in the revised manuscript to highlight that all mRNA data is generated from three independent biological replicates. In terms of expanding the analyses to additional genes, we utilised a set of 149 genes previously shown by Hayashi et al., 2011¹⁰ to be the most significantly differentially expressed during ESC to EpiLC differentiation. We analysed the expression of these genes in our knock-out cell lines and discovered that while the differences in their expression between mutant and WT cell lines were not statistically significant, their expression profiles were the most similar between WT and LKO, and between SKO and Aebp2 KO. We included this additional analysis in the revised Extended Figure 5d-e (shown below), and updated the manuscript text (lines 283-287) accordingly, as follows: “*We compared the expression of 149 genes previously identified as most differentially expressed during ESC to EpiLC differentiation¹⁰. This revealed a trend — though not statistically significant — of similar expression patterns between WT and LKO, and between SKO and Aebp2 KO (Extended Data Fig. 5d-e).*”

d

e

Revised Extended Figure 5d-e.

Importantly, we write immediately after this referring to Figures 3l-m: *"Importantly, to assess the direct consequences of AEBP2 isoform loss, we focused on a set of 398 genes that we recently identified to be repressed during ESC to EpiLC differentiation, with de novo PRC2 recruitment and partial dependence on the Polycomb-like and JARID2 accessory proteins³³. While expression of these genes remained high and unchanged in undifferentiated ESCs (Extended Data Fig. 5f), they were less repressed during EpiLC differentiation in SKO and Aebp2 KO, but not LKO, compared to WT (Fig. 3l-m). Taken together, these data suggest that AEBP2^S is the main AEBP2 isoform contributing, like the Polycomb-like and JARID2 accessory proteins, to de novo repression of Polycomb target genes during ESC to EpiLC differentiation"*.

Figure 3k-m.

The revised manuscript advances our understanding how AEBP2L and AEBP2S regulate PRC2 activity. The authors have successfully addressed most of the concerns raised in the initial review, and I believe the manuscript is now suitable for publication with minor revisions. However, some mechanistic insights remain to be further developed, and the authors should acknowledge these limitations in the discussion.

We sincerely appreciate the Reviewer's time and thoughtful feedback throughout this process. As described above, in the discussion of the revised manuscript, we have acknowledged its limitations and emphasised the remaining mechanistic questions that remain to be further developed.

Reviewer #3

The authors have addressed most of my previous comments. While the revised manuscript shows significant improvement, I found the results presented in the new Figure 4 to be relatively weak. Additionally, although the revised version provides a better characterization of EpiLC differentiation, the manuscript still falls short in demonstrating the broader relevance of these findings beyond ESCs.

We thank Reviewer 3 for their helpful feedback and for acknowledging the improvements made in the revised manuscript. We appreciate that the increases in PRC2 binding observed in ESCs lacking AEBP2^L are moderate in Figure 4. However, we'd like to point out that the increases in SUZ12 and H3K27me3 replicate the consequences of complete loss of AEBP2, as reported originally in 2016 by the Brockdorff and Cooper lab⁶. Importantly, these relatively small increases in PRC2 binding and H3K27me3 deposition were associated with the homeotic transformation characteristic of a Trithorax loss of function phenotype in mice. To address Reviewer 3's important suggestion to better highlight the broader relevance of our findings beyond ESCs, we revised the Discussion of the revised manuscript. Lines 570-576 now read as follows: *“Our work also helps explain the unexpected phenotype of Aebp2 KO mice, which exhibit a Trithorax skeletal phenotype and die perinatally²³. We propose a model in which the antagonistic action of AEBP2^L functions to limit the activity of PRC2 in somatic cells. Importantly, our findings underscore that even moderate increases in PRC2 binding and H3K27me3 deposition at Polycomb target genes, as observed in ESCs lacking AEBP2^L, can have significant biological consequences, consistent with Trithorax loss-of-function phenotypes observed in vivo”*. We also added one and a half lines to the revised paper's abstract to again emphasise the impact and broader relevance of our findings beyond ESCs. One new line (43-45) reads: *“Contrary to prior assumptions that AEBP2 enhances PRC2 function, we find that the widely expressed AEBP2^L isoform inhibits it.”*

Rebuttal letter references:

1. McCole, R. *et al.* A conserved switch to less catalytically active Polycomb repressive complexes in non-dividing cells. *Cell Rep.* **44**, (2025).
2. Healy, E. *et al.* PRC2.1 and PRC2.2 Synergize to Coordinate H3K27 Trimethylation. *Mol. Cell* **76**, 437-452.e6 (2019).
3. Højfeldt, J. W. *et al.* Non-core Subunits of the PRC2 Complex Are Collectively Required for Its Target-Site Specificity. *Mol. Cell* **76**, 423-436.e3 (2019).
4. Lee, C. H. *et al.* Distinct Stimulatory Mechanisms Regulate the Catalytic Activity of Polycomb Repressive Complex 2. *Mol. Cell* **70**, 435-448.e5 (2018).
5. Glancy, E. *et al.* PRC2.1- and PRC2.2-specific accessory proteins drive recruitment of different forms of canonical PRC1. *Mol. Cell* **83**, 1393-1411.e7 (2023).
6. Grijzenhout, A. *et al.* Functional analysis of AEBP2, a PRC2 Polycomb protein, reveals a Trithorax phenotype in embryonic development and in ESCs. *Development* **143**, 2716–2723 (2016).
7. Conway, E. *et al.* A Family of Vertebrate-Specific Polycombs Encoded by the LCOR/LCORL Genes Balance PRC2 Subtype Activities. *Mol. Cell* (2018). doi:10.1016/j.molcel.2018.03.005
8. Chen, S., Jiao, L., Liu, X., Yang, X. & Liu, X. A Dimeric Structural Scaffold for PRC2-PCL Targeting to CpG Island Chromatin. *Mol. Cell* **77**, 1265-1278.e7 (2020).
9. Chen, S., Jiao, L., Shubbar, M., Yang, X. & Liu, X. Unique Structural Platforms of Suz12 Dictate Distinct Classes of PRC2 for Chromatin Binding. *Mol. Cell* (2018). doi:10.1016/j.molcel.2018.01.039
10. Hayashi, K., Ohta, H., Kurimoto, K., Aramaki, S. & Saitou, M. Reconstitution of the mouse germ cell specification pathway in culture by pluripotent stem cells. *Cell* **146**, 519–532 (2011).

Rebuttal:

We raised a set of concerns regarding the previously revised manuscript “Auto-inhibition of PRC2 by the broadly expressed long isoform of AEBP2.” The authors have attempted to address these concerns, but we remain unconvinced that the data support the principal claim of the paper, namely that “alternative isoforms of its accessory subunit AEBP2, namely AEBP2S (short) and AEBP2L (long), perform opposite functions in modulating PRC2 activity.”

We thank Reviewer 1 for their comments. However, as detailed below, their remaining technical criticisms are all categorically wrong and do not affect the conclusions of our study. In light of this and given that the two other reviewers already recommend publication in our journal, we respectfully request to reconsider the editorial decision. We believe that at an editorial overruling, or minimally an arbitration, would be appropriate in this exceptional case.

Below, we provide a point-by-point response addressing the multiple factual flaws and misrepresentations in the remaining technical criticisms made by Reviewer 1:

1) We maintain that the ectopic overexpression experiments do not allow any robust conclusion. AEBP2 L and S are highly overexpressed as compared to the endogenous level. The impact on SUZ12 recruitment of overexpressing the different AEBP2 variants is identical throughout the genome which raises the question of whether this is a true effect or a normalization artifact.

This statement is factually wrong on both counts.

Firstly, the effect of the AEBP2 isoforms on SUZ12 binding are not identical throughout the genome. Instead, our spike-in ChIP-Rx data show that AEBP2^L decreases SUZ12 (representative of core PRC2) binding at Polycomb target genes (Figure 2e-f), across an entire chromosome (Figure 2g) and throughout the genome (Extended Data Figure 2f), whereas AEBP2^S increases it. These data demonstrating clear, opposing effects, as shown below:

Secondly, the reduced binding of SUZ12 (representative of core PRC2) at PcG bound promoters upon ectopic expression of AEBP2^L was independently validated by ChIP-qPCR, confirming it is a true effect and ruling out any normalisation artifacts, as shown by representative ChIP-qPCRs below (from Extended Data Figure 2d):

Note that the ChIP-qPCR method was not capable of detecting SUZ12 enrichment at an intergenic region near the *Hoxb13* gene, as expected. This is because PRC2 binding is known to be much lower at intergenic regions than at PRC2-target sites^{1,2}. Instead, as previously shown^{1,2}, one needs quantitative ChIP-seq (ChIP-Rx) to detect low level signal representative of PRC2 binding at intergenic regions.

In conclusion, the orthogonal and complimentary approaches of ChIP-Rx and ChIP-qPCR provide robust and reproducible evidence that the two AEBP2 isoforms differentially regulate SUZ12 (PRC2) binding. Hence, the above reviewer statement that “*The impact on SUZ12 recruitment of overexpressing the different AEBP2 variants is identical throughout the genome*” **is categorically false.**

Rather than acknowledging this possibility, the authors now adjust the scale of the intergenic signal, which obscures the differences between conditions.

This comment constitutes a clear misrepresentation and a fundamentally flawed technical criticism. Below is the representation of Figure 2f in its original form, before we adjusted the scale of the intergenic regions (left). Side-by-side, we included here also the same figure in its current form, where the scale of the intergenic regions has been adjusted to the same scale as PcG promoters (right). This comparison shows that even if we did not adjust the scale, the original representation (on the left) would support the observation that ectopic expression of AEBP2^S leads to increased SUZ12 binding at Polycomb target genes and intergenic regions. Conversely, AEBP2^L has the opposite effect, by reducing SUZ12 binding at Polycomb target genes and also at intergenic regions:

Panel 2f. Original version (on the left) and revised version (on the right).

Obviously, if one uses a Y-axis of ~20-fold lower range (0-15 on the right versus 0-0.6 on the left), the changes in SUZ12 occupancy might seem more dramatic at intergenic regions than at Polycomb (PcG) target promoters. Reviewer 1 asked us to discuss this in the previous revision round. We did it: we explained that the same Y-axis for PcG promoters and intergenic regions would demonstrate that SUZ12 is predominantly enriched at PcG promoters (see response to point 3 of Reviewer 1 in the previous round). From comparing the above panels, one can see that SUZ12 occupancy is following the same trend at PcG promoters and intergenic regions, despite the expected variations in occupancy (i.e. higher occupancy in PcG promoters, which is expected). Importantly, it is wrong to imply that the representation is being “obscured”, as we are 100% transparent about the representation of SUZ12 genome distribution. Firstly, in addition to the above-mentioned panels, we also show that AEBP2^S increases SUZ12 binding across a whole chromosome 9 in Figure 2g, as shown below:

Figure 2g.

Secondly, we present the genome-wide enrichments of SUZ12 in the mutant cell lines relative to the wild-type ESCs in the Extended Data Figure 2e, as shown below:

Extended Data Figure 2e.

Again, the trend is consistent with the data discussed above: ectopic expression of AEBP2^S leads to increased SUZ12 binding across an entire chromosome (Figure 2g), and on average throughout the genome (Extended data Figure 2e). Conversely, AEBP2^L has the opposite effect.

In conclusion, the above statement by Reviewer 1, that “*The impact on SUZ12 recruitment of overexpressing the different AEBP2 variants is identical throughout the genome*”, **is factually wrong**.

If the intergenic signal were instead used for normalization, the differences at PcG promoters would likely disappear.

The technical basis for this statement is simply wrong: intergenic regions are never appropriate for normalization, especially when using the gold standard ChIP-Rx method³ (>550 citations). ChIP-Rx relies on spiking in chromatin from another species (in this case, human) to allow robust, genome-wide normalization. The suggestion to use of intergenic signal is irrelevant in ChIP-seq normalisations and shows a fundamental flawed understanding of how the method works.

Moreover, suggesting normalization to intergenic regions is not only flawed in principle but also problematic in practice. If there are real, albeit modest, changes in signal at these intergenic regions, normalization to them would systematically under-estimate the changes at peak binding sites (Polycomb target promoters in this case). The above reviewer's suggestion is especially inappropriate given that PRC2 has been shown to bind beyond classical promoter regions through more transient, lower-affinity interactions^{1,2}. These weak intergenic interactions are detectable by ChIP-seq, ChIP-Rx and CUT&RUN-Rx, but occur at lower signal levels, making intergenic signal inherently unreliable as a normalization reference.

Crucially, we validated the ChIP-Rx findings using an independent approach—quantitative ChIP-qPCR (Extended Data Figure 2d), a method which does not rely on any spike-in normalization. As shown and discussed above, both approaches show consistent, opposing effects of the AEBP2 isoforms, further reinforcing the robustness and validity of our conclusions.

Whatever impact AEBP2 may have on SUZ12 (setting aside the doubts just mentioned), it does not translate into any effect on H3K27me3 deposition.

This statement is wrong and misrepresents the data in the paper. The relevant data in Figure 4a and 4c clearly show that knockout of AEBP2^L (LKO), but not AEBP2^S (SKO), leads to an about a two-fold increase in SUZ12 binding at Polycomb target genes. This increase is accompanied by a corresponding increase in H3K27me3 deposition at the same Polycomb bound promoters, as shown here:

Relevant data from Figure 4a.

The above findings directly contradict the Reviewer’s claim and demonstrate that the impact of AEBP2^L loss on SUZ12 does, in fact, translate into a measurable and biologically meaningful effect on H3K27me3 levels. Importantly, the above increases in SUZ12 and in H3K27me3 in the ESCs lacking AEBP2^L (LKO) were reproduced across three independent biological replicates and statistical significance are presented in Figure 4c, as shown below:

Relevant data from Figure 4c.

Importantly, these data very clearly establish that it is the loss of AEBP2^L, not AEBP2^S, that phenocopies the previously reported⁴⁻⁷ increases in PRC2 binding and H3K27me3 upon complete loss of AEBP2 (both isoforms). As we point out in the discussion (lines 570-580), these statistically significant SUZ12 and H3K27me3 increases are also biologically important: *“Our work also helps explain the unexpected phenotype of Aebp2 KO mice, which exhibit a Trithorax skeletal phenotype and die perinatally. We propose a model in which the antagonistic action of AEBP2^L functions to limit the activity of PRC2 in somatic cells. Importantly, our findings underscore that even moderate increases in PRC2 binding and H3K27me3 deposition at Polycomb target genes, as observed in ESCs lacking AEBP2^L, can have significant biological consequences, consistent with Trithorax loss-of-function phenotypes observed in vivo. While this would require further investigation, we speculate that loss of only the AEBP2^L isoform would lead to increased H3K27me3 and consequent cPRC1-mediated repression of Polycomb target genes, thereby phenocopying the Trithorax phenotype observed in Aebp2 KO mice (Fig. 7)”*.

In summary, our data above demonstrate robustly that AEBP2^L normally acts to restrain PRC2 activity and limit H3K27me3 deposition. The inhibitory role of AEBP2^L is a central finding of our study, and the data in Figure 4 provide direct and quantitative support for it.

It is actually the opposite, as is eventually acknowledged by the authors in the most recent rebuttal: *“to illustrate why overexpression of AEBP2 isoforms could increase artifacts...”* The authors go on to note that the two isoforms have a similar impact on H3K27me3 levels, which strongly undermines the central plank of their model.

This misrepresents our work and what we clearly explained both in the manuscript and in response to Point 4 of reviewer 1 in the previous round of review. Contrary to the above argument, our manuscript clearly emphasised that the overexpression assay *“is valuable for investigating the relative chromatin binding abilities of AEBP2 isoforms in cells”* (L196).

We acknowledged that “the striking opposite effects we see [on the chromatin occupancy of SUZ12] are likely exaggerated due to their overexpression” (L197). This was the rationale to carry out the next experiments in Figure 4, using the isoform-specific knockout cell lines: “Therefore, to explore the effects on H3K27me3 deposition or transcription, we instead focused on developing a system to specifically knock out each isoform and study their individual functions (Figures 3-4)” (L198). As said, the experiments in Figure 4 show increased PRC2 binding and H3K27me3 deposition upon the knockout of AEBP2^L (LKO), directly supporting our model that AEBP2^L inhibits PRC2 function.

To reiterate, the data in Figure 2 strongly support our model that AEBP2^S binds better than AEBP2^L throughout the genome, with the former increasing SUZ12 binding and the later decreasing SUZ12 binding. Additionally, in the previous revision, we clarified in the results section that the appropriate context for assessing effects on H3K27me3 is through knockout of the individual AEBP2 isoforms, compared to matched wild-type and full *Aebp2* knockout ESCs, as done in Figure 4, and shown above.

2) The selective knockouts used in Figure 3 are too leaky to allow any robust conclusion. This is wrong. The KO of AEBP2^L is not ‘leaky’ but rather leads to a 100% depletion of AEBP2^L. Please see immunoblotting in Fig 3g and quantifications of three independent biological replicates in Extended Data Figure 4a, as shown below:

Figure 3g and Extended Data Figure 4a.

The AEBP2^S KO, shown above, achieves an average of 89% reduction in ESCs (Extended Data Fig. 4a), and the remaining low expression level of AEBP2^S (11%) does not change the conclusions. In fact, even **Reviewer 1 previously acknowledged this** (from their point 5 of the past revision): “The single knockout of AEBP2S in mouse ES cells is not complete, as can be seen in Figure 3g and Extended Data Figure 4c. **This does not undermine the message, since the effects of the SKO are clear, and one can reasonably assume that they would be even more pronounced if the knockout were complete**”.

Importantly, contrary to the wrong statement made by Reviewer 1 above, these isoform-specific KO ESC lines enabled us to make robust and reliable conclusions. As shown in Figure 4, the loss of AEBP2^L—but not the loss of AEBP2^S—leads to increased SUZ12 binding

and H3K27me3 deposition at Polycomb target genes in ESCs. These findings are based on three independent biological replicates and are further supported by quantitative ChIP-qPCR analysis, which avoids complications from spike-in normalization, as described above, and featured again here for clarity:

Relevant data from Figure 4a.

Relevant data from Figure 4c.

Together, these robust datasets support one of the central conclusions of the paper: AEBP2^L uniquely limits PRC2 chromatin binding and activity, consistent with our characterisation of its N-terminal region in Figures 1, 2, 5, and 6.

The quantification now provided in Extended Data Figure 4a is inconsistent with the Figure 3g and Extended Data Figure 5b. In the LKO condition, AEBP2^S increases by much more than 17% in mESC at D0.

It is correct that AEBP2^S levels increase in the absence of AEBP2^L, and the magnitude of this increase varies between replicates. This variability likely reflects a biological effect. Specifically, it is likely that the knockout of AEBP2^L increases the availability of PRC2, which in turn stabilizes AEBP2^S. Therefore, the observed increase in AEBP2^S levels is an expected consequence of biological variability in this system.

One therefore cannot rule out that the modest effect reported in Figure 4 reflects a change in the global amount of AEBP2 (S+L) rather than perturbation of specific isoforms.

This statement is wrong on two levels:

Firstly, the reviewer mischaracterizes the effects observed in Figure 4 as “modest”. In fact, the increases in SUZ12 binding and H3K27me3 deposition upon AEBP2^L knockout (LKO) are statistically significant across three independent biological replicates (see Figures 4a and 4c, above). These are biologically meaningful changes, comparable in magnitude to those reported by multiple laboratories upon AEBP2 depletion⁴⁻⁷.

Secondly, the reviewer’s suggestion that the changes reflect altered global levels of total AEBP2 (S+L) rather than isoform-specific perturbation is logically flawed and inconsistent with the data. If global AEBP2 levels were the driving factor, then one would expect WT and LKO to behave similarly, since both highly express total AEBP2. On the same rationale, SKO should more closely resemble the full KO, given its reduced total AEBP2 levels. Yet, this is not what the data show. Instead, the phenotypes observed in LKO ESCs closely resemble those of the full AEBP2 knockout, while SKO ESCs do not. This pattern is clearly demonstrated in Figure 4 and Extended Data Figure 6 and is consistent with results across multiple independent experiments (see also Figures 1, 2, 5, and 6). Collectively, these results strongly support our conclusion that AEBP2^L acts via its unique N-terminal region as a negative regulator of PRC2 chromatin binding, thereby modulating H3K27me3 deposition.

In short, the data does not fit with and do not support the reviewer’s speculative interpretation, but instead robustly reinforce our central conclusion regarding isoform-specific function.

This alternative hypothesis is supported by the observation that the EpiLC transcriptomes for WT and LKO on the one hand, and SKO and Aebp2-KO on the other hand, are very similar, closely tracking the similarity in total AEBP2 protein (irrespective of isoform) within each of these respective pairs of samples.

This interpretation is highly implausible and inconsistent with the totality of the data presented in the manuscript. While WT and LKO do share higher overall levels of AEBP2 protein relative to SKO and full KO ESCs, this superficial correlation does not explain the transcriptional phenotypes observed. The modest impairment in the repression of *de novo* Polycomb target genes during ESC-to-EpiLC transition in the full AEBP2 knockout must, by definition, be mediated by loss of either one of the two isoforms or the combined loss of both isoforms. Crucially, Figures 1, 5, and 6 clearly demonstrate that AEBP2^S promotes PRC2 chromatin binding and H3K27me3 deposition, while AEBP2^L exerts the opposite effect, acting as a negative regulator. Thus, the reviewer’s hypothesis—treating total AEBP2 levels as functionally equivalent regardless of isoform—is not only less plausible but reflects a misinterpretation of the data across multiple Figures and an oversimplification or subjective judgment. In conclusion, our proposal that AEBP2^S, but not AEBP2^L, supports PRC2 function during ESC to EpiLC differentiation is the most parsimonious and scientifically supported interpretation.

3) The mass spectrometry provided in Figure 3i, 3j and Reviewer Figure 4 show too much variability to make any robust conclusion.

This is wrong. The mass spectrometry in Figures 3i has small error bars and is based on three biological replicates (SUZ12 IP-MS), as shown below:

Figure 3i.

As detailed in the results section (lines 258-267), these experiments support two clear and robust conclusions: (1) both AEBP2 isoforms (AEBP2^S and AEBP2^L) interact with SUZ12 and JARID2, and (2) JARID2 is still able to bind SUZ12 (i.e., assemble into PRC2) even in the absence of one or both AEBP2 isoforms, though this interaction is notably weakened. Importantly, these findings were further validated by independent EZH2 co-immunoprecipitation and immunoblotting experiments (Extended Data Figure 4d), as shown here:

Extended Data Fig. 4d.

To strengthen these observations and add another layer of validation, we also performed an additional IP-MS of JARID2, in Figure 3j, as shown below in comparison to the SUZ12 IP-MS of SUZ12:

Figure 3i-j.

While the JARID2 antibody is weaker, leading to greater error bars in the JARID2 IP-MS (Fig 3j), the same conclusions hold firm here: (1) both AEBP2 isoforms interact with SUZ12 and

JARID2, and (2) JARID2 can still bind SUZ12 (PRC2), even in the absence of one or both AEBP2 isoforms. Furthermore, these two robust conclusions were further validated in an independent JARID2 IP-immunoblotting experiment (Extended Data Figure 4e), as shown below:

Extended Data Fig. 4e.

With regards to Reviewer Figure 4, mentioned above by Reviewer 1, it was part of the previous response to Reviewer 1, Point 6, and contained data not used in the paper. This data included a JARID2 IP-MS (endogenous and FLAG-tagged) in wild-type cells. That experiment could not address whether (1) both AEBP2 isoforms interact with SUZ12 and JARID2, or, (2) whether JARID2 can still bind SUZ12 (PRC2) even in the absence of one or both AEBP2 isoforms. Instead, this Rebuttal Figure 4 was included previously to show additional evidence that MTF2 can co-IP with JARID2 even though it is not detectible in the relatively weak JARID2 IP-MS (Figure 3j). Importantly, the co-IP of MTF2 with JARID2 was previously reported by the Cooper lab (2016)⁴ and shown above in IP-immunoblotting (Extended Data Fig. 4e). As noted, in the previous response to Point 6 of Reviewer 1 “*While not the focus of this paper, our Western blot analysis in Extended Data Figure 4e could indicate that it's the loss of AEBP2^S that leads to increased MTF2-JARID2-PRC2. However, we believe this would require additional characterisation. Therefore, while interesting, we do not place emphasis on this observation in the paper*”.

In conclusion, Reviewer 1’s assertion about the mass spectrometry provided in Figure 3i-j showing “*too much variability to make any robust conclusion*” is entirely wrong. As clearly outlined above, our mass spectrometry data in Figure 3i-j are robust and reproducible, are in agreement with our independent co-IPs, and they strongly support our main conclusions: (1) both AEBP2 isoforms interact with SUZ12 and JARID2, and (2) JARID2 can still bind SUZ12 (PRC2) even in the absence of one or both AEBP2 isoforms, which are further validated by complementary IP-immunoblotting experiments (Extended Data Figure 4d-e). Importantly, these robust conclusions in Figure 3i-j directly informed the subsequent reconstitution of AEBP2^S- and AEBP2^L-containing PRC2 complexes with JARID2, as shown in Figure 6 and Extended Data Figure 8.

According to the exogenous JARID2 IP, one could argue that EZH2 is a sub-stoichiometric subunit of PRC2.

This is wrong. One cannot “argue that EZH2 is a sub-stoichiometric subunit of PRC2” since there are 25 years of data to show this is an incorrect statement. As shown below, EZH2 is in a 1:1:1 stoichiometry with SUZ12 and EED within PRC2, as shown clearly in our Figure 3i below, based on the SUZ12 IP-MS:

The reviewer’s incorrect interpretation of the JARID2 IP-MS experiment presumably stems from the fact that there are relatively low levels of EZH2, SUZ12 and EED immunoprecipitated with JARID2 (iBAQ relative to bait between 5-10%). However, as mentioned in the manuscript and during the revision stages, this reflects a known biological phenomenon: JARID2 can exist independently of PRC2⁸. Put clearly, the data in Figure 3j fit with a model where the majority of JARID2 is not in PRC2.

4) The artificial recruitment experiment in Extended Data Figure 9 clearly establishes that AEBP2 L and AEBP2 S can recruit PRC2 and trigger the deposition of H3K27me3 to a similar extent.

We agree that when artificially tethered to chromatin, AEBP2^L and AEBP2^S can recruit PRC2 and trigger the deposition of H3K27me3 to a similar extent, which is what this experiment was designed to demonstrate. The results shown in Extended Data Figure 9 confirm that the inhibitory effect of the AEBP2^L N-terminus is specific to impairing PRC2 binding to chromatin (as characterised in Figures 1-2, 4 and 6), and not related to inhibiting the intrinsic histone methyltransferase activity of PRC2. In the tethering assay, AEBP2^L is artificially recruited to chromatin via TetR:tetO interactions, which “bypass” the normal inhibitory function of its N-terminal region. Crucially, the function of the N-terminus of AEBP2^L is never functionally bypassed in normal cells, in the absence of artificial tethering. In all other experiments in our study — conducted without artificial tethering (Figures 1-2, 4 and 6) — the AEBP2^L N-terminus clearly impairs PRC2 binding to chromatin, and, as a result, PRC2 recruitment and H3K27me3 deposition.

This does not in any way support the conclusion that the two isoforms “perform opposite functions,” only that they perform the same function when artificially tethered.

This is a clear misrepresentation. As explained above and thoroughly detailed in the manuscript and rebuttal, the experiment in Extended Data Figure 9 was intentionally designed to demonstrate that that AEBP2^L inhibits PRC2 chromatin binding, rather than

preventing HMTase. Specifically, only by artificially tethering AEBP2^L to chromatin — thereby bypassing its inhibitory activity — can it mimic the activity of AEBP2^S. We very clearly explained the rationale for this experiment in the Results section (lines 462-464), as follows: “If AEBP2^L inhibits PRC2.2 solely by antagonising its ability to bind chromatin in cells, then one would expect that this effect (Fig. 2 and 4) can be nullified by forcing chromatin interaction through ectopic tethering”. Our finding in Extended Data Figure 9, that artificially tethered “AEBP2^L and AEBP2^S can recruit PRC2 and trigger the deposition of H3K27me3 to a similar extent”. This experiment, therefore, establish clearly that AEBP2^L inhibits PRC2.2 solely by antagonising its ability to bind chromatin in cells.

In short, Extended Data Figure 9 in no way contradicts our conclusion that the isoforms have opposing functions under physiological conditions; it simply demonstrates that AEBP2^L inhibits PRC2.2 specifically by preventing its chromatin binding.

This is consistent with the nature of the difference the authors attribute to the two isoforms (chromatin binding) but is also compatible with the isoforms having similar functions, as suggested by remarks 1 and 2 above.

This is wrong. This argument ignores the data across Figures 1-2, 4 and 6, demonstrating that the N-terminal region of AEBP2^L inhibits chromatin binding. The tethering experiment, as explained immediately above, must be discussed in the context of the rest of the manuscript: it is merely an artificial tethering experiment that was designed to test the conclusion that the ability of AEBP2^L to inhibit PRC2 is restricted to its ability to bind chromatin.

In summary, we respectfully ask the editors to re-evaluate the editorial decision in light of our clarifications regarding the wrong and misleading comments made by Reviewer 1, as well as the clear recommendations for publication from Reviewers 2 and 3.

References:

1. Højfeldt, J. W. *et al.* Non-core Subunits of the PRC2 Complex Are Collectively Required for Its Target-Site Specificity. *Mol. Cell* **76**, 423-436.e3 (2019).
2. Healy, E. *et al.* PRC2.1 and PRC2.2 Synergize to Coordinate H3K27 Trimethylation. *Mol. Cell* **76**, 437-452.e6 (2019).
3. Orlando, D. A. *et al.* Quantitative ChIP-Seq normalization reveals global modulation of the epigenome. *Cell Rep.* **9**, 1163–1170 (2014).
4. Grijzenhout, A. *et al.* Functional analysis of AEBP2, a PRC2 Polycomb protein, reveals a Trithorax phenotype in embryonic development and in ESCs. *Development* **143**, 2716–2723 (2016).
5. Conway, E. *et al.* A Family of Vertebrate-Specific Polycombs Encoded by the LCOR/LCORL Genes Balance PRC2 Subtype Activities. *Mol. Cell* (2018). doi:10.1016/j.molcel.2018.03.005
6. Sparbier, C. E. *et al.* Targeting Menin disrupts the KMT2A/B and polycomb balance to paradoxically activate bivalent genes. *Nat. Cell Biol.* 2023 252 **25**, 258–272 (2023).

7. Matsuwaka, M., Kumon, M. & Inoue, A. H3K27 dimethylation dynamics reveal stepwise establishment of facultative heterochromatin in early mouse embryos. *Nat. Cell Biol.* 2024 271 **27**, 28–38 (2024).
8. Al-Raawi, D. *et al.* A novel form of JARID2 is required for differentiation in lineage-committed cells. *EMBO J.* **38**, (2019).

Dear Prof. Bracken,

Thank you for submitting your manuscript which was previously reviewed at a different venue to The EMBO Journal. Your study has now been seen by an arbitrating referee, who finds that the remaining concerns by the original referees have been addressed and recommends publication of the manuscript. There remain only a few mainly editorial points that have to be addressed before I can extend formal acceptance of the manuscript:

- Please upload the manuscript as .docx file and remove the figures and "track changes"
 - Please double-check to make sure to all relevant funding information in the manuscript is also entered into our submission system. (Missing in the system currently: NHMRC Investigator EL1 grant (APP1196365); NIH-NIMH (R01MH122565) and Start-up funding from the Norris Comprehensive Cancer Center at Keck School of Medicine of USC; The Vermeulen lab is part of the Onco institute, which is partly funded by the Dutch Cancer Society (KWF)
 - Please reduce the number of keywords on the abstract page to five (ideally choosing broad general terms).
 - Please adjust the format of the reference list and of the in-text citations according to EMBO Journal format (alphabetical order, author name et al + year.../up to 10 author names in the reference list before et al / please refer to our Guide to Authors for additional information on EMBO J reference format).
 - Please rename the Conflict of Interest section into "Disclosure and Competing Interests Statement", in accordance with our updated Guide to Authors (<https://www.embopress.org/competing-interests>)
 - As we are switching from a free-text author contribution statement towards a more formal statement based on Contributor Role Taxonomy (CRediT) terms, please remove the present Author Contribution section and instead specify each author's contribution(s) directly in the Author Information page of our submission system during upload of the final manuscript. See <https://casrai.org/credit/> for more information.
 - Please list all figure callouts sequentially
 - Please provide either a "Yes" or a "Not Applicable" answer to each one of the questions in your Author Checklist (attached for your convenience) as you have not answered any of them yet. In the last column of this checklist, only the sections of the manuscript where the relevant information can be found should be listed (the information per se should be included in the main manuscript file).
 - Please upload the main and extended view figures as individual, high-resolution Figure files; figure legends should remain in ms file below the References; nomenclature should be Figure EV1-EV9 instead of Extended Data Figure 1-9
 - Please update all source file names, titles, legends and manuscript callouts to Dataset EV1-EV# instead of Table S1-S4, legends should be removed from ms and uploaded as a separate tab/sheet in each Excel file
 - Please provide the Reagent and Tools Table. For more information, please check <https://www.embopress.org/page/journal/14602075/authorguide#structuredmethods> and download the template for Reagent Table
 - Please provide suggestions for a short 'blurb' text prefacing and summing up the conceptual aspect of the study in two sentences (max. 250 characters), followed by 3-5 one-sentence 'bullet points' with brief factual statements of key results of the paper; they will form the basis of an editor-written 'Synopsis' accompanying the online version of the article. Please also provide an altered synopsis image, making sure that the aspect ratio conforms to our website's format - it should be exactly 550 pixels wide and between 300-600 pixels high.
 - Please describe the figure reused in EV Fig 1 and 1C as well as in Extended Data Figure 4 - D&E more clearly in the figure legends.
 - Please provide the specific URLs for GSE217538, 8EQV, EMD-28547, PXD053693 datasets are not provided in the data availability statement.
- Figure Legends (main + EV):
1. Please define the annotated p values ****/****/**/ as well as provide the exact p-values for the same in the legend of figure 3M as appropriate.
 2. Please note that the exact p values are not provided in the legends of figures 4C, 5G
 3. Please note that the box plots need to be defined in terms of minima, maxima, centre, bounds of box and whiskers, and percentile in the legends of figures 3M, 4C, EV5 E, F
 4. Please note that information related to n is missing in the legends of figures 3I, J; 6D, EV2 E, EV8 B
 5. Please note that the error bars are not defined in the legends of figures 3I, J; 6D; EV2 C, EV8 B
 6. Please note that the measure of center for the error bars needs to be defined in the legends of figures 3L; EV2 D, EV5 A
- Please remove the Supplementary data and Table legends should be removed from ms
 - Table 1 should only be included in ms file, placed between main and EV figure legends, it does not need to be uplidd as an

individual file as well

- Sections need to be named and the order should be corrected: Title page - Abstract - Keywords - Introduction - Results - Discussion - Methods - Data Availability - Acknowledgements - Disclosure and Competing Interests Statement - References - Figure Legends - Table(s) - Expanded View Figure Legends.

With best regards,
Cornelius Schneider

Cornelius Schneider, PhD
Editor | The EMBO Journal
c.schneider@embojournal.org

Use the link below to submit your revision:

Referee #1:

Dear Cornelius,

I apologise for the delay in providing my feedback on the manuscript EMBOJ-2025-121717. I have now reviewed all the documents you provided for evaluation.

I concur with your assessment that Dr. Bracken's response to Review #1 is fair and scientifically sound. I have observed Dr. Bracken present the results at a recent Keystone meeting, and the question session indicated that the work generated significant interest.

In conclusion, I believe that this manuscript is suitable for publication in EBMO J without any further modifications.

All editorial and formatting issues were resolved by the authors.

Dear Prof. Bracken,

I am pleased to inform you that your manuscript has been accepted for publication in the EMBO Journal.

Yours sincerely,

Cornelius Schneider, PhD
Editor
The EMBO Journal
c.schneider@embojournal.org
